# A Universal Source-Free Class Unlearning Framework via Synthetic Embeddings

## Abstract

Class unlearning in neural classifiers refers to selectively removing the model's ability to recognize a target (forget) class by reshaping the decision boundaries. This is essential when taxonomies change, labels are corrected, or legal or ethical requirements mandate class removal. The objective is to preserve performance on the remaining (retain) classes while avoiding costly full retraining. Existing methods generally require access to the source, i.e., forget/retain data or a relevant surrogate dataset. This dependency limits their applicability in scenarios where access to source data is restricted or unavailable. Even the recent source-free class unlearning methods rely on generating samples in the data space, which is computationally expensive and not even essential for doing class unlearning. In this work, we propose a novel source-free class unlearning framework that enables existing unlearning methods to operate using only the deployed model. We show that, under weak assumptions on the forget loss with respect to logits, class unlearning can be performed source-free for any given neural classifier by utilizing randomly generated samples within the classifier's intermediate space. Specifically, randomly generated embeddings classified by the model as belonging to the forget or retain classes are sufficient for effective unlearning, regardless of their marginal distribution. We validate our framework on four backbone architectures, ResNet-18, ResNet-50, ViT-B-16, and Swin-T, across three benchmark datasets, CIFAR-10, CIFAR-100, and TinyImageNet. Our experimental results show that existing class unlearning methods can operate within our source-free framework, with minimal impact on their forgetting efficacy and retain class accuracy.

## 1 Introduction

Deep learning models have achieved remarkable performance across domains, but their tendency to memorize training data makes them susceptible to privacy attacks such as membership inference attacks (Salem et al., 2018; Shokri et al., 2017; Song et al., 2019; Yeom et al., 2018) and model inversion attacks (Chen et al., 2021; Fredrikson et al., 2015). These risks pose serious concerns in privacy-sensitive applications, particularly under regulations such as General Data Protection Regulation (GDPR) (Voigt & Von dem Bussche, 2017) and California Consumer Privacy Act (CCPA) (Goldman, 2020) that mandate a "right to be forgotten", requiring effective removal of specific data from trained models. In response, machine unlearning has emerged as a promising direction to remove the influence of specific instances or classes without retraining from scratch. Unlearning methods fall into model-intrinsic (Lin et al., 2023), data-driven (Bourtoule et al., 2021; Hayase et al., 2020), and model-agnostic categories (Kurmanji et al., 2023; Chen et al., 2023; Cotogni et al., 2023; Cha et al., 2024), with a key distinction between exact unlearning (Bourtoule et al., 2021; Yan et al., 2022) and approximate unlearning. Although recent approximate methods reduce retraining overhead, most still require access to the forget set, the retain set, or a surrogate dataset that approximates the training distribution.

This work challenges the widely held assumption that access to original training data is required for class unlearning. Existing source-free class-unlearning methods still reconstruct input-level samples by training generative models, under the assumption that realistic or adversarial surrogates are needed to approximate the decision boundaries. This design makes the unlearning pipeline computationally heavy, tightly coupled to specific generator architectures, and in some cases dependent on additional surrogate models or datasets. We propose a novel framework for source-free class

unlearning that operates entirely without access to original or surrogate forget and retain datasets. Our approach leverages randomly generated embeddings in the intermediate space of the target classifier. More precisely, we generate synthetic, class-conditional synthetic embeddings by randomly sampling in the model's intermediate embedding space and pseudo-labeling them based on the model's predictions. These synthetic embeddings serve as proxies, allowing existing state-of-the-art unlearning methods to be adapted seamlessly to a fully source-free setting. We theoretically prove that these synthetic embeddings are sufficient to induce effective decision boundary adjustments, while preserving accuracy on the retain classes.

This work enables class-level unlearning in a fully source-free setting, which is compatible with a wide range of existing unlearning methods. Our framework successfully adapts several state-of-the-art techniques, including Finetuning (Golatkar et al., 2020), Negative Gradient (Golatkar et al., 2020), Negative Gradient+ (Kurmanji et al., 2023), Random Labels (Hayase et al., 2020), Boundary Expanding (Chen et al., 2023), Boundary Shrink (Chen et al., 2023), DELETE (Zhou et al., 2025), SCRUB (Kurmanji et al., 2023), and SCAR (Bonato et al., 2024), to operate effectively without requiring access to any original training data or relevant surrogate. Our main contributions are summarized as follows:

- We propose a novel *source-free* class unlearning framework that operates solely on a target model and the label of the class to be forgotten, without requiring any access to original, surrogate, or validation dataset. Our method generates synthetic class-conditional embeddings by sampling random vectors within the model's intermediate feature space and pseudo-labeling them using the model itself, enabling the adaptation of existing unlearning methods to a fully source-free regime.
- We show that these synthetic embeddings, regardless of their marginal distribution, are sufficient to induce the decision boundary shifts necessary for effective class unlearning. Remarkably, under our framework, multiple state-of-the-art unlearning techniques perform equivalently well as in data-access settings.
- We empirically validate our framework on ResNet-18, ResNet-50, ViT-B-16, and Swin-T backbones using CIFAR-10, CIFAR-100, and TinyImageNet datasets. The results show that a wide range of existing unlearning methods can function within our source-free setting with minimal degradation in the unlearning performance.

## 2 RELATED WORKS

Class unlearning aims to remove the influence of a target class from a trained model while preserving performance on the remaining classes. Class unlearning methods differ mainly by data access during unlearning: availability of retain data, forget data, both, or neither.

**Methods requiring both retain and forget sets.** Many effective class unlearning methods assume access to both forget and retain datasets. Distillation-based approaches such as SCalable Remembering and Unlearning unBound (SCRUB) (Kurmanji et al., 2023) guide student models via knowledge transfer and pruning. Machine Unlearning with Dimensional Alignment (MUDA) (Seo et al., 2025) introduces dimensional alignment loss and a self-distillation scheme that explicitly leverages both forget and retain sets to erase the influence of forget samples while preserving retain knowledge. The recently proposed SVD-based method (Kodge et al., 2024) performs gradient-free, single-step class unlearning by estimating retain and forget spaces from small subsets of both datasets and suppressing class-discriminatory activations.

**Retain-free methods.** These approaches remove dependence on retain data and operate mainly on forget samples. Negative Gradient reverses the estimated contribution of forget samples to the weights (Golatkar et al., 2020). Boundary Shrink and Boundary Expanding techniques (Chen et al., 2023) adjust decision boundaries by contracting or expanding regions related to forget samples. Partially Blinded Unlearning (PBU) (Panda et al., 2025) perturbs model parameters using a Bayesian loss. Other lines estimate the retain Hessian from forget data and model parameters (Ahmed et al., 2025), or inject targeted label noise to induce misclassification with minimal updates (Ye et al., 2025). Just in Time unlearning (JiT) enforces local Lipschitz regularization on forget samples and their perturbations (Foster et al., 2024), while zero-shot proxy generation synthesizes adversarial retain surrogates followed by subspace projection and pseudo-labeling (Chen et al., 2025). From an

input-sensitivity view, Machine Unlearning by Minimizing input sensitivity (MU-Mis) minimizes the sensitivity gap between target-class and irrelevant-class logits to withdraw forget influence with limited utility loss (Cheng et al., 2024). Zhou et al. (2025) proposes DELETE, a decoupled distillation method that suppresses the forget-class logits with a masking function and distills dark knowledge from the frozen model to preserve remaining classes. Recently, Selective-distillation for Class and Architecture-agnostic unleaRning (SCAR) (Bonato et al., 2024) introduced a retain-free method that leverages Mahalanobis-guided metric learning and a distillation strategy using a surrogate out-of-distribution dataset to preserve model performance. In addition, it proposes a source-free class unlearning variant that requires no access to either retain or forget data, while still relying on the surrogate dataset.

**Forget-free methods.** Some methods operate using retain data and without direct access to forget samples. Fine-tuning approaches update models exclusively on retain data to indirectly remove forget sample influence. Recent work, such as RELOAD (Newatia et al.), introduces blind unlearning, which performs approximate unlearning without access to the forget set. Instead, it leverages cached gradients from the original training and selectively re-initializes parameters most influenced by the forget data, guided by differences between full and retain gradients. Similarly, Unlearning With Single Pass Impair and Repair (UNSIR) (Tarun et al., 2023) operates in a zero-glance setting, where forget samples are entirely inaccessible. More precisely, it employs a single-pass impair-repair strategy using error-maximizing noise and a small retain subset to forget class-level information.

**Source-free methods.** In the source-free unlearning setting, neither forget nor retain data is available. Chundawat et al. (2023) proposes Min–Max noise, which adversarially perturbs weights to raise loss on forget classes while preserving retain accuracy, and Gated Knowledge Transfer (GKT), which distills a student from a teacher while filtering synthetic samples linked to the forget classes. GKT, however, can over-filter (discarding samples that still encode retain information) and exhibits generator imbalance (overproducing forget-class samples), reducing data efficiency. To address these issues, Zhang et al. (2025) introduces the Inhibited Synthesis PostFilter (ISPF) framework, combining Inhibited Synthesis to discourage the generation of forget-class data with a PostFilter to suppress forget-class logits without discarding samples. However, both approaches initialize and train a new model from scratch as part of the distillation process, which incurs substantial computational overhead. Wang et al. proposes Data Synthesis–based Discrimination-Aware (DSDA), which synthesizes data via Accelerated Energy-Guided Langevin Sampling and performs unlearning through Discrimination-Aware Multitask Optimization. Despite efficiency gains, DSDA still incurs nontrivial computational overhead due to the recursive sampling needed to construct synthetic forget and retain datasets. We demonstrate that synthesizing input-level data is not necessary for effective class unlearning, and intermediate random embeddings are sufficient to reshape the decision boundaries. Building on this insight, our proposed framework operates entirely in the intermediate embedding space by sampling synthetic embeddings and pseudo-labeling them using the model itself. This significantly reduces computational overhead while maintaining unlearning effectiveness. Compared to recent source-free methods such as DSDA, ISPF, and GKT, this approach avoids data generators, input reconstruction, and student-teacher training, making it significantly more efficient.

## 3 METHODOLOGY

In this section, we introduce our notations, formalize the problem setting, and lay down the theoretical foundation necessary for source-free class unlearning. Subsequently, we propose our source-free unlearning methodology grounded on this theoretical insight.

### 3.1 NOTATIONS AND PROBLEM SETUP

Consider a pre-trained classifier model defined as $\Phi = h \circ g \circ e$. Here, the feature extractor $e : \mathcal{X} \to \mathbb{R}^d$, parameterized by $\theta_e$, maps input samples $\mathbf{x} \in \mathcal{X}$ to a $d$-dimensional embedding $\mathbf{z} = e(\mathbf{x}) \in \mathbb{R}^d$. An intermediate transformation $g : \mathbb{R}^d \to \mathbb{R}^l$, parameterized by $\theta_g$, then maps $\mathbf{z}$ to an $l$-dimensional latent embedding $g(\mathbf{z}) \in \mathbb{R}^l$. Finally, the classifier head $h : \mathbb{R}^l \to \mathbb{R}^C$, with parameters $\theta_h$ computes class logits $h(g(\mathbf{z})) \in \mathbb{R}^C$. We denote the space of class labels as $\mathcal{Y} = \mathcal{Y}_f \cup \mathcal{Y}_r$, where $\mathcal{Y}_f$ is the set of classes targeted for unlearning (forget classes), and $\mathcal{Y}_r$ is the set of retain classes with $\mathcal{Y}_f \cap \mathcal{Y}_r = \emptyset$. In this work, we primarily focus on unlearning a single class, denoted as $c_f$, and thus $\mathcal{Y}_f = \{c_f\}$ and $\mathcal{Y}_r = \mathcal{Y} \setminus \{c_f\}$. Under this notation, *class unlearning* is defined as the process of selectively

removing the model's ability to recognize the target class $c_f$ by reshaping the decision boundary, while preserving predictive performance on the remaining classes $\mathcal{Y}_r$.

## 3.2 Proposed Methodology

We assume availability of embeddings drawn from an arbitrary intermediate embedding space, such as the output of the feature extractor $e$. Formally, we denote embeddings in this space as random variables $\mathbf{z} \in \mathbb{R}^d$, sampled from an arbitrary distribution $p_{\mathbf{z}}(\mathbf{z})$. These embeddings do not necessarily follow any particular distribution from the original training data. More precisely, given a classifier model $\Phi = h \circ g \circ e$, we obtain pseudo-labels for each embedding $\mathbf{z}_i$ by applying the intermediate transformation and the classifier head:

$$\hat{y}_i = \arg \max_{k \in \mathcal{Y}} [h(g(\mathbf{z}_i))]_k. \tag{1}$$

Using these pseudo-labels, we construct two embedding subsets including the forget set $\mathcal{E}_f$ and the retain set $\mathcal{E}_r$, defined as follows:

$$\mathcal{E}_f = \{\mathbf{z}_i \in \mathbb{R}^d \mid \hat{y}_i = c_f\}_{i=1}^{N_f}, \tag{2}$$

$$\mathcal{E}_r = \{\mathbf{z}_i \in \mathbb{R}^d \mid \hat{y}_i \in \mathcal{Y}_r\}_{i=1}^{N_r}, \tag{3}$$

where $N_f$ and $N_r$ are the sizes of the forget and retain sets, respectively. In class unlearning methods, the overall objective is often formulated as a combination of two components: a forget loss $\mathcal{L}_f$ computed on the forget set $\mathcal{E}_f$, and a retain loss $\mathcal{L}_r$ computed on the retain set $\mathcal{E}_r$. The total unlearning loss is typically expressed as $\mathcal{L}_u = \mathcal{L}_f + \lambda \mathcal{L}_r$, where $\lambda$ controls the trade-off between forgetting and utility preservation. The forget loss $\mathcal{L}_f$ encourages the model to remove knowledge related to the forget class by reshaping the decision boundary, while the retain loss $\mathcal{L}_r$ is used to preserve performance on the retain classes and prevent catastrophic forgetting. In the following proposition, we theoretically prove that by having access solely to these sets of embeddings—independent of the underlying embedding distribution $p_{\mathbf{z}}(\mathbf{z})$—it is possible to perform class unlearning effectively.

**Assumptions:** We begin by stating two assumptions regarding the forget loss function $\mathcal{L}_f$. First, we assume that $\mathcal{L}_f$ is differentiable with respect to the model's parameters. Second, we assume monotonicity conditions on the logits produced by the classifier head. Specifically, for every embedding $\mathbf{z}_i \in \mathcal{E}_f$:

$$\frac{\partial \mathcal{L}_f}{\partial [h(g(\mathbf{z}_i))]_k} \begin{cases} > 0 & k = c_f \quad \text{(monotonically increasing),} \\ < 0 & k \in \mathcal{Y}_r \quad \text{(monotonically decreasing),} \end{cases} \tag{4}$$

where $[h(g(\mathbf{z}_i))]_k = (\theta_h)_k^\top g(\mathbf{z}_i)$ denotes the logit for class $k$, and $(\theta_h)_k \in \mathbb{R}^l$ is the $k$-th row of classifier parameter matrix $\theta_h \in \mathbb{R}^{C \times l}$.

**Proposition 1** (Distribution-Agnostic Class Unlearning). *Consider a trained classifier model $\Phi = h \circ g \circ e$ with parameters defined as above, and assume the availability of the embedding sets $\mathcal{E}_f$ and $\mathcal{E}_r$ derived from an arbitrary embedding distribution $p_{\mathbf{z}}(\mathbf{z})$. Let class unlearning be performed by minimizing a forget loss function $\mathcal{L}_f$, defined over embeddings in $\mathcal{E}_f$. Then, class unlearning of the target class $c_f$ can be effectively achieved regardless of the choice of embedding distribution $p_{\mathbf{z}}(\mathbf{z})$.*

*Proof.* Since decision boundaries between classes are directly governed by the classifier parameters $\theta_h$, gradient-based updates explicitly reshape these boundaries. Consider a gradient descent update at iteration $j$ with learning rate $\alpha > 0$:

$$\theta_h^{(j+1)} = \theta_h^{(j)} - \alpha \frac{\partial \mathcal{L}_f}{\partial \theta_h^{(j)}}. \tag{5}$$

Applying the chain rule, the gradient of $\mathcal{L}_f$ with respect to $(\theta_h)_k$ is:

$$\frac{\partial \mathcal{L}_f}{\partial (\theta_h)_k^{(j)}} = \frac{1}{N_f} \sum_{\mathbf{z}_i \in \mathcal{E}_f} \frac{\partial \mathcal{L}_f}{\partial [h(g(\mathbf{z}_i))]_k} g(\mathbf{z}_i). \tag{6}$$

Thus, the update for the logit of class $k$ can be generally expressed as:

$$[h(g(\mathbf{z}_i))]_k^{(j+1)} = [h(g(\mathbf{z}_i))]_k^{(j)} - \frac{\alpha}{N_f} \sum_{\mathbf{z}_i \in \mathcal{E}_f} \frac{\partial \mathcal{L}_f}{\partial [h(g(\mathbf{z}_i))]_k} \|g(\mathbf{z}_i)\|^2. \tag{7}$$

By substituting the monotonicity assumption into equation 7, we have that the forget-class logit $[h(g(\mathbf{z}_i))]_{c_f}$ consistently decreases in response to $\mathbf{z}_i \in \mathcal{E}_f$, due to positive gradients. Conversely, logits corresponding to retain classes $k \in \mathcal{Y}_r$ consistently increase as their gradients are negative. Consequently, embeddings initially assigned to the forget class are systematically reclassified toward retain classes, progressively contracting the decision region associated with class $c_f$. Importantly, this reasoning relies only on embeddings classified as the forget, independent of their underlying distribution $p_\mathbf{z}(\mathbf{z})$. Hence, the effectiveness of class unlearning is guaranteed irrespective of the specific embedding distribution employed.

□

Building on Proposition 1, we propose a practical and fully source-free class unlearning framework. The central idea is to leverage synthetic embeddings sampled from an arbitrary distribution $p_\mathbf{z}(\mathbf{z})$ in the intermediate embedding space, using the classifier head to form synthetic forget and retain sets. These synthetic sets serve as surrogates for original data, enabling effective unlearning through gradient-based minimization of the forget loss $\mathcal{L}_f$. Figure 1 visually illustrates our proposed source-free unlearning pipeline, while Algorithm 1 summarizes the procedure in detail.

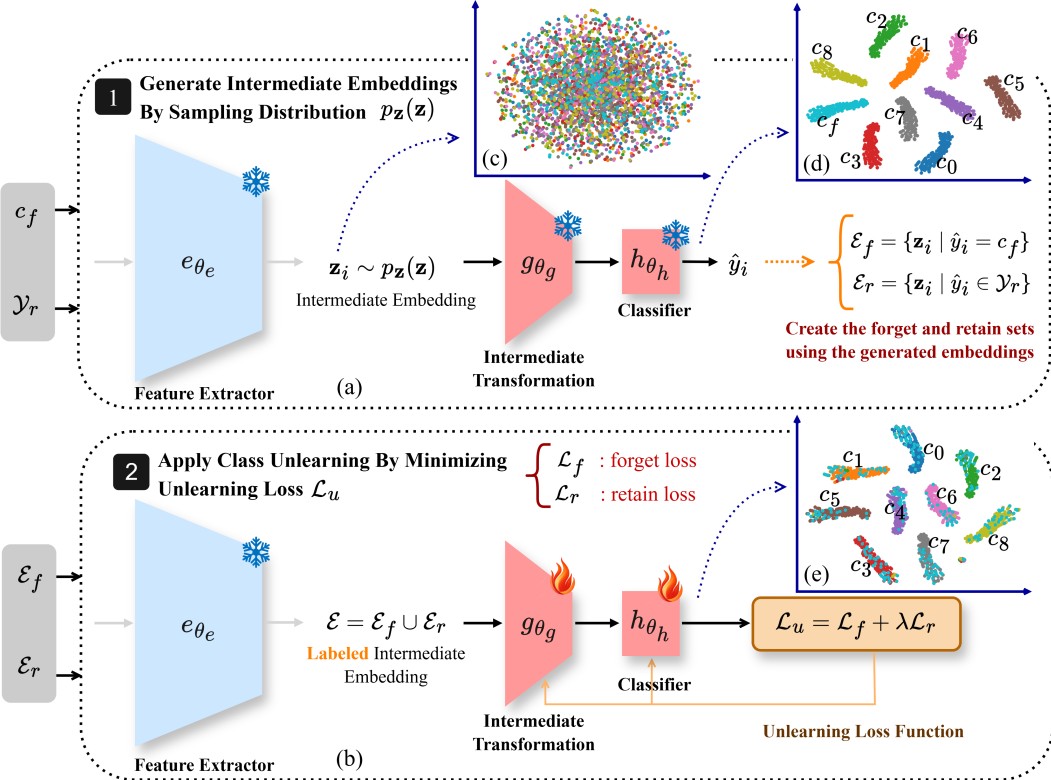

Figure 1: Illustration of the proposed source-free class unlearning framework. (a) **Step 1:** synthetic embeddings are sampled randomly from an arbitrary distribution in the intermediate embedding space and pseudo-labeled by the model to form the synthetic forget set $\mathcal{E}_f$ and retain set $\mathcal{E}_r$. (b) **Step 2:** the subsequent layers of the model are updated using these embeddings by minimizing the forget loss $\mathcal{L}_f$ to forget the target class set $\mathcal{Y}_f = \{c_f\}$, while optionally preserving performance on retain classes $\mathcal{Y}_r$ through the retain loss $\mathcal{L}_r$. (c) t-SNE of intermediate embeddings. (d) t-SNE of softmax probability before unlearning. (e) t-SNE of softmax probability after unlearning.

# 4 EXPERIMENTS

## 4.1 EXPERIMENTAL SETUP

We evaluate the efficacy of our proposed source-free framework by integrating it with a diverse set of state-of-the-art class unlearning methods, tested across three widely used benchmark datasets. Experiments are conducted using four backbone architectures, ResNet-18 (He et al., 2016), ResNet-50 (He et al., 2016), ViT-B-16 (Dosovitskiy et al., 2020), and Swin-T (Liu et al., 2021), although our framework is architecture-agnostic and can be extended to other network architectures without modification.

---

**Algorithm 1** Source-Free Class Unlearning Framework

---

**Require:** Pre-trained classifier model $\Phi = h \circ g \circ e$, target class to forget $c_f$, number of synthetic embeddings $N$, embedding distribution $p_{\mathbf{z}}(\mathbf{z})$, forget loss function $\mathcal{L}_f$, retain loss function $\mathcal{L}_r$, unlearning loss function $\mathcal{L}_u$, learning rate $\alpha$
1: **Initialize:** synthetic forget set $\mathcal{E}_f = \emptyset$ and retain set $\mathcal{E}_r = \emptyset$
2: **for** $i = 1$ to $N$ **do**
3:     Sample embedding $\mathbf{z}_i \sim p_{\mathbf{z}}(\mathbf{z})$
4:     Obtain pseudo-label: $\hat{y}_i = \arg\max_{k \in \mathcal{Y}}[h(g(\mathbf{z}_i))]_k$
5:     **if** $\hat{y}_i = c_f$ **then**
6:         $\mathcal{E}_f \leftarrow \mathcal{E}_f \cup \{\mathbf{z}_i\}$
7:     **else**
8:         $\mathcal{E}_r \leftarrow \mathcal{E}_r \cup \{\mathbf{z}_i\}$
9:     **end if**
10: **end for**
11: **for** each gradient update step **do**
12:     Compute loss $\mathcal{L}_u = \mathcal{L}_f + \lambda \mathcal{L}_r$: compute $\mathcal{L}_f$ using $\mathcal{E}_f$ and $\mathcal{L}_r$ using $\mathcal{E}_r$
13:     Backpropagate and update parameters $\theta = (\theta_g, \theta_h)$ via $\theta \leftarrow \theta - \alpha \nabla_\theta \mathcal{L}_u$
14: **end for**
15: **return** updated model $\Phi' = h' \circ g' \circ e$

---

**Datasets** —We conduct experiments on CIFAR-10 (Krizhevsky et al., 2009), CIFAR-100 (Krizhevsky et al., 2009), and TinyImageNet (Le & Yang, 2015). CIFAR-10 and CIFAR-100 comprise 60,000 color images of resolution $32 \times 32$, split into 50,000 training and 10,000 testing samples, with 10 and 100 classes respectively. TinyImageNet contains 110,000 images of resolution $64 \times 64$, distributed across 200 classes, with 100,000 samples for training and 10,000 for testing. In this work, we utilize only the test sets of these datasets to evaluate the effectiveness of the unlearning methods within our source-free framework.

**Baselines** —We benchmark our approach against a comprehensive suite of methods, including classical retraining, fine-tuning-based unlearning, and recent state-of-the-art techniques such as Boundary Shrink (BS) (Chen et al., 2023), Boundary Expanding (BE) (Chen et al., 2023), DELETE (Zhou et al., 2025), SCRUB (Kurmanji et al., 2023), SCAR (Bonato et al., 2024), Negative Gradient (NG) (Golatkar et al., 2020), Negative Gradient+ (NG+) (Kurmanji et al., 2023), and Random Labels (RL) (Hayase et al., 2020). The *Original* models denote ResNet-18, ResNet-50, ViT-B-16, and Swin-T architectures trained on the full training set for 300 epochs with cosine annealing learning rate scheduling, serving as the baseline before unlearning. The *Retrained* models are trained from scratch for 200 epochs exclusively on the retain subset, representing an upper-bound performance as they have no exposure to data from the forget set.

**Evaluation Metrics** —We assess unlearning performance using three primary metrics, including retain test accuracy ($\mathcal{A}_r^t$), forget test accuracy ($\mathcal{A}_f^t$), and the Adaptive Unlearning Score (AUS) (Cotogni et al., 2023). The objective is to maximize $\mathcal{A}_r^t$, thereby preserving retain knowledge, while minimizing $\mathcal{A}_f^t$, indicating effective unlearning. The AUS combines these aspects into a single scalar score that balances utility and unlearning:

$$\text{AUS} = \left(1 - \left(\mathcal{A}_r^{\text{or}-t} - \mathcal{A}_r^{\text{un}-t}\right)\right) / \left(1 + \left|\mathcal{A}_f^{\text{ideal}-t} - \mathcal{A}_f^{\text{un}-t}\right|\right), \tag{8}$$

where $\mathcal{A}_r^{\mathrm{or}-t}$ is the retain test accuracy of the original model, $\mathcal{A}_r^{\mathrm{un}-t}$ and $\mathcal{A}_f^{\mathrm{un}-t}$ are the retain and forget test accuracies of the unlearned model respectively, and $\mathcal{A}_f^{\mathrm{ideal}-t}$ denotes the target forget accuracy (ideally zero). Higher AUS values indicate superior unlearning performance, i.e., effective forgetting while preserving the retain classes' accuracy.

## 4.2 MAIN RESULTS

For each dataset, we conduct experiments using five independently initialized models, applying class-wise unlearning separately to each class. Each experiment is repeated across five random seeds, and the results reported correspond to the mean and standard deviation aggregated over all classes and seeds. To ensure a fair comparison among unlearning methods, the number of synthetic samples generated per class matches the size of the original training class (see Appendix A for the required minimum number of synthetic embeddings). These synthetic embeddings are sampled from the intermediate feature space immediately preceding the model's classification head (see Appendix B for the effect of embedding distribution). The overall performance is summarized in Table 1 and Table 2. Across all methods, datasets, and backbone architectures, our source-free framework consistently achieves near-complete forgetting as indicated by the minimized forget test accuracy ($\mathcal{A}_f^t$), while maintaining strong classification accuracy on retain classes ($\mathcal{A}_r^t$). Moreover, the AUS obtained close approximations to retraining-based baselines with full access to the retain set. In addition, a detailed class-level evaluation of different unlearning methods within our source-free framework is provided in Appendix E an anonymized code link is provided in Appendix D.

**Impact of Embedding Location on Source-Free Unlearning** —To evaluate the flexibility of our framework, we examine how the depth at which synthetic embeddings are generated influences unlearning performance. Specifically, we compare embeddings produced at two distinct locations: (1) immediately preceding the classifier head, which serves as our default configuration, and (2) earlier in the network, e.g., before the final convolutional block within ResNet-18's layer 4. As reported in Table 3, embeddings generated at the earlier stage continue to deliver strong unlearning performance, with results closely matching those obtained from embeddings sampled before the classifier head (see Table 1). The marginal differences observed underscore the robustness of our method to the choice of embedding depth. Furthermore, synthetic embeddings consistently achieve competitive results when directly compared to original embeddings extracted from the same intermediate layer, indicating their effectiveness as surrogate representations. Collectively, these findings confirm that our framework supports effective unlearning at multiple depths within the network, offering a layer-agnostic capability that enhances adaptability to diverse architectural configurations, privacy considerations, and computational constraints, thereby broadening its practical applicability.

**Impact of the Number of Synthetic Embeddings per Class on Unlearning Performance** —We investigate how the number of synthetic embeddings generated per class influences the unlearning efficacy. To this end, the ResNet-18 trained on CIFAR-100 is considered in the main text, with additional results for ResNet-18 on CIFAR-10 and TinyImageNet, as well as ViT-B-16 on CIFAR-10 and CIFAR-100, provided in the Appendix C. As illustrated in Figure 2, increasing the number of synthetic samples consistently enhances retain class accuracy ($\mathcal{A}_r^t$) and the AUS, while reducing forget class accuracy ($\mathcal{A}_f^t$). This behavior indicates that generating a larger set of representative embeddings more effectively approximates the decision boundaries of the forget and retain classes, thereby improving source-free unlearning performances. Notably, performance gains saturate beyond a certain sample size, which means that generating additional synthetic embeddings beyond this point yields minimal improvement. This allows for efficient use of computational resources without compromising unlearning quality.

## 5 CONCLUSION

We introduced a novel source-free framework for class unlearning, which removes specific class knowledge from a trained model without requiring access to the original training data, including forget, retain, or surrogate sets. By leveraging the internal structure of the model to synthesize class-conditional embeddings, we enable the adaptation of various state-of-the-art unlearning techniques to a fully source-free regime. Our experiments demonstrate that the proposed approach retains high accuracy on retain classes while effectively forgetting the target class across multiple datasets

Table 1: Class unlearning performance for CIFAR-10, CIFAR-100, and TinyImageNet using ResNet-18 and ResNet-50 as the base architecture. Rows highlighted in gray represent our results using synthetic embeddings, while the corresponding non-shaded rows use original embeddings with the same method. Columns $\mathcal{D}_r$-free and $\mathcal{D}_f$-free indicate whether the method operates without access to the retain or forget set, respectively, with (✓) denoting true and (✗) denoting false.

| Method | $\mathcal{D}_r$ free | $\mathcal{D}_f$ free | CIFAR-10 $\mathcal{A}_r^t \uparrow$ | $\mathcal{A}_f^t \downarrow$ | AUS $\uparrow$ | CIFAR-100 $\mathcal{A}_r^t \uparrow$ | $\mathcal{A}_f^t \downarrow$ | AUS $\uparrow$ | TinyImageNet $\mathcal{A}_r^t \uparrow$ | $\mathcal{A}_f^t \downarrow$ | AUS $\uparrow$ |
|---|---|---|---|---|---|---|---|---|---|---|---|
| **ResNet-18:** | | | | | | | | | | | |
| Original | – | – | $86.58_{\pm1.83}$ | $86.58_{\pm6.67}$ | $0.537_{\pm0.020}$ | $78.16_{\pm1.07}$ | $78.16_{\pm11.15}$ | $0.564_{\pm0.037}$ | $71.30_{\pm0.29}$ | $71.30_{\pm12.46}$ | $0.587_{\pm0.045}$ |
| Retrained | – | – | $86.95_{\pm1.17}$ | $0.0_{\pm0.0}$ | $1.004_{\pm0.006}$ | $77.92_{\pm0.80}$ | $0.0_{\pm0.0}$ | $0.998_{\pm0.013}$ | $63.01_{\pm2.76}$ | $0.0_{\pm0.0}$ | $0.917_{\pm0.028}$ |
| FT (Golatkar et al., 2020) | ✗ | ✓ | $87.43_{\pm1.02}$ | $0.0_{\pm0.0}$ | $1.009_{\pm0.004}$ | $78.20_{\pm1.00}$ | $0.0_{\pm0.0}$ | $1.000_{\pm0.003}$ | $71.32_{\pm0.35}$ | $0.0_{\pm0.0}$ | $1.000_{\pm0.002}$ |
| | ✓ | ✓ | $87.37_{\pm1.11}$ | $0.0_{\pm0.0}$ | $1.008_{\pm0.003}$ | $78.29_{\pm1.04}$ | $0.0_{\pm0.0}$ | $1.001_{\pm0.001}$ | $71.25_{\pm0.32}$ | $0.0_{\pm0.1}$ | $0.999_{\pm0.001}$ |
| NG (Golatkar et al., 2020) | ✓ | ✗ | $87.31_{\pm1.13}$ | $0.0_{\pm0.0}$ | $1.007_{\pm0.003}$ | $78.28_{\pm1.07}$ | $0.0_{\pm0.0}$ | $1.001_{\pm0.001}$ | $71.36_{\pm0.30}$ | $0.0_{\pm0.0}$ | $1.001_{\pm0.001}$ |
| | ✓ | ✗ | $87.40_{\pm1.14}$ | $0.0_{\pm0.0}$ | $1.008_{\pm0.004}$ | $78.28_{\pm1.05}$ | $0.0_{\pm0.1}$ | $1.001_{\pm0.002}$ | $71.30_{\pm0.29}$ | $0.0_{\pm0.0}$ | $1.001_{\pm0.000}$ |
| RL (Hayase et al., 2020) | ✓ | ✗ | $87.43_{\pm1.16}$ | $0.0_{\pm0.0}$ | $1.008_{\pm0.004}$ | $78.36_{\pm1.05}$ | $0.0_{\pm0.0}$ | $1.002_{\pm0.001}$ | $71.35_{\pm0.32}$ | $0.0_{\pm0.0}$ | $1.001_{\pm0.001}$ |
| | ✓ | ✓ | $87.33_{\pm1.11}$ | $0.0_{\pm0.0}$ | $1.008_{\pm0.004}$ | $78.12_{\pm1.03}$ | $0.0_{\pm0.0}$ | $1.000_{\pm0.001}$ | $71.27_{\pm0.32}$ | $0.0_{\pm0.0}$ | $1.000_{\pm0.001}$ |
| BS (Chen et al., 2023) | ✓ | ✗ | $86.29_{\pm1.09}$ | $0.2_{\pm0.4}$ | $0.996_{\pm0.009}$ | $74.32_{\pm1.72}$ | $0.1_{\pm0.5}$ | $0.960_{\pm0.017}$ | $70.24_{\pm0.87}$ | $0.1_{\pm0.5}$ | $0.988_{\pm0.010}$ |
| | ✓ | ✓ | $87.37_{\pm1.16}$ | $0.0_{\pm0.0}$ | $1.008_{\pm0.004}$ | $77.25_{\pm1.05}$ | $0.1_{\pm0.8}$ | $0.990_{\pm0.011}$ | $70.36_{\pm0.99}$ | $0.0_{\pm0.1}$ | $0.991_{\pm0.009}$ |
| BE (Chen et al., 2023) | ✓ | ✗ | $84.72_{\pm1.61}$ | $0.5_{\pm1.2}$ | $0.977_{\pm0.021}$ | $71.23_{\pm2.43}$ | $0.1_{\pm0.6}$ | $0.930_{\pm0.024}$ | $62.67_{\pm2.68}$ | $1.3_{\pm2.0}$ | $0.902_{\pm0.030}$ |
| | ✓ | ✓ | $86.51_{\pm0.81}$ | $0.0_{\pm0.0}$ | $0.999_{\pm0.001}$ | $78.02_{\pm1.10}$ | $0.0_{\pm0.0}$ | $0.999_{\pm0.003}$ | $71.23_{\pm0.30}$ | $0.0_{\pm0.0}$ | $0.999_{\pm0.001}$ |
| DELETE (Zhou et al., 2025) | ✓ | ✗ | $87.33_{\pm1.12}$ | $0.0_{\pm0.0}$ | $1.008_{\pm0.004}$ | $78.28_{\pm1.06}$ | $0.0_{\pm0.0}$ | $1.001_{\pm0.001}$ | $71.43_{\pm0.30}$ | $0.0_{\pm0.0}$ | $1.001_{\pm0.000}$ |
| | ✓ | ✓ | $87.36_{\pm1.13}$ | $0.0_{\pm0.0}$ | $1.008_{\pm0.004}$ | $78.26_{\pm1.07}$ | $0.0_{\pm0.0}$ | $1.001_{\pm0.001}$ | $71.36_{\pm0.30}$ | $0.0_{\pm0.0}$ | $1.001_{\pm0.000}$ |
| NG+ (Kurmanji et al., 2023) | ✗ | ✗ | $85.31_{\pm9.73}$ | $0.0_{\pm0.0}$ | $0.987_{\pm0.095}$ | $77.57_{\pm6.40}$ | $0.0_{\pm0.0}$ | $0.994_{\pm0.062}$ | $71.21_{\pm0.86}$ | $0.0_{\pm0.0}$ | $0.999_{\pm0.008}$ |
| | ✓ | ✓ | $87.38_{\pm1.14}$ | $0.0_{\pm0.0}$ | $1.008_{\pm0.004}$ | $78.33_{\pm1.00}$ | $0.0_{\pm0.0}$ | $1.002_{\pm0.001}$ | $71.35_{\pm0.33}$ | $0.0_{\pm0.0}$ | $1.000_{\pm0.001}$ |
| SCRUB (Kurmanji et al., 2023) | ✗ | ✗ | $87.11_{\pm1.04}$ | $0.0_{\pm0.0}$ | $1.005_{\pm0.003}$ | $77.52_{\pm1.06}$ | $0.0_{\pm0.0}$ | $0.994_{\pm0.002}$ | $67.60_{\pm1.51}$ | $0.0_{\pm0.4}$ | $0.963_{\pm0.014}$ |
| | ✓ | ✓ | $87.45_{\pm1.17}$ | $0.0_{\pm0.0}$ | $1.009_{\pm0.004}$ | $78.22_{\pm1.01}$ | $0.0_{\pm0.0}$ | $1.001_{\pm0.001}$ | $71.15_{\pm0.37}$ | $0.0_{\pm0.0}$ | $0.999_{\pm0.001}$ |
| SCAR (Bonato et al., 2024) | ✗ | ✗ | $87.44_{\pm1.15}$ | $0.0_{\pm0.0}$ | $1.009_{\pm0.004}$ | $78.34_{\pm1.09}$ | $0.0_{\pm0.0}$ | $1.002_{\pm0.002}$ | $71.50_{\pm0.30}$ | $0.0_{\pm0.0}$ | $1.002_{\pm0.001}$ |
| | ✓ | ✓ | $87.38_{\pm1.12}$ | $0.0_{\pm0.0}$ | $1.008_{\pm0.004}$ | $78.33_{\pm1.05}$ | $0.0_{\pm0.0}$ | $1.002_{\pm0.001}$ | $71.41_{\pm0.30}$ | $0.0_{\pm0.0}$ | $1.001_{\pm0.000}$ |
| **ResNet-50:** | | | | | | | | | | | |
| Original | – | – | $88.28_{\pm0.86}$ | $88.28_{\pm5.92}$ | $0.532_{\pm0.017}$ | $82.62_{\pm0.79}$ | $82.62_{\pm9.29}$ | $0.549_{\pm0.029}$ | $75.91_{\pm1.25}$ | $75.91_{\pm11.32}$ | $0.571_{\pm0.038}$ |
| Retrained | – | – | $89.03_{\pm1.04}$ | $0.0_{\pm0.0}$ | $1.008_{\pm0.007}$ | $81.73_{\pm0.99}$ | $0.0_{\pm0.0}$ | $0.991_{\pm0.013}$ | $76.21_{\pm2.31}$ | $0.0_{\pm0.0}$ | $1.003_{\pm0.026}$ |
| FT (Golatkar et al., 2020) | ✗ | ✓ | $89.40_{\pm0.98}$ | $0.0_{\pm0.0}$ | $1.011_{\pm0.005}$ | $82.79_{\pm0.75}$ | $0.0_{\pm0.0}$ | $1.002_{\pm0.001}$ | $75.80_{\pm1.25}$ | $0.0_{\pm0.2}$ | $0.999_{\pm0.003}$ |
| | ✓ | ✓ | $88.98_{\pm1.03}$ | $0.0_{\pm0.0}$ | $1.007_{\pm0.003}$ | $82.68_{\pm0.77}$ | $0.0_{\pm0.0}$ | $1.001_{\pm0.001}$ | $75.80_{\pm1.29}$ | $0.0_{\pm0.0}$ | $0.999_{\pm0.001}$ |
| NG (Golatkar et al., 2020) | ✓ | ✗ | $88.96_{\pm1.66}$ | $0.0_{\pm0.0}$ | $1.005_{\pm0.013}$ | $82.71_{\pm0.79}$ | $0.0_{\pm0.0}$ | $1.001_{\pm0.001}$ | $75.97_{\pm1.24}$ | $0.0_{\pm0.0}$ | $1.001_{\pm0.000}$ |
| | ✓ | ✗ | $89.04_{\pm1.10}$ | $0.0_{\pm0.0}$ | $1.008_{\pm0.004}$ | $82.70_{\pm0.79}$ | $0.0_{\pm0.0}$ | $1.001_{\pm0.001}$ | $75.95_{\pm1.25}$ | $0.0_{\pm0.0}$ | $1.000_{\pm0.000}$ |
| RL (Hayase et al., 2020) | ✓ | ✗ | $89.06_{\pm1.07}$ | $0.0_{\pm0.0}$ | $1.008_{\pm0.003}$ | $82.72_{\pm0.79}$ | $0.0_{\pm0.0}$ | $1.001_{\pm0.001}$ | $75.95_{\pm1.24}$ | $0.0_{\pm0.0}$ | $1.000_{\pm0.001}$ |
| | ✓ | ✓ | $88.92_{\pm1.04}$ | $0.0_{\pm0.0}$ | $1.006_{\pm0.003}$ | $82.76_{\pm0.78}$ | $0.0_{\pm0.0}$ | $1.001_{\pm0.001}$ | $75.90_{\pm1.22}$ | $0.0_{\pm0.0}$ | $1.000_{\pm0.001}$ |
| BS (Chen et al., 2023) | ✓ | ✗ | $87.68_{\pm1.18}$ | $0.4_{\pm0.9}$ | $0.990_{\pm0.014}$ | $82.28_{\pm0.94}$ | $0.0_{\pm0.1}$ | $0.997_{\pm0.014}$ | $74.44_{\pm1.67}$ | $0.1_{\pm0.5}$ | $0.984_{\pm0.013}$ |
| | ✓ | ✓ | $89.24_{\pm0.97}$ | $0.0_{\pm0.0}$ | $1.007_{\pm0.003}$ | $82.55_{\pm0.80}$ | $0.0_{\pm0.0}$ | $0.999_{\pm0.001}$ | $75.19_{\pm1.21}$ | $0.0_{\pm0.0}$ | $0.993_{\pm0.002}$ |
| BE (Chen et al., 2023) | ✓ | ✗ | $87.44_{\pm1.56}$ | $0.3_{\pm0.9}$ | $0.989_{\pm0.015}$ | $82.14_{\pm0.85}$ | $0.0_{\pm0.0}$ | $0.995_{\pm0.002}$ | $68.12_{\pm2.81}$ | $0.5_{\pm1.2}$ | $0.917_{\pm0.021}$ |
| | ✓ | ✓ | $88.22_{\pm0.86}$ | $0.0_{\pm0.0}$ | $0.999_{\pm0.000}$ | $82.62_{\pm0.79}$ | $0.0_{\pm0.0}$ | $1.000_{\pm0.000}$ | $75.89_{\pm1.25}$ | $0.0_{\pm0.0}$ | $1.000_{\pm0.000}$ |
| DELETE (Zhou et al., 2025) | ✓ | ✗ | $88.99_{\pm1.06}$ | $0.0_{\pm0.0}$ | $1.007_{\pm0.004}$ | $82.71_{\pm0.79}$ | $0.0_{\pm0.0}$ | $1.001_{\pm0.000}$ | $75.98_{\pm1.24}$ | $0.0_{\pm0.0}$ | $1.001_{\pm0.000}$ |
| | ✓ | ✓ | $88.98_{\pm1.07}$ | $0.0_{\pm0.0}$ | $1.007_{\pm0.003}$ | $82.70_{\pm0.79}$ | $0.0_{\pm0.0}$ | $1.001_{\pm0.001}$ | $75.95_{\pm1.24}$ | $0.0_{\pm0.0}$ | $1.000_{\pm0.000}$ |
| NG+ (Kurmanji et al., 2023) | ✗ | ✗ | $89.12_{\pm1.00}$ | $0.0_{\pm0.0}$ | $1.008_{\pm0.004}$ | $82.78_{\pm0.77}$ | $0.0_{\pm0.0}$ | $1.002_{\pm0.001}$ | $76.24_{\pm1.06}$ | $0.0_{\pm0.0}$ | $1.001_{\pm0.001}$ |
| | ✓ | ✓ | $88.99_{\pm1.05}$ | $0.0_{\pm0.0}$ | $1.007_{\pm0.003}$ | $82.79_{\pm0.90}$ | $0.0_{\pm0.0}$ | $1.001_{\pm0.001}$ | $75.99_{\pm1.23}$ | $0.0_{\pm0.0}$ | $1.001_{\pm0.000}$ |
| SCRUB (Kurmanji et al., 2023) | ✗ | ✗ | $88.96_{\pm0.95}$ | $0.0_{\pm0.0}$ | $1.008_{\pm0.004}$ | $82.76_{\pm0.75}$ | $0.0_{\pm0.0}$ | $1.001_{\pm0.001}$ | $70.65_{\pm2.51}$ | $0.3_{\pm1.0}$ | $0.944_{\pm0.015}$ |
| | ✓ | ✓ | $89.11_{\pm1.10}$ | $0.0_{\pm0.0}$ | $1.008_{\pm0.004}$ | $82.72_{\pm0.77}$ | $0.0_{\pm0.0}$ | $1.001_{\pm0.001}$ | $75.86_{\pm1.28}$ | $0.0_{\pm0.0}$ | $0.999_{\pm0.001}$ |
| SCAR (Bonato et al., 2024) | ✗ | ✗ | $89.11_{\pm1.08}$ | $0.0_{\pm0.0}$ | $1.008_{\pm0.004}$ | $82.47_{\pm0.97}$ | $0.0_{\pm0.1}$ | $0.998_{\pm0.008}$ | $76.01_{\pm1.22}$ | $0.0_{\pm0.0}$ | $1.001_{\pm0.001}$ |
| | ✓ | ✓ | $89.02_{\pm1.07}$ | $0.0_{\pm0.0}$ | $1.007_{\pm0.003}$ | $82.73_{\pm0.79}$ | $0.0_{\pm0.0}$ | $1.001_{\pm0.001}$ | $76.04_{\pm1.24}$ | $0.0_{\pm0.0}$ | $1.001_{\pm0.000}$ |

Figure 2: Effect of the number of synthetic embeddings per class on unlearning performance. Results are averaged over three independently trained models, with class-wise unlearning performed separately for each class. Error bars indicate $95\%$ confidence intervals. Experiments use the ResNet-18 architecture on the CIFAR-100 dataset.

Table 2: Class unlearning performance for CIFAR-10, CIFAR-100, and TinyImageNet using ViT-B-16 and Swin-T as the base architecture. Rows highlighted in gray represent our results using synthetic embeddings, while the corresponding non-shaded rows use original embeddings with the same method. Columns $\mathcal{D}_r$-free and $\mathcal{D}_f$-free indicate whether the method operates without access to the retain or forget set, respectively, with (✓) denoting true and (✗) denoting false.

| Method | $\mathcal{D}_r$ free | $\mathcal{D}_f$ free | CIFAR-10 $\mathcal{A}_r^t\uparrow$ | $\mathcal{A}_f^t\downarrow$ | AUS↑ | CIFAR-100 $\mathcal{A}_r^t\uparrow$ | $\mathcal{A}_f^t\downarrow$ | AUS↑ | TinyImageNet $\mathcal{A}_r^t\uparrow$ | $\mathcal{A}_f^t\downarrow$ | AUS↑ |
|---|---|---|---|---|---|---|---|---|---|---|---|
| **ViT-B-16:** | | | | | | | | | | | |
| Original | – | – | $97.69_{\pm0.18}$ | $97.69_{\pm1.30}$ | $0.506_{\pm0.003}$ | $87.22_{\pm0.26}$ | $87.22_{\pm7.83}$ | $0.535_{\pm0.023}$ | $88.20_{\pm0.14}$ | $88.20_{\pm7.29}$ | $0.532_{\pm0.022}$ |
| Retrained | – | – | $98.38_{\pm0.21}$ | $0.0_{\pm0.0}$ | $1.007_{\pm0.002}$ | $88.74_{\pm0.21}$ | $0.0_{\pm0.0}$ | $1.015_{\pm0.003}$ | $89.59_{\pm0.13}$ | $0.0_{\pm0.0}$ | $1.014_{\pm0.002}$ |
| NG (Golatkar et al., 2020) | ✓ | ✗ | $97.89_{\pm0.25}$ | $0.0_{\pm0.0}$ | $1.002_{\pm0.001}$ | $87.29_{\pm0.27}$ | $0.0_{\pm0.0}$ | $1.001_{\pm0.001}$ | $88.23_{\pm0.14}$ | $0.0_{\pm0.0}$ | $1.000_{\pm0.000}$ |
| | ✓ | ✓ | $97.90_{\pm0.24}$ | $0.0_{\pm0.0}$ | $1.002_{\pm0.001}$ | $87.30_{\pm0.27}$ | $0.0_{\pm0.0}$ | $1.001_{\pm0.001}$ | $88.23_{\pm0.14}$ | $0.0_{\pm0.0}$ | $1.000_{\pm0.000}$ |
| RL (Hayase et al., 2020) | ✓ | ✗ | $97.91_{\pm0.25}$ | $0.0_{\pm0.0}$ | $1.002_{\pm0.001}$ | $87.31_{\pm0.28}$ | $0.0_{\pm0.0}$ | $1.001_{\pm0.001}$ | $88.24_{\pm0.14}$ | $0.0_{\pm0.0}$ | $1.000_{\pm0.000}$ |
| | ✓ | ✓ | $97.93_{\pm0.24}$ | $0.0_{\pm0.0}$ | $1.002_{\pm0.001}$ | $87.35_{\pm0.28}$ | $0.0_{\pm0.0}$ | $1.001_{\pm0.001}$ | $88.27_{\pm0.14}$ | $0.0_{\pm0.0}$ | $1.001_{\pm0.001}$ |
| BS (Chen et al., 2023) | ✓ | ✗ | $97.76_{\pm0.22}$ | $0.0_{\pm0.0}$ | $1.001_{\pm0.001}$ | $87.27_{\pm0.27}$ | $0.0_{\pm0.0}$ | $1.000_{\pm0.000}$ | $88.22_{\pm0.14}$ | $0.0_{\pm0.0}$ | $1.000_{\pm0.000}$ |
| | ✓ | ✓ | $97.89_{\pm0.23}$ | $0.0_{\pm0.0}$ | $1.002_{\pm0.001}$ | $87.22_{\pm0.28}$ | $0.0_{\pm0.0}$ | $1.000_{\pm0.001}$ | $88.08_{\pm0.16}$ | $0.0_{\pm0.1}$ | $0.999_{\pm0.001}$ |
| DELETE (Zhou et al., 2025) | ✓ | ✗ | $97.89_{\pm0.25}$ | $0.0_{\pm0.0}$ | $1.002_{\pm0.001}$ | $87.30_{\pm0.27}$ | $0.0_{\pm0.0}$ | $1.001_{\pm0.001}$ | $88.23_{\pm0.14}$ | $0.0_{\pm0.0}$ | $1.000_{\pm0.000}$ |
| | ✓ | ✓ | $97.91_{\pm0.25}$ | $0.0_{\pm0.0}$ | $1.002_{\pm0.001}$ | $87.32_{\pm0.27}$ | $0.0_{\pm0.0}$ | $1.001_{\pm0.001}$ | $88.25_{\pm0.14}$ | $0.0_{\pm0.0}$ | $1.001_{\pm0.000}$ |
| NG+ (Kurmanji et al., 2023) | ✗ | ✗ | $97.88_{\pm0.25}$ | $0.0_{\pm0.0}$ | $1.002_{\pm0.001}$ | $87.15_{\pm0.29}$ | $0.0_{\pm0.2}$ | $0.999_{\pm0.003}$ | $87.64_{\pm0.27}$ | $0.1_{\pm0.4}$ | $0.993_{\pm0.005}$ |
| | ✓ | ✓ | $97.92_{\pm0.25}$ | $0.0_{\pm0.0}$ | $1.002_{\pm0.001}$ | $87.32_{\pm0.30}$ | $0.0_{\pm0.0}$ | $1.001_{\pm0.001}$ | $88.28_{\pm0.15}$ | $0.0_{\pm0.0}$ | $1.001_{\pm0.000}$ |
| **Swin-T:** | | | | | | | | | | | |
| Original | – | – | $97.73_{\pm0.17}$ | $97.73_{\pm1.47}$ | $0.506_{\pm0.004}$ | $87.58_{\pm0.53}$ | $87.58_{\pm9.01}$ | $0.534_{\pm0.029}$ | $86.18_{\pm0.09}$ | $86.18_{\pm7.59}$ | $0.538_{\pm0.023}$ |
| Retrained | – | – | $98.36_{\pm0.23}$ | $0.0_{\pm0.0}$ | $1.006_{\pm0.001}$ | $88.89_{\pm0.21}$ | $0.0_{\pm0.0}$ | $1.013_{\pm0.005}$ | $87.13_{\pm0.13}$ | $0.0_{\pm0.0}$ | $1.010_{\pm0.002}$ |
| NG (Golatkar et al., 2020) | ✓ | ✗ | $97.93_{\pm0.27}$ | $0.0_{\pm0.0}$ | $1.002_{\pm0.001}$ | $87.65_{\pm0.54}$ | $0.0_{\pm0.0}$ | $1.001_{\pm0.001}$ | $86.21_{\pm0.10}$ | $0.0_{\pm0.0}$ | $1.000_{\pm0.000}$ |
| | ✓ | ✓ | $97.64_{\pm0.86}$ | $0.5_{\pm1.0}$ | $0.995_{\pm0.017}$ | $83.19_{\pm3.93}$ | $1.7_{\pm1.7}$ | $0.941_{\pm0.047}$ | $80.79_{\pm4.72}$ | $1.9_{\pm1.6}$ | $0.929_{\pm0.051}$ |
| NG+ (Kurmanji et al., 2023) | ✗ | ✗ | $97.83_{\pm0.27}$ | $0.0_{\pm0.0}$ | $1.001_{\pm0.001}$ | $87.60_{\pm0.54}$ | $0.0_{\pm0.0}$ | $1.000_{\pm0.002}$ | $84.46_{\pm1.19}$ | $0.0_{\pm0.3}$ | $0.982_{\pm0.012}$ |
| | ✓ | ✓ | $93.50_{\pm7.54}$ | $1.1_{\pm1.3}$ | $0.948_{\pm0.080}$ | $86.84_{\pm0.95}$ | $0.3_{\pm0.8}$ | $0.990_{\pm0.014}$ | $85.28_{\pm0.76}$ | $0.4_{\pm1.0}$ | $0.987_{\pm0.014}$ |
| SCRUB (Kurmanji et al., 2023) | ✗ | ✗ | $97.85_{\pm0.25}$ | $0.0_{\pm0.0}$ | $1.001_{\pm0.001}$ | $87.73_{\pm0.47}$ | $0.0_{\pm0.0}$ | $1.001_{\pm0.001}$ | $86.19_{\pm0.09}$ | $0.0_{\pm0.0}$ | $1.000_{\pm0.001}$ |
| | ✓ | ✓ | $97.39_{\pm1.11}$ | $0.0_{\pm0.0}$ | $0.997_{\pm0.011}$ | $87.07_{\pm0.65}$ | $0.0_{\pm0.3}$ | $0.995_{\pm0.007}$ | $84.92_{\pm0.73}$ | $0.1_{\pm0.4}$ | $0.987_{\pm0.008}$ |

Table 3: Class unlearning performance using random samples generated from layer 4 (immediately before the last convolutional layer) of ResNet-18 as the base architecture. Rows highlighted in gray show results obtained with synthetic embeddings.

| Method | $\mathcal{D}_r$ free | $\mathcal{D}_f$ free | CIFAR-10 $\mathcal{A}_r^t\uparrow$ | $\mathcal{A}_f^t\downarrow$ | AUS↑ | CIFAR-100 $\mathcal{A}_r^t\uparrow$ | $\mathcal{A}_f^t\downarrow$ | AUS↑ | TinyImageNet $\mathcal{A}_r^t\uparrow$ | $\mathcal{A}_f^t\downarrow$ | AUS↑ |
|---|---|---|---|---|---|---|---|---|---|---|---|
| Original | – | – | $86.58_{\pm0.83}$ | $86.58_{\pm6.67}$ | $0.537_{\pm0.020}$ | $78.16_{\pm1.07}$ | $78.16_{\pm11.15}$ | $0.564_{\pm0.037}$ | $71.30_{\pm0.29}$ | $71.30_{\pm12.46}$ | $0.587_{\pm0.045}$ |
| Retrained | – | – | $86.95_{\pm1.22}$ | $0.0_{\pm0.0}$ | $1.000_{\pm0.005}$ | $77.92_{\pm0.80}$ | $0.0_{\pm0.0}$ | $0.956_{\pm0.036}$ | $63.01_{\pm2.77}$ | $0.0_{\pm0.0}$ | $0.855_{\pm0.029}$ |
| FT (Golatkar et al., 2020) | ✗ | ✓ | $87.55_{\pm1.09}$ | $0.2_{\pm0.9}$ | $1.007_{\pm0.010}$ | $76.80_{\pm4.06}$ | $0.2_{\pm0.6}$ | $0.985_{\pm0.042}$ | $71.72_{\pm0.33}$ | $0.6_{\pm1.2}$ | $0.998_{\pm0.012}$ |
| | ✓ | ✓ | $81.03_{\pm3.82}$ | $0.0_{\pm0.1}$ | $0.944_{\pm0.037}$ | $76.09_{\pm1.10}$ | $0.0_{\pm0.3}$ | $0.979_{\pm0.009}$ | $69.64_{\pm0.46}$ | $0.0_{\pm0.0}$ | $0.983_{\pm0.002}$ |
| NG (Golatkar et al., 2020) | ✓ | ✗ | $87.30_{\pm1.23}$ | $0.0_{\pm0.0}$ | $1.007_{\pm0.005}$ | $78.29_{\pm1.08}$ | $0.0_{\pm0.0}$ | $1.001_{\pm0.001}$ | $70.51_{\pm1.02}$ | $0.1_{\pm0.5}$ | $0.991_{\pm0.011}$ |
| | ✓ | ✓ | $87.24_{\pm1.16}$ | $0.0_{\pm0.1}$ | $1.006_{\pm0.004}$ | $76.28_{\pm1.40}$ | $0.0_{\pm0.1}$ | $0.981_{\pm0.011}$ | $71.30_{\pm0.46}$ | $0.0_{\pm0.0}$ | $1.000_{\pm0.003}$ |
| RL (Hayase et al., 2020) | ✓ | ✗ | $87.27_{\pm1.08}$ | $0.0_{\pm0.0}$ | $1.007_{\pm0.003}$ | $78.32_{\pm1.06}$ | $0.0_{\pm0.0}$ | $1.002_{\pm0.001}$ | $71.56_{\pm0.39}$ | $0.0_{\pm0.0}$ | $1.003_{\pm0.001}$ |
| | ✓ | ✓ | $87.18_{\pm1.24}$ | $0.0_{\pm0.1}$ | $1.006_{\pm0.007}$ | $77.76_{\pm1.65}$ | $0.0_{\pm0.2}$ | $0.996_{\pm0.013}$ | $71.62_{\pm0.45}$ | $0.0_{\pm0.0}$ | $1.003_{\pm0.002}$ |
| DELETE (Zhou et al., 2025) | ✓ | ✗ | $77.62_{\pm15.23}$ | $0.4_{\pm0.8}$ | $0.905_{\pm0.150}$ | $75.97_{\pm4.21}$ | $0.1_{\pm0.6}$ | $0.978_{\pm0.039}$ | $54.84_{\pm6.63}$ | $1.4_{\pm1.7}$ | $0.819_{\pm0.069}$ |
| | ✓ | ✓ | $87.02_{\pm1.11}$ | $0.0_{\pm0.1}$ | $1.004_{\pm0.005}$ | $74.29_{\pm2.31}$ | $1.3_{\pm1.4}$ | $0.948_{\pm0.026}$ | $68.89_{\pm0.99}$ | $0.0_{\pm0.3}$ | $0.972_{\pm0.010}$ |
| NG+ (Kurmanji et al., 2023) | ✗ | ✗ | $83.82_{\pm0.70}$ | $0.0_{\pm0.0}$ | $0.972_{\pm0.010}$ | $78.20_{\pm1.01}$ | $0.0_{\pm0.1}$ | $1.000_{\pm0.002}$ | $70.41_{\pm0.44}$ | $0.0_{\pm0.0}$ | $0.991_{\pm0.003}$ |
| | ✓ | ✓ | $87.16_{\pm1.17}$ | $0.1_{\pm0.5}$ | $1.005_{\pm0.007}$ | $78.18_{\pm1.06}$ | $0.0_{\pm0.2}$ | $1.000_{\pm0.004}$ | $71.37_{\pm0.43}$ | $0.0_{\pm0.1}$ | $1.001_{\pm0.002}$ |

and unlearning strategies. The framework's compatibility with existing methods and complete independence from training data position it as a strong candidate for class unlearning in real-world scenarios. Future work includes extending this approach to instance-level unlearning and applying the technique to domains beyond image classification, such as language models.

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

# A DETERMINING THE MINIMUM NUMBER OF SYNTHETIC EMBEDDINGS FOR RELIABLE CLASS COVERAGE

In the proposed source-free settings, synthetic embeddings are generated by sampling random vectors in the classifier's intermediate embedding space. The underlying sampling distribution significantly influences predicted class distribution, often causing class imbalance. To address this, we employ a class-aware rejection sampling strategy that continues sampling until a predefined minimum number of samples is obtained for each class. This ensures a balanced synthetic dataset and establishes a stable basis for source-free unlearning. To guarantee sufficient representation of all target classes, we estimate the minimum number of synthetic samples $N$ required such that the probability of having at least one sample from a given class $c$ exceeds a confidence threshold $p$. We first generate a large pilot batch $\{z_i\}_{i=1}^{N_{\text{pilot}}}$ of embeddings sampled from an arbitrary distribution in the intermediate embedding space, and obtain their predicted labels $\hat{y}_i$. The empirical class probability for class $c$ is then estimated as

$$q_c = \frac{1}{N_{\text{pilot}}} \sum_{i=1}^{N_{\text{pilot}}} \mathbb{1}\{\hat{y}_i = c\}, \tag{9}$$

where $\mathbb{1}\{\cdot\}$ is the indicator function that equals one if the condition inside is true, and zero otherwise. Assuming independent sampling, the probability that none of the $N$ synthetic embeddings fall into class $c$ is $(1 - q_c)^N$. To ensure that at least one embedding belongs to class $c$ with confidence $p$, we require $1 - (1 - q_c)^N \geq p$, which yields

$$N \geq \frac{\ln(1 - p)}{\ln(1 - q_c)}, \tag{10}$$

where $\ln(1 - q_c) < 0$ ensures the inequality holds in the correct direction. This expression provides a principled estimate for the number of synthetic embeddings required to achieve class-wise coverage with the desired confidence level.

We empirically validate this estimate by reporting the minimum number of synthetic embeddings required to ensure, with high confidence, that at least one embedding is classified into each target class. Table 4 summarizes statistics computed for a ResNet-18 classifier on CIFAR-10, CIFAR-100, and TinyImageNet datasets, using Gaussian, Laplace, and Uniform embedding distributions. We report the lower bound, average, and upper bound for the total number of synthetic embeddings needed across all classes for each dataset and embedding distribution. These values correspond, respectively, to the easiest, average, and most difficult classes to cover. This analysis shows the impact of dataset complexity and embeddings distribution on sample requirements for achieving reliable class representation in source-free unlearning.

Table 4: Estimated minimum total number of synthetic embeddings required to guarantee, with high confidence, that a forget class is represented by at least one embedding. Results correspond to the ResNet-18 architecture evaluated on CIFAR-10, CIFAR-100, and TinyImageNet datasets, using Gaussian, Laplace, and Uniform distributions for embedding generation.

| Dataset | Embedding Distribution | Lower bound (across classes) | Average (across classes) | Upper bound (across classes) |
|---|---|---|---|---|
| CIFAR-10 | Gaussian | 32 | 46 | 55 |
| | Laplace | 33 | 46 | 53 |
| | Uniform | 29 | 48 | 60 |
| CIFAR-100 | Gaussian | 223 | 494 | 1041 |
| | Laplace | 269 | 483 | 822 |
| | Uniform | 139 | 544 | 1735 |
| TinyImageNet | Gaussian | 407 | 990 | 2550 |
| | Laplace | 427 | 987 | 2437 |
| | Uniform | 353 | 1011 | 2880 |

In the worst-case scenario, where the rarest class has empirical probability $q_{\text{min}}$, the minimum number of synthetic embeddings needed to ensure, with confidence $p$, that at least one embedding be-

longs to this class is $N_{\text{worst}} = \frac{\ln(1-p)}{\ln(1-q_{\min})}$. If a stricter criterion is imposed to require at least $m$ embeddings from this rarest class, the required number of embeddings increases significantly. This corresponds to solving

$$1 - \sum_{k=0}^{m-1} \binom{N}{k} q_{\min}^k (1 - q_{\min})^{N-k} \geq p, \tag{11}$$

which involves computing the cumulative distribution function of a Binomial distribution. Although no closed-form solution exists, this inequality can be estimated numerically.

# B   IMPACT OF EMBEDDING DISTRIBUTION AND SAMPLING STRATEGY ON UNLEARNING PERFORMANCE

We investigate the effect of different embedding distributions on class-wise unlearning by sampling embeddings from Gaussian, Laplace, and Uniform distributions. As reported in Table 5 and Table 6, the choice of embedding distribution does impact downstream unlearning performance. Nevertheless, all three distributions achieve competitive results, demonstrating near-complete forgetting alongside strong accuracy on the retain classes. These findings highlight the robustness of our framework to variations in the sampling strategy, as expected from the Proposition 1.

Table 5: Effect of embedding distribution on data-free class unlearning performance of some of methods on CIFAR-10, CIFAR-100, and TinyImageNet using ResNet-18 as the backbone architecture. Rows highlighted in gray represent our results using synthetic embeddings, while the corresponding non-shaded rows use original embeddings with the same method.

| Method | Embedding Distribution | $\mathcal{D}_r$ free | $\mathcal{D}_f$ free | CIFAR-10 $\mathcal{A}_r^t \uparrow$ | $\mathcal{A}_f^t \downarrow$ | AUS $\uparrow$ | CIFAR-100 $\mathcal{A}_r^t \uparrow$ | $\mathcal{A}_f^t \downarrow$ | AUS $\uparrow$ | TinyImageNet $\mathcal{A}_r^t \uparrow$ | $\mathcal{A}_f^t \downarrow$ | AUS $\uparrow$ |
|---|---|---|---|---|---|---|---|---|---|---|---|---|
| Original | – | – | – | $86.58_{\pm0.83}$ | $86.58_{\pm6.67}$ | $0.537_{\pm0.020}$ | $78.16_{\pm1.07}$ | $78.16_{\pm11.15}$ | $0.564_{\pm0.037}$ | $71.30_{\pm0.29}$ | $71.30_{\pm12.46}$ | $0.587_{\pm0.045}$ |
| Retrained | – | – | – | $86.95_{\pm1.22}$ | $0.0_{\pm0.0}$ | $1.000_{\pm0.005}$ | $77.92_{\pm0.80}$ | $0.0_{\pm0.0}$ | $0.956_{\pm0.036}$ | $63.01_{\pm2.77}$ | $0.0_{\pm0.0}$ | $0.855_{\pm0.029}$ |
| RL (Hayase et al., 2020) | Real distribution | ✓ | ✗ | $87.43_{\pm1.16}$ | $0.0_{\pm0.0}$ | $1.008_{\pm0.004}$ | $78.36_{\pm1.05}$ | $0.0_{\pm0.0}$ | $1.002_{\pm0.001}$ | $71.35_{\pm0.32}$ | $0.0_{\pm0.0}$ | $1.001_{\pm0.001}$ |
|  | Gaussian | ✓ | ✓ | $87.25_{\pm1.10}$ | $0.0_{\pm0.0}$ | $1.007_{\pm0.003}$ | $77.98_{\pm1.03}$ | $0.0_{\pm0.0}$ | $0.998_{\pm0.002}$ | $71.10_{\pm0.34}$ | $0.0_{\pm0.0}$ | $0.998_{\pm0.001}$ |
|  | Laplace | ✓ | ✓ | $87.25_{\pm1.09}$ | $0.0_{\pm0.0}$ | $1.007_{\pm0.003}$ | $78.00_{\pm1.04}$ | $0.0_{\pm0.0}$ | $0.998_{\pm0.002}$ | $71.18_{\pm0.34}$ | $0.0_{\pm0.0}$ | $0.999_{\pm0.001}$ |
|  | Uniform | ✓ | ✓ | $87.30_{\pm1.12}$ | $0.0_{\pm0.0}$ | $1.007_{\pm0.004}$ | $78.01_{\pm1.02}$ | $0.0_{\pm0.0}$ | $0.999_{\pm0.002}$ | $71.19_{\pm0.33}$ | $0.0_{\pm0.0}$ | $0.999_{\pm0.001}$ |
| DELETE (Zhou et al., 2025) | Real distribution | ✓ | ✗ | $87.33_{\pm1.12}$ | $0.0_{\pm0.0}$ | $1.008_{\pm0.004}$ | $78.28_{\pm1.06}$ | $0.0_{\pm0.0}$ | $1.001_{\pm0.001}$ | $71.43_{\pm0.30}$ | $0.0_{\pm0.0}$ | $1.001_{\pm0.000}$ |
|  | Gaussian | ✓ | ✓ | $87.35_{\pm1.13}$ | $0.0_{\pm0.0}$ | $1.008_{\pm0.004}$ | $78.25_{\pm1.07}$ | $0.0_{\pm0.1}$ | $1.001_{\pm0.000}$ | $71.36_{\pm0.30}$ | $0.0_{\pm0.0}$ | $1.001_{\pm0.000}$ |
|  | Laplace | ✓ | ✓ | $87.35_{\pm1.13}$ | $0.0_{\pm0.0}$ | $1.008_{\pm0.004}$ | $78.25_{\pm1.07}$ | $0.0_{\pm0.0}$ | $1.001_{\pm0.001}$ | $71.36_{\pm0.30}$ | $0.0_{\pm0.0}$ | $1.001_{\pm0.000}$ |
|  | Uniform | ✓ | ✓ | $87.33_{\pm1.13}$ | $0.0_{\pm0.0}$ | $1.008_{\pm0.004}$ | $78.25_{\pm1.07}$ | $0.0_{\pm0.0}$ | $1.001_{\pm0.001}$ | $71.35_{\pm0.30}$ | $0.3_{\pm1.2}$ | $0.998_{\pm0.011}$ |
| NG+ (Kurmanji et al., 2023) | Real distribution | ✗ | ✗ | $85.31_{\pm9.73}$ | $0.0_{\pm0.0}$ | $0.987_{\pm0.095}$ | $77.57_{\pm6.40}$ | $0.0_{\pm0.0}$ | $0.994_{\pm0.062}$ | $71.21_{\pm0.86}$ | $0.0_{\pm0.0}$ | $0.999_{\pm0.008}$ |
|  | Gaussian | ✓ | ✓ | $87.33_{\pm1.12}$ | $0.0_{\pm0.0}$ | $1.007_{\pm0.004}$ | $78.26_{\pm1.04}$ | $0.0_{\pm0.1}$ | $1.001_{\pm0.002}$ | $71.29_{\pm0.36}$ | $0.0_{\pm0.1}$ | $1.000_{\pm0.001}$ |
|  | Laplace | ✓ | ✓ | $87.35_{\pm1.13}$ | $0.0_{\pm0.0}$ | $1.008_{\pm0.004}$ | $78.31_{\pm0.99}$ | $0.0_{\pm0.0}$ | $1.001_{\pm0.001}$ | $71.06_{\pm0.46}$ | $0.0_{\pm0.2}$ | $0.997_{\pm0.004}$ |
|  | Uniform | ✓ | ✓ | $87.32_{\pm1.12}$ | $0.0_{\pm0.0}$ | $1.007_{\pm0.003}$ | $78.27_{\pm1.05}$ | $0.0_{\pm0.0}$ | $1.001_{\pm0.001}$ | $71.33_{\pm0.33}$ | $0.0_{\pm0.0}$ | $1.000_{\pm0.001}$ |
| SCRUB (Kurmanji et al., 2023) | Real distribution | ✗ | ✗ | $87.11_{\pm1.04}$ | $0.0_{\pm0.0}$ | $1.005_{\pm0.003}$ | $77.52_{\pm1.06}$ | $0.0_{\pm0.0}$ | $0.994_{\pm0.002}$ | $67.60_{\pm1.51}$ | $0.0_{\pm0.4}$ | $0.963_{\pm0.014}$ |
|  | Gaussian | ✓ | ✓ | $87.41_{\pm1.16}$ | $0.0_{\pm0.0}$ | $1.008_{\pm0.004}$ | $78.10_{\pm1.06}$ | $0.0_{\pm0.0}$ | $0.999_{\pm0.001}$ | $71.02_{\pm0.42}$ | $0.0_{\pm0.0}$ | $0.997_{\pm0.002}$ |
|  | Laplace | ✓ | ✓ | $87.41_{\pm1.15}$ | $0.0_{\pm0.0}$ | $1.008_{\pm0.004}$ | $78.19_{\pm1.00}$ | $0.0_{\pm0.0}$ | $1.000_{\pm0.001}$ | $71.11_{\pm0.37}$ | $0.0_{\pm0.0}$ | $0.998_{\pm0.001}$ |
|  | Uniform | ✓ | ✓ | $87.41_{\pm1.15}$ | $0.0_{\pm0.0}$ | $1.008_{\pm0.004}$ | $78.09_{\pm1.05}$ | $0.0_{\pm0.0}$ | $0.999_{\pm0.001}$ | $70.88_{\pm0.35}$ | $0.0_{\pm0.0}$ | $0.996_{\pm0.001}$ |

Table 6: Effect of embedding distribution on data-free class unlearning performance of some of methods on CIFAR-10, CIFAR-100, and TinyImageNet using ViT-B-16 as the backbone architecture. Rows highlighted in gray represent our results using synthetic embeddings, while the corresponding non-shaded rows use original embeddings with the same method.

| Method | Embedding Distribution | $\mathcal{D}_r$ free | $\mathcal{D}_f$ free | CIFAR-10 $\mathcal{A}_r^t \uparrow$ | $\mathcal{A}_f^t \downarrow$ | AUS $\uparrow$ | CIFAR-100 $\mathcal{A}_r^t \uparrow$ | $\mathcal{A}_f^t \downarrow$ | AUS $\uparrow$ | TinyImageNet $\mathcal{A}_r^t \uparrow$ | $\mathcal{A}_f^t \downarrow$ | AUS $\uparrow$ |
|---|---|---|---|---|---|---|---|---|---|---|---|---|
| Original | – | – | – | $97.69_{\pm0.18}$ | $97.69_{\pm1.30}$ | $0.506_{\pm0.003}$ | $87.22_{\pm0.26}$ | $87.22_{\pm7.83}$ | $0.535_{\pm0.023}$ | $88.20_{\pm0.14}$ | $88.20_{\pm7.29}$ | $0.532_{\pm0.022}$ |
| Retrained | – | – | – | $98.38_{\pm0.21}$ | $0.0_{\pm0.0}$ | $1.007_{\pm0.002}$ | $88.74_{\pm0.21}$ | $0.0_{\pm0.0}$ | $1.015_{\pm0.003}$ | $89.60_{\pm0.13}$ | $0.0_{\pm0.0}$ | $1.014_{\pm0.002}$ |
| RL (Hayase et al., 2020) | Real distribution | ✓ | ✗ | $97.91_{\pm0.25}$ | $0.0_{\pm0.0}$ | $1.002_{\pm0.001}$ | $87.31_{\pm0.28}$ | $0.0_{\pm0.0}$ | $1.001_{\pm0.001}$ | $88.24_{\pm0.14}$ | $0.0_{\pm0.0}$ | $1.000_{\pm0.000}$ |
|  | Gaussian | ✓ | ✓ | $97.92_{\pm0.25}$ | $0.0_{\pm0.0}$ | $1.002_{\pm0.001}$ | $87.30_{\pm0.29}$ | $0.0_{\pm0.0}$ | $1.001_{\pm0.001}$ | $88.23_{\pm0.14}$ | $0.0_{\pm0.0}$ | $1.000_{\pm0.001}$ |
|  | Laplace | ✓ | ✓ | $97.90_{\pm0.23}$ | $0.0_{\pm0.0}$ | $1.002_{\pm0.001}$ | $87.30_{\pm0.28}$ | $0.0_{\pm0.0}$ | $1.001_{\pm0.001}$ | $88.23_{\pm0.14}$ | $0.0_{\pm0.0}$ | $1.000_{\pm0.001}$ |
|  | Uniform | ✓ | ✓ | $97.92_{\pm0.24}$ | $0.0_{\pm0.0}$ | $1.002_{\pm0.001}$ | $87.29_{\pm0.28}$ | $0.0_{\pm0.0}$ | $1.001_{\pm0.001}$ | $88.17_{\pm0.14}$ | $0.0_{\pm0.0}$ | $1.000_{\pm0.001}$ |
| DELETE (Zhou et al., 2025) | Real distribution | ✓ | ✗ | $97.89_{\pm0.25}$ | $0.0_{\pm0.0}$ | $1.002_{\pm0.001}$ | $87.30_{\pm0.27}$ | $0.0_{\pm0.0}$ | $1.001_{\pm0.001}$ | $88.23_{\pm0.14}$ | $0.0_{\pm0.0}$ | $1.000_{\pm0.000}$ |
|  | Gaussian | ✓ | ✓ | $97.90_{\pm0.25}$ | $0.0_{\pm0.0}$ | $1.002_{\pm0.001}$ | $87.30_{\pm0.27}$ | $0.0_{\pm0.0}$ | $1.001_{\pm0.001}$ | $88.23_{\pm0.14}$ | $2.7_{\pm8.2}$ | $0.979_{\pm0.060}$ |
|  | Laplace | ✓ | ✓ | $97.90_{\pm0.25}$ | $0.0_{\pm0.0}$ | $1.002_{\pm0.001}$ | $87.24_{\pm0.26}$ | $0.0_{\pm0.0}$ | $1.001_{\pm0.001}$ | $88.24_{\pm0.14}$ | $0.0_{\pm0.0}$ | $1.000_{\pm0.000}$ |
|  | Uniform | ✓ | ✓ | $97.89_{\pm0.25}$ | $0.0_{\pm0.0}$ | $1.002_{\pm0.001}$ | $87.30_{\pm0.27}$ | $0.0_{\pm0.0}$ | $1.001_{\pm0.001}$ | $88.24_{\pm0.14}$ | $0.0_{\pm0.0}$ | $1.000_{\pm0.000}$ |
| NG+ (Kurmanji et al., 2023) | Real distribution | ✗ | ✗ | $97.88_{\pm0.25}$ | $0.0_{\pm0.0}$ | $1.002_{\pm0.001}$ | $87.15_{\pm0.29}$ | $0.0_{\pm0.2}$ | $0.999_{\pm0.003}$ | $87.64_{\pm0.27}$ | $0.1_{\pm0.4}$ | $0.993_{\pm0.005}$ |
|  | Gaussian | ✓ | ✓ | $97.91_{\pm0.25}$ | $0.0_{\pm0.0}$ | $1.002_{\pm0.001}$ | $87.30_{\pm0.31}$ | $0.0_{\pm0.0}$ | $1.001_{\pm0.001}$ | $88.25_{\pm0.15}$ | $0.0_{\pm0.0}$ | $1.001_{\pm0.000}$ |
|  | Laplace | ✓ | ✓ | $97.91_{\pm0.25}$ | $0.0_{\pm0.0}$ | $1.002_{\pm0.001}$ | $87.29_{\pm0.31}$ | $0.0_{\pm0.0}$ | $1.001_{\pm0.001}$ | $88.24_{\pm0.15}$ | $0.0_{\pm0.0}$ | $1.001_{\pm0.000}$ |
|  | Uniform | ✓ | ✓ | $97.90_{\pm0.25}$ | $0.0_{\pm0.0}$ | $1.002_{\pm0.001}$ | $87.30_{\pm0.30}$ | $0.0_{\pm0.0}$ | $1.001_{\pm0.001}$ | $88.26_{\pm0.15}$ | $0.0_{\pm0.0}$ | $1.001_{\pm0.000}$ |

## C IMPACT OF THE NUMBER OF SYNTHETIC EMBEDDINGS PER CLASS ON UNLEARNING PERFORMANCE

This part extends the ablation in Section 4 (see Figure 2) by considering additional backbones and datasets such as ResNet-18 on CIFAR-10 (Figure 3), ResNet-18 on TinyImageNet (Figure 4), ViT-B-16 on CIFAR-10 (Figure 5), and ViT-B-16 on CIFAR-100 (Figure 6). For each setting, we vary the number of synthetic embeddings per class and measure retain accuracy $\mathcal{A}_r^t$, forget accuracy $\mathcal{A}_f^t$, and AUS. Across all configurations, the trend is consistent. The pattern is consistent across configurations: increasing the number of synthetic embeddings raises $\mathcal{A}_r^t$ and AUS while reducing $\mathcal{A}_f^t$.

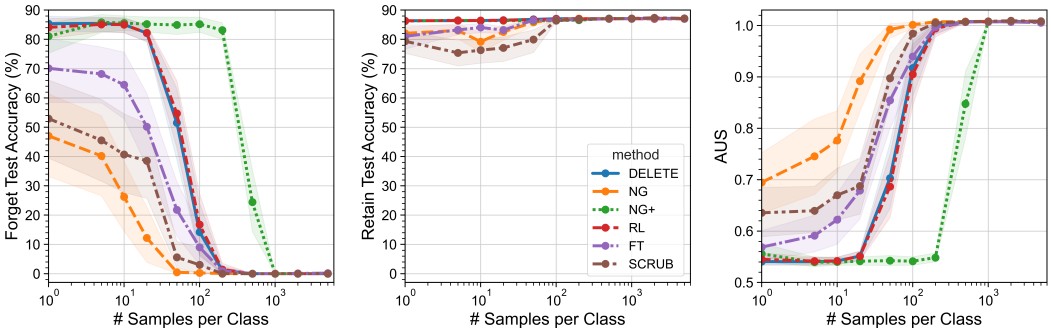

Figure 3: Effect of the number of synthetic embeddings per class on unlearning performance. Results are averaged over three independently trained models, with class-wise unlearning performed separately for each class. Error bars indicate $95\%$ confidence intervals. Experiments use the ResNet-18 architecture on the CIFAR-10 dataset.

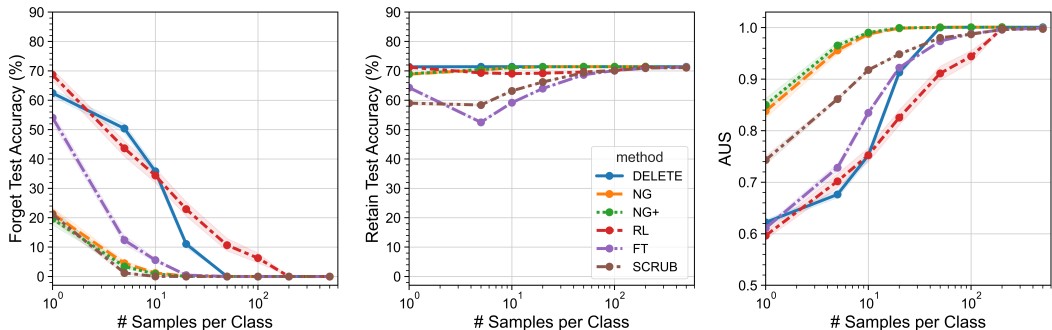

Figure 4: Effect of the number of synthetic embeddings per class on unlearning performance. Results are averaged over three independently trained models, with class-wise unlearning performed separately for each class. Error bars indicate $95\%$ confidence intervals. Experiments use the ResNet-18 architecture on the TinyImageNet dataset.

## D CODE

Our code is available at this repository.[1]

---

[1] https://anonymous.4open.science/r/MU_source_free_iclr.

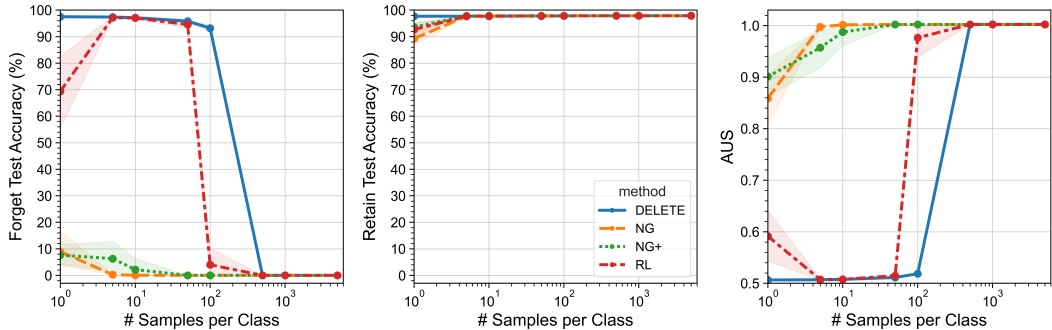

Figure 5: Effect of the number of synthetic embeddings per class on unlearning performance. Results are averaged over three independently trained models, with class-wise unlearning performed separately for each class. Error bars indicate 95% confidence intervals. Experiments use the ViT-B-16 architecture on the CIFAR-10 dataset.

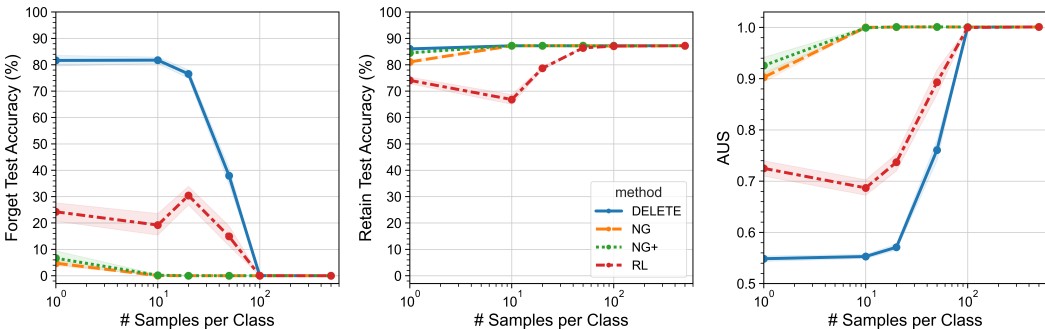

Figure 6: Effect of the number of synthetic embeddings per class on unlearning performance. Results are averaged over three independently trained models, with class-wise unlearning performed separately for each class. Error bars indicate 95% confidence intervals. Experiments use the ViT-B-16 architecture on the CIFAR-100 dataset.

# E    PER-CLASS UNLEARNING RESULTS ON CIFAR-10

To supplement the average unlearning performance presented in Table 1 and 2, we provide a detailed per-class evaluation in Table 7 for ResNet-18, Table 8 for ResNet-50, Table 9 for ViT-B-16 and Table 10 for Swin-T. These tables present class-wise unlearning metrics on CIFAR-10 using ResNet-18, ResNet-50, ViT-B-16, and Swin-T backbones, respectively. The results illustrate variability in both unlearning effectiveness and the retain accuracy across target classes, highlighting the impact of semantic complexity and class-specific challenges.

Table 7: Class unlearning performance for CIFAR-10 using ResNet-18, averaged over 5 random trials. Rows highlighted in gray represent our results using synthetic embeddings, while the corresponding non-shaded rows use original embeddings with the same method.

| Method | Metric | Forget Class 0 | 1 | 2 | 3 | 4 | 5 | 6 | 7 | 8 | 9 |
|---|---|---|---|---|---|---|---|---|---|---|---|
| Original | $\mathcal{A}_r^t \uparrow$ | $86.22_{\pm0.54}$ | $85.91_{\pm0.40}$ | $86.91_{\pm0.47}$ | $88.30_{\pm0.29}$ | $86.50_{\pm0.50}$ | $87.43_{\pm0.42}$ | $86.05_{\pm0.43}$ | $86.29_{\pm0.46}$ | $86.01_{\pm0.38}$ | $86.16_{\pm0.33}$ |
|  | $\mathcal{A}_f^t \downarrow$ | $89.8_{\pm1.1}$ | $92.6_{\pm0.7}$ | $83.6_{\pm0.8}$ | $71.0_{\pm2.0}$ | $87.3_{\pm0.9}$ | $78.9_{\pm0.8}$ | $91.4_{\pm1.0}$ | $89.2_{\pm0.7}$ | $91.7_{\pm0.8}$ | $90.3_{\pm1.4}$ |
|  | AUS $\uparrow$ | $0.527_{\pm0.003}$ | $0.519_{\pm0.002}$ | $0.545_{\pm0.002}$ | $0.585_{\pm0.007}$ | $0.534_{\pm0.002}$ | $0.559_{\pm0.002}$ | $0.523_{\pm0.003}$ | $0.529_{\pm0.002}$ | $0.522_{\pm0.002}$ | $0.525_{\pm0.004}$ |
| Retrained | $\mathcal{A}_r^t \uparrow$ | 86.43 | 86.29 | 87.38 | 89.53 | 86.79 | 88.66 | 86.16 | 86.24 | 85.92 | 86.14 |
|  | $\mathcal{A}_f^t \downarrow$ | 0.0 | 0.0 | 0.0 | 0.0 | 0.0 | 0.0 | 0.0 | 0.0 | 0.0 | 0.0 |
|  | AUS $\uparrow$ | 1.002 | 1.004 | 1.005 | 1.012 | 1.003 | 1.012 | 1.001 | 1.000 | 0.999 | 1.000 |
| FT (Golatkar et al., 2020) | $\mathcal{A}_r^t \uparrow$ | $87.01_{\pm0.26}$ | $86.58_{\pm0.13}$ | $87.82_{\pm0.17}$ | $89.64_{\pm0.22}$ | $87.38_{\pm0.29}$ | $88.83_{\pm0.27}$ | $86.77_{\pm0.10}$ | $86.85_{\pm0.27}$ | $86.53_{\pm0.25}$ | $86.91_{\pm0.28}$ |
|  | $\mathcal{A}_f^t \downarrow$ | $0.0_{\pm0.0}$ | $0.0_{\pm0.0}$ | $0.0_{\pm0.0}$ | $0.0_{\pm0.0}$ | $0.0_{\pm0.0}$ | $0.0_{\pm0.0}$ | $0.0_{\pm0.0}$ | $0.0_{\pm0.0}$ | $0.0_{\pm0.0}$ | $0.0_{\pm0.0}$ |
|  | AUS $\uparrow$ | $1.008_{\pm0.003}$ | $1.007_{\pm0.003}$ | $1.009_{\pm0.004}$ | $1.013_{\pm0.002}$ | $1.009_{\pm0.003}$ | $1.014_{\pm0.005}$ | $1.007_{\pm0.003}$ | $1.006_{\pm0.003}$ | $1.005_{\pm0.001}$ | $1.007_{\pm0.001}$ |
|  | $\mathcal{A}_r^t \uparrow$ | $86.92_{\pm0.43}$ | $86.50_{\pm0.41}$ | $87.79_{\pm0.31}$ | $89.74_{\pm0.30}$ | $87.26_{\pm0.48}$ | $88.78_{\pm0.31}$ | $86.66_{\pm0.35}$ | $86.77_{\pm0.48}$ | $86.40_{\pm0.43}$ | $86.88_{\pm0.44}$ |
|  | $\mathcal{A}_f^t \downarrow$ | $0.0_{\pm0.0}$ | $0.0_{\pm0.0}$ | $0.0_{\pm0.0}$ | $0.0_{\pm0.0}$ | $0.0_{\pm0.0}$ | $0.0_{\pm0.0}$ | $0.0_{\pm0.0}$ | $0.0_{\pm0.0}$ | $0.0_{\pm0.0}$ | $0.0_{\pm0.0}$ |
|  | AUS $\uparrow$ | $1.007_{\pm0.002}$ | $1.006_{\pm0.001}$ | $1.009_{\pm0.002}$ | $1.014_{\pm0.001}$ | $1.008_{\pm0.000}$ | $1.013_{\pm0.002}$ | $1.006_{\pm0.001}$ | $1.005_{\pm0.001}$ | $1.004_{\pm0.001}$ | $1.007_{\pm0.001}$ |
| NG (Golatkar et al., 2020) | $\mathcal{A}_r^t \uparrow$ | $86.89_{\pm0.54}$ | $86.46_{\pm0.36}$ | $87.71_{\pm0.41}$ | $89.71_{\pm0.34}$ | $87.20_{\pm0.54}$ | $88.68_{\pm0.43}$ | $86.59_{\pm0.41}$ | $86.71_{\pm0.54}$ | $86.37_{\pm0.50}$ | $86.76_{\pm0.40}$ |
|  | $\mathcal{A}_f^t \downarrow$ | $0.0_{\pm0.0}$ | $0.0_{\pm0.0}$ | $0.0_{\pm0.0}$ | $0.0_{\pm0.0}$ | $0.0_{\pm0.0}$ | $0.0_{\pm0.0}$ | $0.0_{\pm0.0}$ | $0.0_{\pm0.0}$ | $0.0_{\pm0.0}$ | $0.0_{\pm0.0}$ |
|  | AUS $\uparrow$ | $1.007_{\pm0.001}$ | $1.005_{\pm0.001}$ | $1.008_{\pm0.001}$ | $1.014_{\pm0.001}$ | $1.007_{\pm0.001}$ | $1.012_{\pm0.001}$ | $1.005_{\pm0.001}$ | $1.004_{\pm0.002}$ | $1.004_{\pm0.001}$ | $1.006_{\pm0.001}$ |
|  | $\mathcal{A}_r^t \uparrow$ | $86.98_{\pm0.46}$ | $86.47_{\pm0.41}$ | $87.79_{\pm0.35}$ | $89.82_{\pm0.42}$ | $87.27_{\pm0.52}$ | $88.89_{\pm0.31}$ | $86.69_{\pm0.33}$ | $86.77_{\pm0.46}$ | $86.46_{\pm0.44}$ | $86.86_{\pm0.41}$ |
|  | $\mathcal{A}_f^t \downarrow$ | $0.0_{\pm0.0}$ | $0.0_{\pm0.0}$ | $0.0_{\pm0.0}$ | $0.0_{\pm0.0}$ | $0.0_{\pm0.0}$ | $0.0_{\pm0.0}$ | $0.0_{\pm0.0}$ | $0.0_{\pm0.0}$ | $0.0_{\pm0.0}$ | $0.0_{\pm0.0}$ |
|  | AUS $\uparrow$ | $1.008_{\pm0.001}$ | $1.006_{\pm0.000}$ | $1.009_{\pm0.002}$ | $1.015_{\pm0.001}$ | $1.008_{\pm0.001}$ | $1.015_{\pm0.001}$ | $1.006_{\pm0.001}$ | $1.005_{\pm0.001}$ | $1.004_{\pm0.001}$ | $1.007_{\pm0.001}$ |
| RL (Hayase et al., 2020) | $\mathcal{A}_r^t \uparrow$ | $86.99_{\pm0.51}$ | $86.48_{\pm0.41}$ | $87.83_{\pm0.35}$ | $89.83_{\pm0.45}$ | $87.35_{\pm0.42}$ | $88.99_{\pm0.44}$ | $86.73_{\pm0.33}$ | $86.81_{\pm0.47}$ | $86.43_{\pm0.44}$ | $86.82_{\pm0.44}$ |
|  | $\mathcal{A}_f^t \downarrow$ | $0.0_{\pm0.0}$ | $0.0_{\pm0.0}$ | $0.0_{\pm0.0}$ | $0.0_{\pm0.0}$ | $0.0_{\pm0.0}$ | $0.0_{\pm0.0}$ | $0.0_{\pm0.0}$ | $0.0_{\pm0.0}$ | $0.0_{\pm0.0}$ | $0.0_{\pm0.0}$ |
|  | AUS $\uparrow$ | $1.008_{\pm0.001}$ | $1.006_{\pm0.000}$ | $1.009_{\pm0.001}$ | $1.015_{\pm0.002}$ | $1.008_{\pm0.001}$ | $1.016_{\pm0.002}$ | $1.007_{\pm0.001}$ | $1.005_{\pm0.001}$ | $1.004_{\pm0.001}$ | $1.007_{\pm0.001}$ |
|  | $\mathcal{A}_r^t \uparrow$ | $86.93_{\pm0.46}$ | $86.38_{\pm0.38}$ | $87.77_{\pm0.28}$ | $89.65_{\pm0.33}$ | $87.22_{\pm0.48}$ | $88.82_{\pm0.34}$ | $86.64_{\pm0.37}$ | $86.78_{\pm0.45}$ | $86.41_{\pm0.41}$ | $86.72_{\pm0.36}$ |
|  | $\mathcal{A}_f^t \downarrow$ | $0.0_{\pm0.0}$ | $0.0_{\pm0.0}$ | $0.0_{\pm0.0}$ | $0.0_{\pm0.0}$ | $0.0_{\pm0.0}$ | $0.0_{\pm0.0}$ | $0.0_{\pm0.0}$ | $0.0_{\pm0.0}$ | $0.0_{\pm0.0}$ | $0.0_{\pm0.0}$ |
|  | AUS $\uparrow$ | $1.007_{\pm0.002}$ | $1.005_{\pm0.000}$ | $1.009_{\pm0.002}$ | $1.013_{\pm0.001}$ | $1.007_{\pm0.001}$ | $1.014_{\pm0.002}$ | $1.006_{\pm0.001}$ | $1.005_{\pm0.001}$ | $1.004_{\pm0.000}$ | $1.006_{\pm0.001}$ |
| BS (Chen et al., 2023) | $\mathcal{A}_r^t \uparrow$ | $85.31_{\pm1.26}$ | $85.83_{\pm0.58}$ | $86.87_{\pm0.57}$ | $88.18_{\pm0.65}$ | $85.98_{\pm0.33}$ | $87.44_{\pm0.94}$ | $85.74_{\pm0.89}$ | $86.08_{\pm0.55}$ | $85.45_{\pm0.52}$ | $86.06_{\pm0.34}$ |
|  | $\mathcal{A}_f^t \downarrow$ | $0.5_{\pm0.8}$ | $0.2_{\pm0.2}$ | $0.1_{\pm0.1}$ | $0.0_{\pm0.0}$ | $0.5_{\pm1.0}$ | $0.0_{\pm0.0}$ | $0.0_{\pm0.0}$ | $0.1_{\pm0.2}$ | $0.1_{\pm0.2}$ | $0.0_{\pm0.0}$ |
|  | AUS $\uparrow$ | $0.986_{\pm0.017}$ | $0.997_{\pm0.005}$ | $0.998_{\pm0.005}$ | $0.999_{\pm0.005}$ | $0.990_{\pm0.013}$ | $1.000_{\pm0.007}$ | $0.997_{\pm0.006}$ | $0.997_{\pm0.002}$ | $0.993_{\pm0.006}$ | $0.999_{\pm0.001}$ |
|  | $\mathcal{A}_r^t \uparrow$ | $86.83_{\pm0.50}$ | $86.46_{\pm0.37}$ | $87.70_{\pm0.37}$ | $89.81_{\pm0.34}$ | $87.26_{\pm0.50}$ | $89.01_{\pm0.32}$ | $86.62_{\pm0.37}$ | $86.77_{\pm0.48}$ | $86.45_{\pm0.41}$ | $86.81_{\pm0.36}$ |
|  | $\mathcal{A}_f^t \downarrow$ | $0.0_{\pm0.0}$ | $0.0_{\pm0.0}$ | $0.0_{\pm0.0}$ | $0.0_{\pm0.0}$ | $0.0_{\pm0.0}$ | $0.0_{\pm0.0}$ | $0.0_{\pm0.0}$ | $0.0_{\pm0.0}$ | $0.0_{\pm0.0}$ | $0.0_{\pm0.0}$ |
|  | AUS $\uparrow$ | $1.006_{\pm0.001}$ | $1.005_{\pm0.000}$ | $1.008_{\pm0.002}$ | $1.015_{\pm0.001}$ | $1.008_{\pm0.001}$ | $1.016_{\pm0.001}$ | $1.006_{\pm0.002}$ | $1.005_{\pm0.001}$ | $1.004_{\pm0.001}$ | $1.007_{\pm0.001}$ |
| BE (Chen et al., 2023) | $\mathcal{A}_r^t \uparrow$ | $82.40_{\pm3.28}$ | $84.66_{\pm1.05}$ | $85.63_{\pm0.18}$ | $85.60_{\pm0.64}$ | $85.32_{\pm0.77}$ | $84.51_{\pm1.89}$ | $84.56_{\pm0.64}$ | $85.49_{\pm0.57}$ | $83.78_{\pm1.71}$ | $85.23_{\pm0.61}$ |
|  | $\mathcal{A}_f^t \downarrow$ | $1.4_{\pm1.7}$ | $0.0_{\pm0.0}$ | $0.1_{\pm0.2}$ | $1.0_{\pm2.2}$ | $0.5_{\pm1.1}$ | $0.5_{\pm1.0}$ | $0.6_{\pm0.9}$ | $0.0_{\pm0.0}$ | $0.9_{\pm1.7}$ | $0.0_{\pm0.1}$ |
|  | AUS $\uparrow$ | $0.949_{\pm0.040}$ | $0.987_{\pm0.008}$ | $0.986_{\pm0.004}$ | $0.964_{\pm0.020}$ | $0.983_{\pm0.010}$ | $0.966_{\pm0.016}$ | $0.980_{\pm0.005}$ | $0.992_{\pm0.002}$ | $0.969_{\pm0.029}$ | $0.990_{\pm0.004}$ |
|  | $\mathcal{A}_r^t \uparrow$ | $86.01_{\pm0.60}$ | $85.92_{\pm0.38}$ | $86.82_{\pm0.47}$ | $88.15_{\pm0.39}$ | $86.50_{\pm0.45}$ | $87.33_{\pm0.54}$ | $86.04_{\pm0.43}$ | $86.25_{\pm0.48}$ | $85.95_{\pm0.30}$ | $86.13_{\pm0.27}$ |
|  | $\mathcal{A}_f^t \downarrow$ | $0.0_{\pm0.0}$ | $0.0_{\pm0.0}$ | $0.0_{\pm0.0}$ | $0.0_{\pm0.0}$ | $0.0_{\pm0.0}$ | $0.0_{\pm0.0}$ | $0.0_{\pm0.0}$ | $0.0_{\pm0.0}$ | $0.0_{\pm0.0}$ | $0.0_{\pm0.0}$ |
|  | AUS $\uparrow$ | $0.998_{\pm0.002}$ | $1.000_{\pm0.000}$ | $0.999_{\pm0.001}$ | $0.999_{\pm0.001}$ | $1.000_{\pm0.001}$ | $0.999_{\pm0.002}$ | $1.000_{\pm0.000}$ | $1.000_{\pm0.000}$ | $0.999_{\pm0.001}$ | $1.000_{\pm0.001}$ |
| DELETE (Zhou et al., 2025) | $\mathcal{A}_r^t \uparrow$ | $86.93_{\pm0.44}$ | $86.42_{\pm0.38}$ | $87.74_{\pm0.32}$ | $89.71_{\pm0.38}$ | $87.22_{\pm0.48}$ | $88.80_{\pm0.34}$ | $86.60_{\pm0.32}$ | $86.74_{\pm0.46}$ | $86.41_{\pm0.42}$ | $86.75_{\pm0.39}$ |
|  | $\mathcal{A}_f^t \downarrow$ | $0.0_{\pm0.0}$ | $0.0_{\pm0.0}$ | $0.0_{\pm0.0}$ | $0.0_{\pm0.0}$ | $0.0_{\pm0.0}$ | $0.0_{\pm0.0}$ | $0.0_{\pm0.0}$ | $0.0_{\pm0.0}$ | $0.0_{\pm0.0}$ | $0.0_{\pm0.0}$ |
|  | AUS $\uparrow$ | $1.007_{\pm0.001}$ | $1.005_{\pm0.000}$ | $1.008_{\pm0.002}$ | $1.014_{\pm0.001}$ | $1.007_{\pm0.001}$ | $1.014_{\pm0.001}$ | $1.006_{\pm0.001}$ | $1.004_{\pm0.001}$ | $1.004_{\pm0.001}$ | $1.006_{\pm0.001}$ |
|  | $\mathcal{A}_r^t \uparrow$ | $86.95_{\pm0.46}$ | $86.44_{\pm0.39}$ | $87.76_{\pm0.34}$ | $89.75_{\pm0.40}$ | $87.24_{\pm0.52}$ | $88.83_{\pm0.39}$ | $86.63_{\pm0.34}$ | $86.76_{\pm0.48}$ | $86.43_{\pm0.44}$ | $86.79_{\pm0.40}$ |
|  | $\mathcal{A}_f^t \downarrow$ | $0.0_{\pm0.0}$ | $0.0_{\pm0.0}$ | $0.0_{\pm0.0}$ | $0.0_{\pm0.0}$ | $0.0_{\pm0.0}$ | $0.0_{\pm0.0}$ | $0.0_{\pm0.0}$ | $0.0_{\pm0.0}$ | $0.0_{\pm0.0}$ | $0.0_{\pm0.0}$ |
|  | AUS $\uparrow$ | $1.007_{\pm0.001}$ | $1.005_{\pm0.000}$ | $1.009_{\pm0.001}$ | $1.014_{\pm0.001}$ | $1.007_{\pm0.001}$ | $1.014_{\pm0.001}$ | $1.006_{\pm0.001}$ | $1.005_{\pm0.001}$ | $1.004_{\pm0.001}$ | $1.006_{\pm0.001}$ |
| NG+ (Kurmanji et al., 2023) | $\mathcal{A}_r^t \uparrow$ | $86.31_{\pm1.26}$ | $86.18_{\pm0.52}$ | $87.41_{\pm0.38}$ | $89.23_{\pm0.30}$ | $86.99_{\pm0.50}$ | $88.08_{\pm0.33}$ | $85.58_{\pm1.39}$ | $83.70_{\pm6.71}$ | $73.08_{\pm29.61}$ | $86.60_{\pm0.39}$ |
|  | $\mathcal{A}_f^t \downarrow$ | $0.0_{\pm0.0}$ | $0.0_{\pm0.0}$ | $0.0_{\pm0.0}$ | $0.0_{\pm0.0}$ | $0.0_{\pm0.0}$ | $0.0_{\pm0.0}$ | $0.0_{\pm0.0}$ | $0.0_{\pm0.0}$ | $0.0_{\pm0.0}$ | $0.0_{\pm0.0}$ |
|  | AUS $\uparrow$ | $1.001_{\pm0.008}$ | $1.003_{\pm0.001}$ | $1.005_{\pm0.002}$ | $1.009_{\pm0.003}$ | $1.005_{\pm0.001}$ | $1.006_{\pm0.002}$ | $0.995_{\pm0.011}$ | $0.974_{\pm0.064}$ | $0.871_{\pm0.293}$ | $1.004_{\pm0.002}$ |
|  | $\mathcal{A}_r^t \uparrow$ | $86.95_{\pm0.49}$ | $86.45_{\pm0.41}$ | $87.82_{\pm0.34}$ | $89.79_{\pm0.42}$ | $87.27_{\pm0.54}$ | $88.82_{\pm0.32}$ | $86.63_{\pm0.33}$ | $86.77_{\pm0.46}$ | $86.46_{\pm0.48}$ | $86.79_{\pm0.44}$ |
|  | $\mathcal{A}_f^t \downarrow$ | $0.0_{\pm0.0}$ | $0.0_{\pm0.0}$ | $0.0_{\pm0.0}$ | $0.0_{\pm0.0}$ | $0.0_{\pm0.0}$ | $0.0_{\pm0.0}$ | $0.0_{\pm0.0}$ | $0.0_{\pm0.0}$ | $0.0_{\pm0.0}$ | $0.0_{\pm0.0}$ |
|  | AUS $\uparrow$ | $1.007_{\pm0.001}$ | $1.005_{\pm0.000}$ | $1.009_{\pm0.001}$ | $1.015_{\pm0.001}$ | $1.008_{\pm0.001}$ | $1.014_{\pm0.001}$ | $1.006_{\pm0.001}$ | $1.005_{\pm0.001}$ | $1.005_{\pm0.001}$ | $1.006_{\pm0.001}$ |
| SCRUB (Kurmanji et al., 2023) | $\mathcal{A}_r^t \uparrow$ | $86.48_{\pm0.74}$ | $86.33_{\pm0.44}$ | $87.53_{\pm0.28}$ | $89.32_{\pm0.32}$ | $86.96_{\pm0.42}$ | $88.41_{\pm0.22}$ | $86.44_{\pm0.23}$ | $86.69_{\pm0.38}$ | $86.35_{\pm0.37}$ | $86.61_{\pm0.51}$ |
|  | $\mathcal{A}_f^t \downarrow$ | $0.0_{\pm0.0}$ | $0.0_{\pm0.0}$ | $0.0_{\pm0.0}$ | $0.0_{\pm0.0}$ | $0.0_{\pm0.0}$ | $0.0_{\pm0.0}$ | $0.0_{\pm0.0}$ | $0.0_{\pm0.0}$ | $0.0_{\pm0.0}$ | $0.0_{\pm0.0}$ |
|  | AUS $\uparrow$ | $1.003_{\pm0.002}$ | $1.004_{\pm0.001}$ | $1.006_{\pm0.002}$ | $1.010_{\pm0.002}$ | $1.005_{\pm0.004}$ | $1.010_{\pm0.003}$ | $1.004_{\pm0.002}$ | $1.004_{\pm0.002}$ | $1.003_{\pm0.001}$ | $1.004_{\pm0.003}$ |
|  | $\mathcal{A}_r^t \uparrow$ | $87.01_{\pm0.46}$ | $86.54_{\pm0.39}$ | $87.82_{\pm0.30}$ | $89.97_{\pm0.40}$ | $87.28_{\pm0.53}$ | $88.96_{\pm0.35}$ | $86.69_{\pm0.32}$ | $86.82_{\pm0.46}$ | $86.50_{\pm0.45}$ | $86.89_{\pm0.38}$ |
|  | $\mathcal{A}_f^t \downarrow$ | $0.0_{\pm0.0}$ | $0.0_{\pm0.0}$ | $0.0_{\pm0.0}$ | $0.0_{\pm0.0}$ | $0.0_{\pm0.0}$ | $0.0_{\pm0.0}$ | $0.0_{\pm0.0}$ | $0.0_{\pm0.0}$ | $0.0_{\pm0.0}$ | $0.0_{\pm0.0}$ |
|  | AUS $\uparrow$ | $1.008_{\pm0.001}$ | $1.006_{\pm0.000}$ | $1.009_{\pm0.002}$ | $1.017_{\pm0.001}$ | $1.008_{\pm0.001}$ | $1.015_{\pm0.001}$ | $1.006_{\pm0.001}$ | $1.005_{\pm0.001}$ | $1.005_{\pm0.001}$ | $1.007_{\pm0.001}$ |
| SCAR (Bonato et al., 2024) | $\mathcal{A}_r^t \uparrow$ | $87.03_{\pm0.47}$ | $86.50_{\pm0.37}$ | $87.85_{\pm0.32}$ | $89.87_{\pm0.39}$ | $87.31_{\pm0.52}$ | $88.96_{\pm0.41}$ | $86.73_{\pm0.35}$ | $86.81_{\pm0.46}$ | $86.49_{\pm0.43}$ | $86.88_{\pm0.43}$ |
|  | $\mathcal{A}_f^t \downarrow$ | $0.0_{\pm0.0}$ | $0.0_{\pm0.0}$ | $0.0_{\pm0.0}$ | $0.0_{\pm0.0}$ | $0.0_{\pm0.0}$ | $0.0_{\pm0.0}$ | $0.0_{\pm0.0}$ | $0.0_{\pm0.0}$ | $0.0_{\pm0.0}$ | $0.0_{\pm0.0}$ |
|  | AUS $\uparrow$ | $1.008_{\pm0.001}$ | $1.006_{\pm0.000}$ | $1.009_{\pm0.002}$ | $1.016_{\pm0.001}$ | $1.008_{\pm0.001}$ | $1.015_{\pm0.002}$ | $1.007_{\pm0.001}$ | $1.005_{\pm0.001}$ | $1.005_{\pm0.001}$ | $1.007_{\pm0.001}$ |
|  | $\mathcal{A}_r^t \uparrow$ | $86.97_{\pm0.45}$ | $86.46_{\pm0.37}$ | $87.80_{\pm0.31}$ | $89.77_{\pm0.37}$ | $87.27_{\pm0.49}$ | $88.85_{\pm0.34}$ | $86.66_{\pm0.30}$ | $86.78_{\pm0.46}$ | $86.46_{\pm0.42}$ | $86.80_{\pm0.39}$ |
|  | $\mathcal{A}_f^t \downarrow$ | $0.0_{\pm0.0}$ | $0.0_{\pm0.0}$ | $0.0_{\pm0.0}$ | $0.0_{\pm0.0}$ | $0.0_{\pm0.0}$ | $0.0_{\pm0.0}$ | $0.0_{\pm0.0}$ | $0.0_{\pm0.0}$ | $0.0_{\pm0.0}$ | $0.0_{\pm0.0}$ |
|  | AUS $\uparrow$ | $1.007_{\pm0.001}$ | $1.005_{\pm0.001}$ | $1.009_{\pm0.002}$ | $1.015_{\pm0.001}$ | $1.008_{\pm0.001}$ | $1.014_{\pm0.001}$ | $1.006_{\pm0.001}$ | $1.005_{\pm0.001}$ | $1.004_{\pm0.001}$ | $1.006_{\pm0.001}$ |

# F  CLASS UNLEARNING IN MULTI-CLASS SETTING.

Beyond the single-class setting, we evaluate whether our source-free class-unlearning framework scale to multi-class setting on CIFAR-100 using a ResNet-18 backbone (Table 11). We consider unlearning 2, 5, and 10 classes, with label sets $\mathcal{Y}_f = \{25, 58\}$, $\mathcal{Y}_f = \{25, 58, 38, 23, 96\}$, and $\mathcal{Y}_f = \{25, 58, 38, 23, 96, 54, 51, 49, 98, 66\}$, respectively, following the CIFAR-100 setup in (Zhou et al., 2025). In our multi-class experiments, all classes in $\mathcal{Y}_f$ are forgotten simultaneously in a single unlearning run. Each experiment is repeated across five random seeds.

Table 8: Class unlearning performance for CIFAR-10 using ResNet-50, averaged over 5 random trials. Rows highlighted in gray represent our results using synthetic embeddings, while the corresponding non-shaded rows use original embeddings with the same method.

| Method | Metric | Forget Class 0 | 1 | 2 | 3 | 4 | 5 | 6 | 7 | 8 | 9 |
|---|---|---|---|---|---|---|---|---|---|---|---|
| Original | $\mathcal{A}_r^t \uparrow$ | 88.18±0.55 | 87.84±0.51 | 88.57±0.62 | 89.58±0.50 | 88.10±0.69 | 89.26±0.73 | 87.77±0.57 | 87.98±0.71 | 87.73±0.83 | 87.74±0.63 |
| | $\mathcal{A}_f^t \downarrow$ | 89.1±3.1 | 92.2±2.2 | 85.6±1.2 | 76.5±2.4 | 89.9±0.6 | 79.4±0.9 | 92.8±1.3 | 90.9±0.8 | 93.2±2.4 | 93.1±0.8 |
| | AUS ↑ | 0.529±0.009 | 0.520±0.006 | 0.539±0.004 | 0.567±0.008 | 0.527±0.002 | 0.557±0.003 | 0.519±0.003 | 0.524±0.002 | 0.518±0.006 | 0.518±0.002 |
| Retrained | $\mathcal{A}_r^t \uparrow$ | 88.79 | 88.42 | 89.40 | 91.09 | 89.04 | 90.66 | 87.92 | 88.82 | 87.92 | 88.27 |
| | $\mathcal{A}_f^t \downarrow$ | 0.0 | 0.0 | 0.0 | 0.0 | 0.0 | 0.0 | 0.0 | 0.0 | 0.0 | 0.0 |
| | AUS ↑ | 1.006 | 1.006 | 1.008 | 1.015 | 1.009 | 1.014 | 1.002 | 1.008 | 1.002 | 1.005 |
| FT (Golatkar et al., 2020) | $\mathcal{A}_r^t \uparrow$ | 89.17±0.37 | 88.62±0.35 | 89.73±0.27 | 91.46±0.49 | 89.39±0.27 | 90.64±0.37 | 88.68±0.42 | 88.95±0.32 | 88.55±0.50 | 88.83±0.34 |
| | $\mathcal{A}_f^t \downarrow$ | 0.0±0.0 | 0.0±0.0 | 0.0±0.0 | 0.0±0.0 | 0.0±0.0 | 0.0±0.0 | 0.0±0.0 | 0.0±0.0 | 0.0±0.0 | 0.0±0.0 |
| | AUS ↑ | 1.010±0.003 | 1.008±0.003 | 1.012±0.004 | 1.019±0.003 | 1.013±0.005 | 1.014±0.005 | 1.009±0.003 | 1.010±0.005 | 1.008±0.004 | 1.011±0.004 |
| | $\mathcal{A}_r^t \uparrow$ | 88.80±0.60 | 88.27±0.57 | 89.34±0.60 | 90.92±0.50 | 88.88±0.57 | 90.18±0.62 | 88.31±0.58 | 88.55±0.64 | 88.25±0.67 | 88.30±0.59 |
| | $\mathcal{A}_f^t \downarrow$ | 0.0±0.0 | 0.0±0.0 | 0.0±0.0 | 0.0±0.0 | 0.0±0.0 | 0.0±0.0 | 0.0±0.0 | 0.0±0.0 | 0.0±0.0 | 0.0±0.0 |
| | AUS ↑ | 1.006±0.002 | 1.004±0.001 | 1.008±0.001 | 1.013±0.001 | 1.008±0.002 | 1.009±0.001 | 1.005±0.001 | 1.006±0.002 | 1.005±0.002 | 1.006±0.001 |
| NG (Golatkar et al., 2020) | $\mathcal{A}_r^t \uparrow$ | 87.20±3.36 | 88.34±0.44 | 89.54±0.42 | 91.24±0.20 | 89.09±0.40 | 90.54±0.35 | 88.53±0.37 | 88.67±0.41 | 88.03±0.35 | 88.43±0.42 |
| | $\mathcal{A}_f^t \downarrow$ | 0.0±0.0 | 0.0±0.0 | 0.0±0.0 | 0.0±0.0 | 0.0±0.0 | 0.0±0.0 | 0.0±0.0 | 0.0±0.0 | 0.1±0.1 | 0.0±0.0 |
| | AUS ↑ | 0.990±0.035 | 1.004±0.000 | 1.008±0.001 | 1.015±0.002 | 1.007±0.001 | 1.010±0.002 | 1.006±0.001 | 1.006±0.002 | 1.000±0.006 | 1.005±0.001 |
| | $\mathcal{A}_r^t \uparrow$ | 88.75±0.62 | 88.19±0.55 | 89.35±0.62 | 91.18±0.58 | 88.99±0.60 | 90.41±0.58 | 88.33±0.60 | 88.61±0.65 | 88.29±0.61 | 88.33±0.46 |
| | $\mathcal{A}_f^t \downarrow$ | 0.0±0.0 | 0.0±0.0 | 0.0±0.0 | 0.0±0.0 | 0.0±0.0 | 0.0±0.0 | 0.0±0.0 | 0.0±0.0 | 0.0±0.0 | 0.0±0.0 |
| | AUS ↑ | 1.006±0.001 | 1.004±0.001 | 1.008±0.001 | 1.016±0.001 | 1.009±0.001 | 1.012±0.002 | 1.006±0.001 | 1.006±0.003 | 1.006±0.003 | 1.006±0.002 |
| RL (Hayase et al., 2020) | $\mathcal{A}_r^t \uparrow$ | 88.86±0.60 | 88.25±0.55 | 89.38±0.60 | 91.14±0.54 | 89.02±0.58 | 90.30±0.59 | 88.39±0.59 | 88.59±0.63 | 88.30±0.66 | 88.40±0.50 |
| | $\mathcal{A}_f^t \downarrow$ | 0.0±0.0 | 0.0±0.0 | 0.0±0.0 | 0.0±0.0 | 0.0±0.0 | 0.0±0.0 | 0.0±0.0 | 0.0±0.0 | 0.0±0.0 | 0.0±0.0 |
| | AUS ↑ | 1.007±0.002 | 1.004±0.001 | 1.008±0.001 | 1.016±0.001 | 1.009±0.002 | 1.010±0.002 | 1.006±0.001 | 1.006±0.003 | 1.006±0.002 | 1.007±0.001 |
| | $\mathcal{A}_r^t \uparrow$ | 88.79±0.60 | 88.15±0.57 | 89.28±0.62 | 90.93±0.46 | 88.81±0.57 | 90.14±0.63 | 88.21±0.50 | 88.43±0.63 | 88.20±0.67 | 88.30±0.56 |
| | $\mathcal{A}_f^t \downarrow$ | 0.0±0.0 | 0.0±0.0 | 0.0±0.0 | 0.0±0.0 | 0.0±0.0 | 0.0±0.0 | 0.0±0.0 | 0.0±0.0 | 0.0±0.0 | 0.0±0.0 |
| | AUS ↑ | 1.006±0.001 | 1.003±0.001 | 1.007±0.001 | 1.013±0.002 | 1.007±0.001 | 1.009±0.002 | 1.004±0.001 | 1.004±0.002 | 1.005±0.002 | 1.006±0.001 |
| BS (Chen et al., 2023) | $\mathcal{A}_r^t \uparrow$ | 88.15±0.70 | 87.73±0.51 | 88.18±0.70 | 87.62±1.28 | 87.21±0.91 | 87.41±2.87 | 87.32±1.15 | 87.98±0.89 | 87.71±0.80 | 87.70±0.67 |
| | $\mathcal{A}_f^t \downarrow$ | 3.8±5.2 | 0.3±0.6 | 0.0±0.0 | 0.0±0.0 | 0.0±0.0 | 3.7±8.2 | 0.7±1.3 | 0.6±1.1 | 0.4±0.5 | 0.6±1.1 |
| | AUS ↑ | 0.965±0.045 | 0.999±0.002 | 0.993±0.008 | 0.980±0.012 | 0.991±0.008 | 0.950±0.066 | 0.989±0.014 | 0.995±0.012 | 0.996±0.004 | 0.993±0.009 |
| | $\mathcal{A}_r^t \uparrow$ | 88.68±0.58 | 88.44±0.46 | 89.48±0.33 | 91.14±0.58 | 89.15±0.29 | 90.58±0.32 | 88.66±0.37 | 88.78±0.42 | 88.61±0.06 | 88.71±0.39 |
| | $\mathcal{A}_f^t \downarrow$ | 0.0±0.0 | 0.0±0.0 | 0.0±0.0 | 0.0±0.0 | 0.0±0.0 | 0.0±0.0 | 0.0±0.0 | 0.0±0.0 | 0.0±0.0 | 0.0±0.0 |
| | AUS ↑ | 1.005±0.001 | 1.005±0.001 | 1.007±0.002 | 1.014±0.002 | 1.008±0.002 | 1.011±0.003 | 1.007±0.003 | 1.005±0.002 | 1.003±0.001 | 1.006±0.001 |
| BE (Chen et al., 2023) | $\mathcal{A}_r^t \uparrow$ | 87.68±0.52 | 87.24±0.53 | 88.27±0.64 | 86.68±4.42 | 87.86±0.57 | 89.21±0.68 | 86.89±0.69 | 87.39±0.78 | 87.14±0.80 | 86.79±0.60 |
| | $\mathcal{A}_f^t \downarrow$ | 0.0±0.0 | 0.0±0.0 | 0.0±0.0 | 1.2±1.6 | 0.5±1.1 | 10.1±14.9 | 0.0±0.0 | 0.0±0.0 | 0.0±0.0 | 0.0±0.0 |
| | AUS ↑ | 0.995±0.002 | 0.994±0.004 | 0.997±0.001 | 0.959±0.033 | 0.993±0.010 | 0.919±0.107 | 0.991±0.003 | 0.994±0.001 | 0.994±0.002 | 0.990±0.002 |
| | $\mathcal{A}_r^t \uparrow$ | 88.14±0.58 | 87.81±0.50 | 88.51±0.61 | 89.47±0.55 | 88.08±0.70 | 89.19±0.76 | 87.70±0.61 | 87.97±0.71 | 87.67±0.85 | 87.69±0.65 |
| | $\mathcal{A}_f^t \downarrow$ | 0.0±0.0 | 0.0±0.0 | 0.0±0.0 | 0.0±0.0 | 0.0±0.0 | 0.0±0.0 | 0.0±0.0 | 0.0±0.0 | 0.0±0.0 | 0.0±0.0 |
| | AUS ↑ | 1.000±0.000 | 1.000±0.000 | 0.999±0.000 | 0.999±0.001 | 1.000±0.000 | 0.999±0.000 | 0.999±0.001 | 1.000±0.000 | 0.999±0.001 | 1.000±0.000 |
| DELETE (Zhou et al., 2025) | $\mathcal{A}_r^t \uparrow$ | 88.78±0.60 | 88.17±0.56 | 89.33±0.62 | 91.04±0.51 | 88.90±0.59 | 90.25±0.59 | 88.33±0.53 | 88.52±0.63 | 88.24±0.66 | 88.32±0.50 |
| | $\mathcal{A}_f^t \downarrow$ | 0.0±0.0 | 0.0±0.0 | 0.0±0.0 | 0.0±0.0 | 0.0±0.0 | 0.0±0.0 | 0.0±0.0 | 0.0±0.0 | 0.0±0.0 | 0.0±0.0 |
| | AUS ↑ | 1.006±0.001 | 1.003±0.001 | 1.008±0.001 | 1.014±0.002 | 1.008±0.001 | 1.010±0.001 | 1.006±0.001 | 1.005±0.002 | 1.005±0.002 | 1.006±0.001 |
| | $\mathcal{A}_r^t \uparrow$ | 88.76±0.62 | 88.16±0.55 | 89.33±0.63 | 91.04±0.51 | 88.91±0.60 | 90.26±0.61 | 88.30±0.62 | 88.50±0.66 | 88.23±0.64 | 88.32±0.52 |
| | $\mathcal{A}_f^t \downarrow$ | 0.0±0.0 | 0.0±0.0 | 0.0±0.0 | 0.0±0.0 | 0.0±0.0 | 0.0±0.0 | 0.0±0.0 | 0.0±0.0 | 0.0±0.0 | 0.0±0.0 |
| | AUS ↑ | 1.006±0.001 | 1.003±0.001 | 1.008±0.001 | 1.014±0.002 | 1.008±0.001 | 1.010±0.001 | 1.005±0.001 | 1.005±0.002 | 1.005±0.002 | 1.006±0.001 |
| NG+ (Kurmanji et al., 2023) | $\mathcal{A}_r^t \uparrow$ | 88.91±0.60 | 88.47±0.57 | 89.54±0.54 | 90.96±0.49 | 89.09±0.57 | 90.33±0.62 | 88.43±0.71 | 88.64±0.62 | 88.24±0.51 | 88.59±0.48 |
| | $\mathcal{A}_f^t \downarrow$ | 0.0±0.0 | 0.0±0.0 | 0.0±0.0 | 0.0±0.0 | 0.0±0.0 | 0.0±0.0 | 0.0±0.0 | 0.0±0.0 | 0.0±0.0 | 0.0±0.0 |
| | AUS ↑ | 1.007±0.003 | 1.006±0.003 | 1.008±0.003 | 1.014±0.003 | 1.010±0.003 | 1.011±0.003 | 1.007±0.003 | 1.007±0.003 | 1.005±0.004 | 1.009±0.003 |
| | $\mathcal{A}_r^t \uparrow$ | 88.77±0.66 | 88.22±0.57 | 89.35±0.64 | 90.96±0.52 | 88.97±0.62 | 90.24±0.63 | 88.32±0.62 | 88.51±0.64 | 88.25±0.65 | 88.33±0.55 |
| | $\mathcal{A}_f^t \downarrow$ | 0.0±0.0 | 0.0±0.0 | 0.0±0.0 | 0.0±0.0 | 0.0±0.0 | 0.0±0.0 | 0.0±0.0 | 0.0±0.0 | 0.0±0.0 | 0.0±0.0 |
| | AUS ↑ | 1.006±0.002 | 1.004±0.001 | 1.008±0.001 | 1.014±0.001 | 1.009±0.001 | 1.010±0.002 | 1.006±0.001 | 1.005±0.001 | 1.005±0.002 | 1.006±0.001 |
| SCRUB (Kurmanji et al., 2023) | $\mathcal{A}_r^t \uparrow$ | 88.87±0.57 | 88.33±0.47 | 89.33±0.44 | 90.70±0.35 | 88.92±0.53 | 90.16±0.62 | 88.28±0.64 | 88.47±0.51 | 88.16±0.64 | 88.39±0.59 |
| | $\mathcal{A}_f^t \downarrow$ | 0.0±0.0 | 0.0±0.0 | 0.0±0.0 | 0.0±0.0 | 0.0±0.0 | 0.0±0.0 | 0.0±0.0 | 0.0±0.0 | 0.0±0.0 | 0.0±0.0 |
| | AUS ↑ | 1.007±0.002 | 1.006±0.003 | 1.009±0.003 | 1.013±0.002 | 1.010±0.003 | 1.011±0.003 | 1.007±0.003 | 1.006±0.003 | 1.006±0.004 | 1.008±0.002 |
| | $\mathcal{A}_r^t \uparrow$ | 88.82±0.56 | 88.32±0.55 | 89.37±0.65 | 91.25±0.59 | 89.02±0.57 | 90.46±0.67 | 88.37±0.60 | 88.66±0.64 | 88.32±0.59 | 88.46±0.53 |
| | $\mathcal{A}_f^t \downarrow$ | 0.0±0.0 | 0.0±0.0 | 0.0±0.0 | 0.0±0.0 | 0.0±0.0 | 0.0±0.0 | 0.0±0.0 | 0.0±0.0 | 0.0±0.0 | 0.0±0.0 |
| | AUS ↑ | 1.006±0.002 | 1.005±0.001 | 1.008±0.001 | 1.017±0.002 | 1.009±0.002 | 1.012±0.002 | 1.006±0.002 | 1.007±0.003 | 1.006±0.003 | 1.007±0.001 |
| SCAR (Bonato et al., 2024) | $\mathcal{A}_r^t \uparrow$ | 88.87±0.58 | 88.31±0.54 | 89.39±0.45 | 91.25±0.52 | 89.02±0.61 | 90.43±0.50 | 88.37±0.58 | 88.64±0.66 | 88.37±0.60 | 88.44±0.48 |
| | $\mathcal{A}_f^t \downarrow$ | 0.0±0.0 | 0.0±0.0 | 0.0±0.0 | 0.0±0.0 | 0.0±0.0 | 0.0±0.0 | 0.0±0.0 | 0.0±0.0 | 0.0±0.0 | 0.0±0.0 |
| | AUS ↑ | 1.007±0.002 | 1.005±0.001 | 1.008±0.001 | 1.017±0.002 | 1.009±0.001 | 1.012±0.003 | 1.006±0.001 | 1.007±0.003 | 1.006±0.002 | 1.007±0.002 |
| | $\mathcal{A}_r^t \uparrow$ | 88.81±0.63 | 88.21±0.56 | 89.36±0.64 | 91.06±0.51 | 88.95±0.60 | 90.29±0.63 | 88.34±0.61 | 88.56±0.65 | 88.26±0.64 | 88.36±0.54 |
| | $\mathcal{A}_f^t \downarrow$ | 0.0±0.0 | 0.0±0.0 | 0.0±0.0 | 0.0±0.0 | 0.0±0.0 | 0.0±0.0 | 0.0±0.0 | 0.0±0.0 | 0.0±0.0 | 0.0±0.0 |
| | AUS ↑ | 1.006±0.001 | 1.004±0.001 | 1.008±0.001 | 1.015±0.002 | 1.008±0.001 | 1.010±0.001 | 1.006±0.001 | 1.006±0.002 | 1.005±0.002 | 1.006±0.001 |

Table 9: Class unlearning performance for CIFAR-10 using ViT-B-16, averaged over 5 random trials. Rows highlighted in gray represent our results using synthetic embeddings, while the corresponding non-shaded rows use original embeddings with the same method.

| Method | Metric | Forget Class | | | | | | | | | |
|---|---|---|---|---|---|---|---|---|---|---|---|
| | | 0 | 1 | 2 | 3 | 4 | 5 | 6 | 7 | 8 | 9 |
| Original | $\mathcal{A}_r^t \uparrow$ | 97.65±0.07 | 97.60±0.11 | 97.71±0.13 | 97.98±0.11 | 97.68±0.18 | 97.88±0.11 | 97.55±0.13 | 97.63±0.08 | 97.55±0.13 | 97.65±0.15 |
| | $\mathcal{A}_f^t \downarrow$ | 98.0±0.6 | 98.5±0.5 | 97.5±0.2 | 95.1±0.8 | 97.8±0.7 | 95.9±0.2 | 98.9±0.3 | 98.2±0.9 | 98.9±0.2 | 98.0±0.4 |
| | AUS ↑ | 0.505±0.002 | 0.504±0.001 | 0.506±0.001 | 0.513±0.002 | 0.506±0.002 | 0.510±0.000 | 0.503±0.001 | 0.505±0.002 | 0.503±0.000 | 0.505±0.001 |
| Retrained | $\mathcal{A}_r^t \uparrow$ | 98.39 | 98.38 | 98.21 | 98.86 | 98.38 | 98.67 | 98.17 | 98.28 | 98.20 | 98.31 |
| | $\mathcal{A}_f^t \downarrow$ | 0.0 | 0.0 | 0.0 | 0.0 | 0.0 | 0.0 | 0.0 | 0.0 | 0.0 | 0.0 |
| | AUS ↑ | 1.007 | 1.008 | 1.005 | 1.009 | 1.007 | 1.008 | 1.006 | 1.006 | 1.006 | 1.007 |
| NG (Golatkar et al., 2020) | $\mathcal{A}_r^t \uparrow$ | 97.80±0.10 | 97.78±0.10 | 97.85±0.12 | 98.34±0.09 | 97.92±0.12 | 98.27±0.14 | 97.64±0.14 | 97.77±0.07 | 97.71±0.13 | 97.82±0.12 |
| | $\mathcal{A}_f^t \downarrow$ | 0.0±0.0 | 0.0±0.0 | 0.0±0.0 | 0.0±0.0 | 0.0±0.0 | 0.0±0.0 | 0.0±0.0 | 0.0±0.0 | 0.0±0.0 | 0.0±0.0 |
| | AUS ↑ | 1.002±0.000 | 1.002±0.000 | 1.001±0.000 | 1.004±0.001 | 1.002±0.001 | 1.004±0.001 | 1.001±0.000 | 1.001±0.001 | 1.002±0.000 | 1.002±0.001 |
| | $\mathcal{A}_r^t \uparrow$ | 97.81±0.10 | 97.78±0.10 | 97.86±0.12 | 98.34±0.09 | 97.93±0.12 | 98.28±0.13 | 97.65±0.14 | 97.77±0.06 | 97.71±0.14 | 97.82±0.12 |
| | $\mathcal{A}_f^t \downarrow$ | 0.0±0.0 | 0.0±0.0 | 0.0±0.0 | 0.0±0.0 | 0.0±0.0 | 0.0±0.0 | 0.0±0.0 | 0.0±0.0 | 0.0±0.0 | 0.0±0.0 |
| | AUS ↑ | 1.002±0.000 | 1.002±0.000 | 1.002±0.000 | 1.004±0.001 | 1.003±0.001 | 1.004±0.001 | 1.001±0.000 | 1.001±0.001 | 1.002±0.000 | 1.002±0.001 |
| RL (Hayase et al., 2020) | $\mathcal{A}_r^t \uparrow$ | 97.81±0.10 | 97.79±0.10 | 97.85±0.12 | 98.37±0.10 | 97.94±0.12 | 98.28±0.12 | 97.68±0.13 | 97.78±0.07 | 97.72±0.14 | 97.83±0.12 |
| | $\mathcal{A}_f^t \downarrow$ | 0.0±0.0 | 0.0±0.0 | 0.0±0.0 | 0.0±0.0 | 0.0±0.0 | 0.0±0.0 | 0.0±0.0 | 0.0±0.0 | 0.0±0.0 | 0.0±0.0 |
| | AUS ↑ | 1.002±0.000 | 1.002±0.000 | 1.001±0.000 | 1.004±0.001 | 1.003±0.001 | 1.004±0.001 | 1.001±0.000 | 1.001±0.001 | 1.002±0.000 | 1.002±0.001 |
| | $\mathcal{A}_r^t \uparrow$ | 97.85±0.07 | 97.84±0.12 | 97.90±0.13 | 98.38±0.11 | 97.96±0.13 | 98.30±0.13 | 97.70±0.12 | 97.83±0.06 | 97.75±0.15 | 97.83±0.12 |
| | $\mathcal{A}_f^t \downarrow$ | 0.0±0.0 | 0.0±0.0 | 0.0±0.0 | 0.0±0.0 | 0.0±0.0 | 0.0±0.0 | 0.0±0.0 | 0.0±0.0 | 0.0±0.0 | 0.0±0.0 |
| | AUS ↑ | 1.002±0.000 | 1.002±0.000 | 1.002±0.000 | 1.004±0.000 | 1.003±0.001 | 1.004±0.000 | 1.001±0.000 | 1.002±0.001 | 1.002±0.000 | 1.002±0.000 |
| BS (Chen et al., 2023) | $\mathcal{A}_r^t \uparrow$ | 97.67±0.12 | 97.75±0.08 | 97.79±0.19 | 98.13±0.24 | 97.74±0.19 | 97.97±0.19 | 97.61±0.15 | 97.68±0.07 | 97.60±0.14 | 97.72±0.22 |
| | $\mathcal{A}_f^t \downarrow$ | 0.0±0.0 | 0.0±0.0 | 0.0±0.0 | 0.0±0.0 | 0.0±0.0 | 0.0±0.0 | 0.0±0.0 | 0.0±0.0 | 0.0±0.0 | 0.0±0.0 |
| | AUS ↑ | 1.000±0.001 | 1.002±0.001 | 1.001±0.001 | 1.002±0.002 | 1.001±0.000 | 1.001±0.001 | 1.001±0.000 | 1.000±0.000 | 1.000±0.001 | 1.001±0.000 |
| | $\mathcal{A}_r^t \uparrow$ | 97.80±0.10 | 97.76±0.07 | 97.86±0.11 | 98.29±0.08 | 97.92±0.13 | 98.25±0.14 | 97.67±0.14 | 97.79±0.05 | 97.69±0.17 | 97.82±0.11 |
| | $\mathcal{A}_f^t \downarrow$ | 0.0±0.0 | 0.0±0.0 | 0.0±0.0 | 0.0±0.0 | 0.0±0.0 | 0.0±0.0 | 0.0±0.0 | 0.0±0.0 | 0.0±0.0 | 0.0±0.0 |
| | AUS ↑ | 1.002±0.000 | 1.002±0.001 | 1.001±0.000 | 1.003±0.001 | 1.002±0.001 | 1.004±0.000 | 1.001±0.000 | 1.002±0.001 | 1.001±0.000 | 1.002±0.001 |
| DELETE (Zhou et al., 2025) | $\mathcal{A}_r^t \uparrow$ | 97.81±0.10 | 97.78±0.09 | 97.85±0.12 | 98.34±0.10 | 97.93±0.12 | 98.28±0.13 | 97.64±0.13 | 97.77±0.07 | 97.71±0.13 | 97.82±0.12 |
| | $\mathcal{A}_f^t \downarrow$ | 0.0±0.0 | 0.0±0.0 | 0.0±0.0 | 0.0±0.0 | 0.0±0.0 | 0.0±0.0 | 0.0±0.0 | 0.0±0.0 | 0.0±0.0 | 0.0±0.0 |
| | AUS ↑ | 1.002±0.000 | 1.002±0.000 | 1.001±0.000 | 1.004±0.001 | 1.003±0.001 | 1.004±0.001 | 1.001±0.000 | 1.001±0.001 | 1.002±0.000 | 1.002±0.001 |
| | $\mathcal{A}_r^t \uparrow$ | 97.81±0.09 | 97.79±0.10 | 97.87±0.13 | 98.35±0.10 | 97.95±0.12 | 98.29±0.13 | 97.66±0.15 | 97.79±0.07 | 97.72±0.14 | 97.83±0.13 |
| | $\mathcal{A}_f^t \downarrow$ | 0.0±0.0 | 0.0±0.0 | 0.0±0.0 | 0.0±0.0 | 0.0±0.0 | 0.0±0.0 | 0.0±0.0 | 0.0±0.0 | 0.0±0.0 | 0.0±0.0 |
| | AUS ↑ | 1.002±0.000 | 1.002±0.000 | 1.002±0.000 | 1.004±0.000 | 1.003±0.001 | 1.004±0.001 | 1.001±0.000 | 1.002±0.001 | 1.002±0.000 | 1.002±0.000 |
| NG+ (Kurmanji et al., 2023) | $\mathcal{A}_r^t \uparrow$ | 97.79±0.10 | 97.77±0.13 | 97.83±0.13 | 98.34±0.09 | 97.91±0.13 | 98.26±0.15 | 97.64±0.14 | 97.75±0.08 | 97.69±0.14 | 97.81±0.13 |
| | $\mathcal{A}_f^t \downarrow$ | 0.0±0.0 | 0.0±0.0 | 0.0±0.0 | 0.0±0.0 | 0.0±0.0 | 0.0±0.0 | 0.0±0.0 | 0.0±0.0 | 0.0±0.0 | 0.0±0.0 |
| | AUS ↑ | 1.001±0.000 | 1.002±0.000 | 1.001±0.000 | 1.004±0.001 | 1.002±0.001 | 1.004±0.001 | 1.001±0.000 | 1.001±0.001 | 1.001±0.000 | 1.002±0.001 |
| | $\mathcal{A}_r^t \uparrow$ | 97.82±0.10 | 97.80±0.10 | 97.88±0.13 | 98.37±0.10 | 97.96±0.12 | 98.31±0.14 | 97.67±0.14 | 97.79±0.07 | 97.74±0.14 | 97.85±0.13 |
| | $\mathcal{A}_f^t \downarrow$ | 0.0±0.0 | 0.0±0.0 | 0.0±0.0 | 0.0±0.0 | 0.0±0.0 | 0.0±0.0 | 0.0±0.0 | 0.0±0.0 | 0.0±0.0 | 0.0±0.0 |
| | AUS ↑ | 1.002±0.000 | 1.002±0.000 | 1.002±0.000 | 1.004±0.000 | 1.003±0.001 | 1.004±0.001 | 1.001±0.000 | 1.002±0.001 | 1.002±0.000 | 1.002±0.000 |

Table 10: Class unlearning performance for CIFAR-10 using Swin-T, averaged over 5 random trials. Rows highlighted in gray represent our results using synthetic embeddings, while the corresponding non-shaded rows use original embeddings with the same method.

| Method | Metric | Forget Class | | | | | | | | | |
|---|---|---|---|---|---|---|---|---|---|---|---|
| | | 0 | 1 | 2 | 3 | 4 | 5 | 6 | 7 | 8 | 9 |
| Original | $\mathcal{A}_r^t \uparrow$ | 97.58±0.08 | 97.65±0.05 | 97.78±0.08 | 97.96±0.15 | 97.74±0.03 | 98.03±0.10 | 97.55±0.05 | 97.63±0.08 | 97.60±0.07 | 97.74±0.09 |
| | $\mathcal{A}_f^t \downarrow$ | 99.0±0.3 | 98.4±0.5 | 97.3±0.6 | 95.6±0.9 | 97.6±0.7 | 95.0±0.9 | 99.3±0.3 | 98.6±0.3 | 98.8±0.1 | 97.6±0.3 |
| | AUS ↑ | 0.502±0.001 | 0.504±0.001 | 0.507±0.002 | 0.511±0.002 | 0.506±0.002 | 0.513±0.002 | 0.502±0.001 | 0.504±0.001 | 0.503±0.001 | 0.506±0.001 |
| Retrained | $\mathcal{A}_r^t \uparrow$ | 98.22 | 98.30 | 98.31 | 98.80 | 98.30 | 98.73 | 98.14 | 98.14 | 98.17 | 98.43 |
| | $\mathcal{A}_f^t \downarrow$ | 0.0 | 0.0 | 0.0 | 0.0 | 0.0 | 0.0 | 0.0 | 0.0 | 0.0 | 0.0 |
| | AUS ↑ | 1.006 | 1.006 | 1.005 | 1.008 | 1.006 | 1.007 | 1.006 | 1.005 | 1.006 | 1.007 |
| NG (Golatkar et al., 2020) | $\mathcal{A}_r^t \uparrow$ | 97.73±0.05 | 97.86±0.05 | 97.88±0.07 | 98.46±0.10 | 97.91±0.04 | 98.37±0.12 | 97.65±0.06 | 97.76±0.06 | 97.74±0.07 | 97.88±0.08 |
| | $\mathcal{A}_f^t \downarrow$ | 0.0±0.0 | 0.0±0.0 | 0.0±0.0 | 0.0±0.0 | 0.0±0.0 | 0.0±0.0 | 0.0±0.0 | 0.0±0.0 | 0.0±0.0 | 0.0±0.0 |
| | AUS ↑ | 1.002±0.000 | 1.002±0.000 | 1.001±0.000 | 1.005±0.001 | 1.002±0.000 | 1.003±0.001 | 1.001±0.000 | 1.001±0.000 | 1.001±0.000 | 1.001±0.000 |
| | $\mathcal{A}_r^t \uparrow$ | 97.71±0.10 | 97.65±0.28 | 97.74±0.08 | 97.31±2.61 | 97.90±0.07 | 97.81±0.60 | 97.28±0.72 | 97.66±0.19 | 97.71±0.08 | 97.71±0.11 |
| | $\mathcal{A}_f^t \downarrow$ | 0.0±0.0 | 1.1±1.6 | 0.3±0.3 | 0.9±1.9 | 0.1±0.2 | 1.1±1.6 | 0.2±0.3 | 0.2±0.2 | 0.1±0.1 | 0.7±0.6 |
| | AUS ↑ | 1.001±0.000 | 0.990±0.017 | 0.997±0.003 | 0.985±0.044 | 1.001±0.002 | 0.987±0.023 | 0.995±0.010 | 0.998±0.004 | 1.001±0.001 | 0.993±0.007 |
| NG+ (Kurmanji et al., 2023) | $\mathcal{A}_r^t \uparrow$ | 97.67±0.07 | 97.67±0.06 | 97.85±0.06 | 98.32±0.13 | 97.81±0.02 | 98.32±0.14 | 97.59±0.05 | 97.66±0.06 | 97.64±0.04 | 97.76±0.07 |
| | $\mathcal{A}_f^t \downarrow$ | 0.0±0.0 | 0.0±0.0 | 0.0±0.0 | 0.0±0.0 | 0.0±0.0 | 0.0±0.0 | 0.0±0.0 | 0.0±0.0 | 0.0±0.0 | 0.0±0.0 |
| | AUS ↑ | 1.001±0.000 | 1.000±0.000 | 1.001±0.000 | 1.004±0.001 | 1.001±0.000 | 1.003±0.001 | 1.000±0.000 | 1.000±0.000 | 1.000±0.000 | 1.000±0.000 |
| | $\mathcal{A}_r^t \uparrow$ | 97.45±0.30 | 90.38±11.22 | 92.22±6.15 | 95.45±3.03 | 97.15±0.70 | 95.16±0.77 | 85.57±16.19 | 94.36±5.09 | 94.61±4.95 | 92.80±4.31 |
| | $\mathcal{A}_f^t \downarrow$ | 0.0±0.1 | 2.1±1.5 | 1.2±0.7 | 0.9±1.4 | 0.1±0.2 | 3.0±0.5 | 1.2±1.8 | 0.8±0.5 | 0.5±0.5 | 2.2±1.6 |
| | AUS ↑ | 0.998±0.004 | 0.908±0.113 | 0.933±0.060 | 0.967±0.041 | 0.993±0.007 | 0.943±0.009 | 0.872±0.172 | 0.960±0.052 | 0.966±0.053 | 0.931±0.054 |
| SCRUB (Kurmanji et al., 2023) | $\mathcal{A}_r^t \uparrow$ | 97.63±0.09 | 97.68±0.06 | 97.84±0.07 | 98.38±0.09 | 97.87±0.03 | 98.22±0.10 | 97.65±0.06 | 97.73±0.04 | 97.69±0.04 | 97.79±0.04 |
| | $\mathcal{A}_f^t \downarrow$ | 0.0±0.0 | 0.0±0.0 | 0.0±0.0 | 0.0±0.0 | 0.0±0.0 | 0.0±0.0 | 0.0±0.0 | 0.0±0.0 | 0.0±0.0 | 0.0±0.0 |
| | AUS ↑ | 1.000±0.001 | 1.000±0.001 | 1.001±0.000 | 1.004±0.001 | 1.001±0.000 | 1.002±0.001 | 1.001±0.000 | 1.001±0.001 | 1.001±0.001 | 1.001±0.001 |
| | $\mathcal{A}_r^t \uparrow$ | 97.59±0.07 | 97.49±0.35 | 97.57±0.18 | 97.93±0.38 | 96.70±2.09 | 97.94±0.37 | 97.23±0.53 | 96.36±2.62 | 97.38±0.29 | 97.71±0.10 |
| | $\mathcal{A}_f^t \downarrow$ | 0.0±0.0 | 0.0±0.0 | 0.0±0.0 | 0.0±0.0 | 0.0±0.0 | 0.0±0.0 | 0.0±0.0 | 0.0±0.0 | 0.0±0.0 | 0.0±0.0 |
| | AUS ↑ | 1.000±0.001 | 0.998±0.003 | 0.998±0.002 | 1.000±0.005 | 0.990±0.021 | 0.999±0.003 | 0.997±0.005 | 0.987±0.026 | 0.998±0.003 | 1.000±0.001 |

Table 11: Multi-class unlearning performance on CIFAR-100 using ResNet-18 as the base architecture. Rows highlighted in gray correspond to methods applied on synthetic embeddings, while the non-shaded rows use original embeddings. Columns $\mathcal{D}_r$-free and $\mathcal{D}_f$-free indicate whether the method operates without access to the retain or forget set, respectively, with (✓) indicating data-free operation and (✗) indicating that the corresponding data is required.

| Method | $\mathcal{D}_r$ free | $\mathcal{D}_f$ free | 2-Classes $\mathcal{A}_r^t\uparrow$ | $\mathcal{A}_f^t\downarrow$ | AUS ↑ | 5-Classes $\mathcal{A}_r^t\uparrow$ | $\mathcal{A}_f^t\downarrow$ | AUS ↑ | 10-Classes $\mathcal{A}_r^t\uparrow$ | $\mathcal{A}_f^t\downarrow$ | AUS ↑ |
|---|---|---|---|---|---|---|---|---|---|---|---|
| Original | – | – | 78.12 ± 1.21 | 80.10 ± 3.19 | 0.555 ± 0.010 | 78.14 ± 1.26 | 78.68 ± 0.92 | 0.560 ± 0.003 | 78.01 ± 1.31 | 79.58 ± 0.93 | 0.557 ± 0.003 |
| FT (Golatkar et al., 2020) | ✗ | ✓ | 78.24 ± 1.07 | 0.00 ± 0.00 | 1.001 ± 0.003 | 78.63 ± 1.22 | 0.00 ± 0.00 | 1.005 ± 0.003 | 78.80 ± 1.04 | 0.02 ± 0.04 | 1.008 ± 0.004 |
|  | ✓ | ✓ | 78.26 ± 1.17 | 0.00 ± 0.00 | 1.001 ± 0.001 | 78.60 ± 1.18 | 0.00 ± 0.00 | 1.005 ± 0.001 | 78.88 ± 1.17 | 0.02 ± 0.04 | 1.009 ± 0.002 |
| NG (Golatkar et al., 2020) | ✓ | ✗ | 78.37 ± 1.15 | 0.00 ± 0.00 | 1.002 ± 0.001 | 78.67 ± 1.19 | 0.00 ± 0.00 | 1.005 ± 0.001 | 78.97 ± 1.20 | 0.00 ± 0.00 | 1.010 ± 0.002 |
|  | ✓ | ✓ | 78.34 ± 1.12 | 0.00 ± 0.00 | 1.002 ± 0.001 | 78.68 ± 1.13 | 0.00 ± 0.00 | 1.005 ± 0.002 | 78.99 ± 1.12 | 0.00 ± 0.00 | 1.010 ± 0.002 |
| RL (Hayase et al., 2020) | ✓ | ✗ | 78.25 ± 1.12 | 0.00 ± 0.00 | 1.001 ± 0.002 | 78.10 ± 1.07 | 0.00 ± 0.00 | 1.000 ± 0.003 | 78.62 ± 1.10 | 0.00 ± 0.00 | 1.006 ± 0.003 |
|  | ✓ | ✓ | 77.95 ± 1.03 | 0.00 ± 0.00 | 0.998 ± 0.003 | 76.25 ± 0.81 | 0.04 ± 0.09 | 0.981 ± 0.011 | 74.38 ± 1.41 | 0.18 ± 0.35 | 0.962 ± 0.014 |
| DELETE (Zhou et al., 2025) | ✓ | ✗ | 78.37 ± 1.12 | 0.00 ± 0.00 | 1.002 ± 0.001 | 78.71 ± 1.14 | 0.00 ± 0.00 | 1.006 ± 0.001 | 79.01 ± 1.13 | 0.00 ± 0.00 | 1.010 ± 0.002 |
|  | ✓ | ✓ | 78.33 ± 1.13 | 0.00 ± 0.00 | 1.002 ± 0.001 | 78.66 ± 1.15 | 0.00 ± 0.00 | 1.005 ± 0.001 | 78.96 ± 1.14 | 0.66 ± 1.01 | 1.003 ± 0.010 |
| NG+ (Kurmanji et al., 2023) | ✗ | ✗ | 78.47 ± 1.05 | 0.00 ± 0.00 | 1.003 ± 0.002 | 78.79 ± 1.04 | 0.00 ± 0.00 | 1.006 ± 0.003 | 79.14 ± 1.02 | 0.00 ± 0.00 | 1.011 ± 0.003 |
|  | ✓ | ✓ | 78.34 ± 1.10 | 0.00 ± 0.00 | 1.002 ± 0.001 | 78.63 ± 1.11 | 0.00 ± 0.00 | 1.005 ± 0.002 | 78.97 ± 1.13 | 0.00 ± 0.00 | 1.010 ± 0.002 |
| SCRUB (Kurmanji et al., 2023) | ✗ | ✗ | 77.61 ± 1.01 | 0.00 ± 0.00 | 0.995 ± 0.003 | 78.27 ± 1.05 | 0.00 ± 0.00 | 1.001 ± 0.003 | 78.93 ± 1.07 | 0.00 ± 0.00 | 1.009 ± 0.003 |
|  | ✓ | ✓ | 78.26 ± 1.04 | 0.00 ± 0.00 | 1.001 ± 0.002 | 78.48 ± 1.12 | 0.00 ± 0.00 | 1.003 ± 0.003 | 78.52 ± 1.08 | 0.00 ± 0.00 | 1.005 ± 0.004 |

