# OpenReview forum: "A Universal Source-Free Class Unlearning Framework via Synthetic Embeddings"
_ICLR.cc/2026/Conference — Submitted to ICLR 2026_

### Official Review · Reviewer_KPAB · 2025-10-27

**Soundness:** 3
**Presentation:** 3
**Contribution:** 3
**Rating:** 6
**Confidence:** 3

**Summary:**

This paper is targeted at source-free class unlearning task. The authors propose one method that only needs to randomly sample from the intermediate distribution space of the classifier. Using the sampled features to calculate the loss of forgetting and retaining is sufficient. The authors use the proposition to support their claim. The experiments also show the effectiveness of their method.

**Strengths:**

1. The writing is easy to follow and understanding.

2. The proposed methods is clear and easy to implement.

3. The method is supported by the proposition proved by the authors.

4. The experiments show the effectiveness of the method.

**Weaknesses:**

1. The results in Table 1 for ResNet-18 is unusual, where the model retrained with access to original samples of forget and retain classes perform worse than all methods. But it should be the upper bound of this task.

2. All experiments are conducted when the number of class to forget is only 1 and the method is also based on this presumption. Could you please conduct experiments when the number of class to forget is greater than 1 and forget the classes one by one/at the same time?

**Questions:**

1. I wonder how the method perform if the sampled features are exactly the features extracted from real samples of forget/retain classes.

2. I wonder whether the method is applicable to other tasks such as detection.

---

> ### Author Response · Authors · 2025-11-21
> **Response to Reviewer KPAB (Part 1)**
>
> We thank the reviewer for the careful reading of our paper and for the positive assessment of the clarity of the writing, the simplicity and implementability of the method, the supporting proposition, and the overall empirical effectiveness. We appreciate your insightful comments and questions.
> ### **Weakness 1. Clarification on retrained baseline performance**
> ---
>
> While the “Retrained” model represents the gold standard for forgetting efficacy (as it has never seen the forget class), it does not necessarily constitute the upper bound for retain accuracy. We attribute the superior retain performance of our source-free unlearning method to two primary factors:
>
> &#9679; The Retrained model is trained from scratch on a smaller subset of data (excluding the forget class), potentially resulting in slightly less robust feature representations compared to the Original model, which was trained on the full dataset. Our unlearning procedure initializes from the Original model, thereby preserving the rich feature representations learned from the complete data distribution while only making targeted adjustments to the decision boundaries.
>
> &#9679; As noted in our method, the synthetic embedding sampling acts as a form of data augmentation and regularization. By generating embeddings with varying confidence levels within the intermediate space, the method may encourage smoother decision boundaries, leading to slightly improved generalization on the retain classes compared to the Retrained baseline.

---

> ### Author Response · Authors · 2025-11-21
> **Response to Reviewer KPAB (Part 2)**
>
> ### **Weakness 2. Evaluation of multi-class unlearning**
> ---
> To address this suggestion, we have extended our experiments to include multi-class unlearning. This is added to the Table 11 in Appendix F. Below, we summarize the experiment and results:
>
> Using the ResNet-18 backbone on CIFAR-100, we evaluated the removal of multiple classes simultaneously. Following the setup in Zhou et al. (2025), we tested three scenarios:
>
> &#9679;  2 classes : $\mathcal{Y}_f=${$25, 58$}
>
> &#9679;  5 classes: $\mathcal{Y}_f=${$25, 58, 38, 23, 96$}
>
> &#9679;  10 classes: $\mathcal{Y}_f=${$25, 58, 38, 23, 96, 54, 51, 49, 98, 66$}
>
> The results, averaged over five random seeds, demonstrate that our framework successfully drives the forget accuracy to near zero for all target classes while maintaining high retain accuracy. This confirms that, as expected, our source-free framework scales effectively beyond single-class unlearning.
>
> [1] Yu Zhou, Dian Zheng, Qijie Mo, Renjie Lu, Kun-Yu Lin, and Wei-Shi Zheng. Decoupled distillation to erase: A general unlearning method for any class-centric tasks. In Proceedings of the Computer Vision and Pattern Recognition Conference, pp. 20350–20359, 2025.
>
> **Multi-class (2 forget classes) unlearning performance on CIFAR-100 using ResNet-18. (real) rows use real embeddings and (synthetic) rows use synthetic embeddings:**
>
> | Method            | $\mathcal{D}_r$-free | $\mathcal{D}_f$-free | $\mathcal{A}^t_r$(%) $\uparrow$        | $\mathcal{A}^t_f$ (%)$\downarrow$        | AUS $\uparrow$              |
> |:-----------------|:--------------------:|:--------------------:|:---------------------------------:|:-----------------------------------:|:---------------------------:|
> | Original          | --                   | --                   | $78.12 \pm 1.21$                  | $80.10 \pm 3.19$                    | $0.555 \pm 0.010$           |
> | FT (real)               | ✗                    | ✓                    | $78.24 \pm 1.07$                  | $0.00 \pm 0.00$                     | $1.001 \pm 0.003$           |
> | FT (synthetic)    | ✓                    | ✓                    | $78.26 \pm 1.17$                  | $0.00 \pm 0.00$                     | $1.001 \pm 0.001$           |
> | NG (real)                | ✓                    | ✗                    | $78.37 \pm 1.15$                  | $0.00 \pm 0.00$                     | $1.002 \pm 0.001$           |
> | NG (synthetic)    | ✓                    | ✓                    | $78.34 \pm 1.12$                  | $0.00 \pm 0.00$                     | $1.002 \pm 0.001$           |
> | RL (real)                | ✓                    | ✗                    | $78.25 \pm 1.12$                  | $0.00 \pm 0.00$                     | $1.001 \pm 0.002$           |
> | RL (synthetic)    | ✓                    | ✓                    | $77.95 \pm 1.03$                  | $0.00 \pm 0.00$                     | $0.998 \pm 0.003$           |
> | DELETE (real)            | ✓                    | ✗                    | $78.37 \pm 1.12$                  | $0.00 \pm 0.00$                     | $1.002 \pm 0.001$           |
> | DELETE (synthetic)| ✓                    | ✓                    | $78.33 \pm 1.13$                  | $0.00 \pm 0.00$                     | $1.002 \pm 0.001$           |
> | NG+  (real)              | ✗                    | ✗                    | $78.47 \pm 1.05$                  | $0.00 \pm 0.00$                     | $1.003 \pm 0.002$           |
> | NG+ (synthetic)   | ✓                    | ✓                    | $78.34 \pm 1.10$                  | $0.00 \pm 0.00$                     | $1.002 \pm 0.001$           |
> | SCRUB (real)             | ✗                    | ✗                    | $77.61 \pm 1.01$                  | $0.00 \pm 0.00$                     | $0.995 \pm 0.003$           |
> | SCRUB (synthetic) | ✓                    | ✓                    | $78.26 \pm 1.04$                  | $0.00 \pm 0.00$                     | $1.001 \pm 0.002$           |

---

> ### Author Response · Authors · 2025-11-21
> **Response to Reviewer KPAB (Part 3)**
>
> **Multi-class (5 forget classes) unlearning performance on CIFAR-100 using ResNet-18. (real) rows use real embeddings and (synthetic) rows use synthetic embeddings:**
> | Method            | $\mathcal{D}_r$-free | $\mathcal{D}_f$-free | $\mathcal{A}^t_r$ (%)$\uparrow$        | $\mathcal{A}^t_f$(%) $\downarrow$        | AUS $\uparrow$              |
> |:-----------------|:--------------------:|:--------------------:|:---------------------------------:|:-----------------------------------:|:---------------------------:|
> | Original          | --                   | --                   | $78.14 \pm 1.26$                  | $78.68 \pm 0.92$                    | $0.560 \pm 0.003$           |
> | FT (real)                | ✗                    | ✓                    | $78.63 \pm 1.22$                  | $0.00 \pm 0.00$                     | $1.005 \pm 0.003$           |
> | FT (synthetic)    | ✓                    | ✓                    | $78.60 \pm 1.18$                  | $0.00 \pm 0.00$                     | $1.005 \pm 0.001$           |
> | NG (real)                | ✓                    | ✗                    | $78.67 \pm 1.19$                  | $0.00 \pm 0.00$                     | $1.005 \pm 0.001$           |
> | NG (synthetic)    | ✓                    | ✓                    | $78.68 \pm 1.13$                  | $0.00 \pm 0.00$                     | $1.005 \pm 0.002$           |
> | RL (real)                | ✓                    | ✗                    | $78.10 \pm 1.07$                  | $0.00 \pm 0.00$                     | $1.000 \pm 0.003$           |
> | RL (synthetic)    | ✓                    | ✓                    | $76.25 \pm 0.81$                  | $0.04 \pm 0.09$                     | $0.981 \pm 0.011$           |
> | DELETE (real)            | ✓                    | ✗                    | $78.71 \pm 1.14$                  | $0.00 \pm 0.00$                     | $1.006 \pm 0.001$           |
> | DELETE (synthetic)| ✓                    | ✓                    | $78.66 \pm 1.15$                  | $0.00 \pm 0.00$                     | $1.005 \pm 0.001$           |
> | NG+ (real)               | ✗                    | ✗                    | $78.79 \pm 1.04$                  | $0.00 \pm 0.00$                     | $1.006 \pm 0.003$           |
> | NG+ (synthetic)   | ✓                    | ✓                    | $78.63 \pm 1.11$                  | $0.00 \pm 0.00$                     | $1.005 \pm 0.002$           |
> | SCRUB (real)             | ✗                    | ✗                    | $78.27 \pm 1.05$                  | $0.00 \pm 0.00$                     | $1.001 \pm 0.003$           |
> | SCRUB (synthetic) | ✓                    | ✓                    | $78.48 \pm 1.12$                  | $0.00 \pm 0.00$                     | $1.003 \pm 0.003$           |

---

> ### Author Response · Authors · 2025-11-21
> **Response to Reviewer KPAB (Part 4)**
>
> **Multi-class (10 forget classes) unlearning performance on CIFAR-100 using ResNet-18. (real) rows use real embeddings and (synthetic) rows use synthetic embeddings:**
> | Method            | $\mathcal{D}_r$-free | $\mathcal{D}_f$-free | $\mathcal{A}^t_r$(%)$\uparrow$        | $\mathcal{A}^t_f$(%)$\downarrow$        | AUS $\uparrow$              |
> |:-----------------|:--------------------:|:--------------------:|:---------------------------------:|:-----------------------------------:|:---------------------------:|
> | Original          | --                   | --                   | $78.01 \pm 1.31$                  | $79.58 \pm 0.93$                    | $0.557 \pm 0.003$           |
> | FT (real)                | ✗                    | ✓                    | $78.80 \pm 1.04$                  | $0.02 \pm 0.04$                     | $1.008 \pm 0.004$           |
> | FT (synthetic)    | ✓                    | ✓                    | $78.88 \pm 1.17$                  | $0.02 \pm 0.04$                     | $1.009 \pm 0.002$           |
> | NG (real)                | ✓                    | ✗                    | $78.97 \pm 1.20$                  | $0.00 \pm 0.00$                     | $1.010 \pm 0.002$           |
> | NG (synthetic)    | ✓                    | ✓                    | $78.99 \pm 1.12$                  | $0.00 \pm 0.00$                     | $1.010 \pm 0.002$           |
> | RL (real)                | ✓                    | ✗                    | $78.62 \pm 1.10$                  | $0.00 \pm 0.00$                     | $1.006 \pm 0.003$           |
> | RL (synthetic)    | ✓                    | ✓                    | $74.38 \pm 1.41$                  | $0.18 \pm 0.35$                     | $0.962 \pm 0.014$           |
> | DELETE (real)            | ✓                    | ✗                    | $79.01 \pm 1.13$                  | $0.00 \pm 0.00$                     | $1.010 \pm 0.002$           |
> | DELETE (synthetic)| ✓                    | ✓                    | $78.96 \pm 1.14$                  | $0.66 \pm 1.01$                     | $1.003 \pm 0.010$           |
> | NG+ (real)               | ✗                    | ✗                    | $79.14 \pm 1.02$                  | $0.00 \pm 0.00$                     | $1.011 \pm 0.003$           |
> | NG+ (synthetic)   | ✓                    | ✓                    | $78.97 \pm 1.13$                  | $0.00 \pm 0.00$                     | $1.010 \pm 0.002$           |
> | SCRUB (real)             | ✗                    | ✗                    | $78.93 \pm 1.07$                  | $0.00 \pm 0.00$                     | $1.009 \pm 0.003$           |
> | SCRUB (synthetic) | ✓                    | ✓                    | $78.52 \pm 1.08$                  | $0.00 \pm 0.00$                     | $1.005 \pm 0.004$           |

---

> ### Author Response · Authors · 2025-11-21
> **Response to Reviewer KPAB (Part 5)**
>
> ### **Question 1. Comparison with real feature embeddings**
> ---
> We confirm that we have indeed compared our synthetic embeddings against real embeddings. In Tables 1 and 2, the non-shaded rows explicitly report the performance of unlearning methods using original (real) embeddings extracted from the actual forget/retain samples. Comparing the shaded rows (synthetic) with the non-shaded rows (real) reveals that the performance is nearly identical across all metrics. This empirical evidence validates our proposition that synthetic embeddings sampled from the intermediate space are sufficient surrogates for real data, functioning as effectively as the model’s actual feature representations for unlearning.
>
> ### **Question 2. Applicability to other tasks**
> ---
> This is an important question. We interpret "detection" here as Object Detection, where the goal is to localize and classify instances. While this work focuses on classification, the core principle of our framework—sampling synthetic embeddings for doing class unlearning—is theoretically transferable to detection architectures. In these models, the classification head typically operates on region proposals or grid cells. Our method could be adapted to unlearn specific classes by extracting synthetic embeddings corresponding to the detector's proposal features, and applying the unlearning loss to the class logits of those proposals. However, detection introduces additional complexities, such as handling the regression heads and preserving the spatial structure of feature maps, which differ from the global pooling used in classification. We view adapting our source-free sampling mechanism to these spatial constraints as non-trivial, but as a possible direction for future work.
>
> Additionally, we experimentally explored one application of our source-free framework for model debiasing. We conducted a new post-hoc debiasing experiment on CelebA. We trained a ResNet-18 with two heads predicting Earrings and Gender. The resulting model exhibits a substantial intersectional bias across the “gender × earrings’’ subgroups: for test images with earrings, accuracy is 79.33% for females but only 30.77% for males; for images without earrings, accuracy is 89.85% for females and 99.64% for males. We then applied our source-free unlearning framework with the Negative Gradient+ method to this already-trained model using only synthetic intermediate embeddings, targeting the minority “male & earring” subgroup. Synthetic embeddings classified as “male & earring’’ define the retain set, while for the other three groups, we split synthetic embeddings by model confidence—high-confidence samples are kept as retain, and low-confidence samples serve as forget samples. After four unlearning epochs, accuracy for “male with earrings’’ improves from 30.77% → 81.32%, while performance on the remaining subgroups remains high (please see the table below). This shows that our framework can be used as a privacy-preserving, post-hoc fairness correction tool, appropriate when retraining is infeasible or original images cannot be accessed. In the table, we report test accuracy (%) of the original model and debiased model after debiasing with our source-free class unlearning.
>
> | Class              | Acc. of the original model(%)                     |                   Acc. of the debiased model(%) |
> |--------------------|:---------------------------:|:------------------------:|
> | Earring & male         | 30.77         |         81.32    |
> | Earring & female      | 79.33         |         74.08    |
> | No earring & male    | 99.64         |         95.29    |
> | No earring & female | 89.85         |         90.69    |

---

### Official Review · Reviewer_ANbF · 2025-10-31

**Soundness:** 3
**Presentation:** 2
**Contribution:** 2
**Rating:** 2
**Confidence:** 4

**Summary:**

The paper shows that class unlearning can be achieved without any data by randomly sampling intermediate embeddings, pseudo-labeling them, and applying existing unlearning losses, making class unlearning truly source-free, simple, and computationally efficient. Overall, while the idea is simple and empirically effective, the paper’s novelty is limited, theoretical analysis is shallow, and the motivation for random embeddings is not sufficiently justified. Strengthening these aspects could significantly improve the manuscript.

**Strengths:**

(1)The proposed method can be universally compatible with existing unlearning methods.

(2)Strong empirical performance across multiple datasets and architectures.

**Weaknesses:**

(1)Limited methodological novelty. Although the paper frames the proposed approach as a universal source-free unlearning framework, the core methodology essentially acts as a “wrapper” that adapts existing unlearning methods to a data-free setting. The central idea, randomly sampling embeddings from intermediate feature space and applying pseudo-labeling, is conceptually simple and lacks technical depth. No new loss function, model design, or optimization technique is introduced, making the methodological innovation relatively limited.

(2)The theoretical justification relies heavily on a monotonicity assumption on the logits with respect to the forget loss. This assumption is extremely weak and is satisfied by almost all existing gradient-based unlearning methods. Consequently, the theory provides limited explanatory power or insight into why random embeddings should be effective in practice. The current analysis feels more like a sanity check than a deep theoretical contribution.

(3)The paper lacks a strong motivation for why random embeddings should be preferable or even comparable to learned or generative embeddings. Existing source-free methods (e.g., GKT, DSDA) invest effort in synthesizing realistic or adversarial samples for a reason, namely, to approximate meaningful decision boundaries. This work assumes that random vectors are sufficient but does not provide a compelling argument or empirical analysis to explain why this is the case. As a result, the motivation behind the proposed approach feels shallow and underdeveloped.

(4)Why are random embeddings sufficient? The results suggest that using arbitrary embeddings is enough to drive decision boundary updates, but the manuscript does not explore why this is true empirically. Are there situations where random embeddings would fail (e.g., highly overfitted models or extremely imbalanced forget classes)?

(5)The current framework is class-specific and relies on pseudo-labeling; it is unclear whether this mechanism can be extended to instance-level settings, which are arguably more practical and challenging.

(6)While the reported results are promising, the main results section suffers from over-claiming, limited mechanistic analysis, lack of robustness discussion, and insufficient explanation of why random embeddings work as well as real embeddings.

(7)In figure 2, the paper shows an empirical trend that performance improves with more synthetic embeddings and saturates after a certain point, but offers no deeper reasoning for why 100–200 random embeddings are sufficient to approximate the decision boundaries of real classes. Without theoretical insights or geometric analysis of the embedding space, the conclusion remains largely observational.

**Questions:**

(1)Insufficient theoretical depth.

(2)Only supports class-level unlearning.

(3)Does the proposed method have failure-case?

---

> ### Author Response · Authors · 2025-11-21
> **Response to Reviewer ANbF (Part 1)**
>
> We thank the reviewer for carefully reading our manuscript and for acknowledging both the universality of our framework and its strong empirical performance across datasets and architectures. We appreciate the thoughtful comments regarding methodological novelty, theoretical depth, and the motivation behind using randomly generated embeddings. We address each concern below.
>
> ### **Weakness (1)— Limited methodological novelty**
> ---
> Our goal is not to introduce a new unlearning loss, but to change the setting in which existing class-unlearning methods can operate. Traditionally, class unlearning methods require access to retain/forget data or rely on input-space generators. Our contribution is to make these methods truly source-free while keeping them compatible with a wide family of unlearning losses.
>
> The resulting framework is intentionally simple:
> 1) Sample synthetic feature vectors from an intermediate feature space.
> 2) Pseudo-label them using the deployed classifier.
> 3) Apply any differentiable class-unlearning loss directly on these synthetic embeddings.
>
> We believe this “wrapper-style” design is a feature, not a limitation: it allows widely used losses (e.g., Negative Gradient, Boundary Shrink, SCRUB, etc.) to function in a fully source-free setting without any modification.
>
> Conceptually, the key insight is that real or generative inputs are not strictly necessary for class unlearning. Under mild conditions, random feature-space sampling combined with pseudo-labeling induces the same decision-boundary dynamics as data-driven approaches. This observation is supported both by theory (distribution-agnostic gradient behavior) and by extensive empirical evidence across four backbones and three datasets. Thus, while the mechanics are simple, we believe the contribution is meaningful: we extend the applicability of existing unlearning methods to a strictly data-free regime and show that this can be done reliably and efficiently.
>
> ### **Weakness (2) & Question (1)— Limited theoretical depth**
> ---
> We believe the simplicity of the proposed framework and the needs of “extremely weak assumptions” add value to our work because we are showing that randomly generated embeddings labeled by the model as forget and retain are enough for doing class unlearning under the mentioned weak assumptions. This enables all the existing class unlearning methods without any need for the real samples, which makes them practical.
>
> The goal of Proposition 1 is to formalize the key design insight behind our method. More concretely, under the monotonicity assumptions, it shows that embeddings predicted as the forget class always push the forget logit down and push retain logits up, shrinking the forget class decision region regardless of the distribution from which those embeddings were sampled. The conceptual contribution is thus a distribution-agnostic guarantee: Any embedding distribution that produces samples classified as the forget class is theoretically sufficient for effective class unlearning. This gives nontrivial explanatory insight: as long as synthetic embeddings fall inside the forget-class decision region, they induce the same logit shrinkage/expansion gradients as real embeddings. Hence, learned or generative embeddings become unnecessary.
>
> ### **Weakness (3)— Insufficient motivation for random embeddings**
> ---
> We have strengthened the motivation in the revised version. Precisely by Proposition 1, under the notion of “an arbitrary embedding distribution $p_{\mathbf{z}}(\mathbf{z})$”, we are showing that any embedding—real, synthetic, off-manifold—exerts the same forget-class-shrinking effect as long as the classifier predicts it as belonging to the forget class. Random embeddings are simply a computationally lightweight mechanism for probing the decision regions of the model. Empirically:
>
> &#9679; Tables 1–2 show that random embeddings achieve unlearning performance comparable to using real embeddings.
>
> &#9679; Appendix B shows that Gaussian, Laplace, and Uniform embeddings behave nearly identically.
>
> &#9679; Figure 2 shows that performance saturates once the forget region receives sufficient coverage.
>
> Together, these observations provide a principled and empirical justification for using random embeddings instead of learned generators.

---

> ### Author Response · Authors · 2025-11-21
> **Response to Reviewer ANbF (Part 2)**
>
> ### **Weakness (4) & Question (3)— Why random embeddings work and failure cases**
> ---
> With Proposition 1, we are showing that embeddings generated with an arbitrary distribution and pseudo-labeled by the classifier as forget/retain are sufficient for doing class unlearning (under weak assumptions mentioned in the paper). As the failure of the proposed approach, we can explore two cases: (i) the number of generated embeddings per forget and retain classes, (ii) the classifier confidence level in assigning them to the forget/retain classes. Regarding the first case, Figure 2 and Appendix C show that unlearning performance degrades when the number of synthetic embeddings is too small. Regarding the second case, we performed a new experiment: On CIFAR-10 with ResNet-18, we varied the confidence thresholds used to select synthetic embeddings for constructing the pseudo-labeled forget/retain sets. We observe that thresholding mainly changes the convergence speed rather than the final unlearning outcome: for the DELETE method, when forgetting class 0, high-confidence synthetic samples reach optimal AUS after 11 epochs, whereas lower-confidence samples require 21 epochs. A similar trend holds for NG+, which needs 20 epochs with high-confidence pseudo-labels and 17 epochs with lower-confidence ones. Importantly, despite these differences in convergence speed, the final unlearning performance is essentially unchanged. This indicates that our method is robust to pseudo-label noise as long as a reasonable number of synthetic embeddings are available.
>
> ### **Weakness (5) & Question (2)— Not supporting instance-level unlearning**
> ---
> We agree that instance-level unlearning is both important and challenging. Our work explicitly targets class-level unlearning in a strictly source-free setting where no real forget samples or stored embeddings are available. This setting aligns with prior source-free works (e.g., Chundawat et al., 2023; Zhang et al., 2025; Wang et al., 2025) that also focus on forgetting classes or attributes. In addition, incorporating the current framework for doing instance-level unlearning was mentioned as a possible future direction of our work, as instance-level unlearning without any access to the specific samples to be removed—or any identifier of them—is fundamentally underdetermined: without the instance, it is unclear which region of parameter space should be modified. Therefore, instance-level unlearning typically requires some instance handle (real sample, stored embedding, or metadata), which is orthogonal to our contribution. Still, the mechanism is not inherently restricted to classes: if an auxiliary classifier or attribute tag identifies a subset of instances, our pseudo-labeling strategy naturally extends to forgetting that attribute or identity.
>
> [1] Vikram S Chundawat, Ayush K Tarun, Murari Mandal, and Mohan Kankanhalli. Zero-shot machine unlearning. IEEE Transactions on Information Forensics and Security, 18:2345–2354, 2023.
>
> [2] Chenhao Zhang, Shaofei Shen, Weitong Chen, and Miao Xu. Toward efficient data-free unlearning. In Proceedings of the AAAI Conference on Artificial Intelligence, volume 39, pp. 22372–22379, 2025.
>
> [3] Xiuyuan Wang, Chaochao Chen, Weiming Liu, Xinting Liao, Fan Wang, and Xiaolin Zheng. Efficient source-free unlearning via energy-guided data synthesis and discrimination-aware multitask optimization. In Forty-second International Conference on Machine Learning, 2025.

---

> > ### Comment · Reviewer_ANbF · 2025-11-26
> >
> > In the real unlearning scenario, compared with the "class-wise forgetting", "random data forgetting" is more universal and requested by the users. However, the proposed method is only suitable in the limited class-wise forgetting scenario. I think this is a big limitation of its real application.

---

> > > ### Comment · Reviewer_ANbF · 2025-11-26
> > >
> > > Regarding the embedding selection, it appears to be a random solution, where no specific rule or knowledge is used to determine this number. So when facing a new dataset, how to quickly obtain the optimal one.

---

> > > > ### Author Response · Authors · 2025-11-27
> > > > **Response to Reviewer ANbF**
> > > >
> > > > We thank the reviewer for the question. The purpose of Figures 2–6 is to provide an ablation study on the number of generated synthetic embeddings per class, demonstrating that our framework can successfully perform unlearning even with a relatively small number of synthetic samples. The goal of this analysis is not to determine a dataset-specific “optimal” value, but to show how performance behaves as coverage increases.
> > > >
> > > > Empirically, we observe a consistent and intuitive pattern, (i) for datasets where each class spans a larger or more diverse region in feature space (e.g., CIFAR-10 with ~5,000 real samples per class), effective unlearning requires more synthetic embeddings to adequately cover the forget region, with saturation around 800–1,000 samples; (ii) for datasets with smaller or narrower class regions (e.g., CIFAR-100 with ~500 samples per class), unlearning reaches saturation much earlier, at approximately 100–200 synthetic embeddings.
> > > >
> > > > Thus, the experiments show that our method does not require precise tuning: even modest sample counts are sufficient, and saturation naturally aligns with the scale of each class in the underlying dataset. For a new dataset, this empirical trend suggests a simple and broadly applicable rule of thumb: generating a number of synthetic embeddings on the same order as the number of real samples per class is effective in practice.

---

> > > ### Author Response · Authors · 2025-11-27
> > > **Response to Reviewer ANbF**
> > >
> > > We thank the reviewer for raising this important point about the scope of unlearning objectives. We agree that instance-level or “random data” unlearning is a valuable setting. However, we respectfully clarify that class-level unlearning is itself a widely studied and practically relevant unlearning regime, and our work follows the well-established definition used in recent approximate machine unlearning literature (Kurmanji et al., 2023; Tarun et al., 2023; Chundawat et al.,2023; Kodge et al., 2024; Panda et al., 2025; Seo et al., 2025).
> > >
> > > Importantly, class- or attribute-level unlearning is highly practical in real-world deployed systems. As an example we can name fairness corrections, which require suppressing the influence of a sensitive attribute. Below is an example we conducted accordingly to demonstrate how our source-free framework applies to a realistic and impactful scenario, we did a new post-hoc debiasing experiment on CelebA. We trained a ResNet-18 with two heads predicting Earrings and Gender. The resulting model exhibits a substantial intersectional bias across the “gender × earrings’’ subgroups: for test images with earrings, accuracy is 79.33% for females but only 30.77% for males; for images without earrings, accuracy is 89.85% for females and 99.64% for males. We then applied our source-free unlearning framework with the Negative Gradient+ method to this already-trained model using only synthetic intermediate embeddings, targeting the minority “male & earring” subgroup. Synthetic embeddings classified as “male & earring’’ define the retain set, while for the other three groups, we split synthetic embeddings by model confidence—high-confidence samples are kept as retain, and low-confidence samples serve as forget samples. After four unlearning epochs, accuracy for “male with earrings’’ improves from 30.77% → 81.32%, while performance on the remaining subgroups remains high (please see the table below). This shows that our framework can be used as a privacy-preserving, post-hoc fairness correction tool, appropriate when retraining is infeasible or original images cannot be accessed. In the table, the report test accuracy (%) of the original model and after debiasing with our source-free class unlearning.
> > >
> > > | Class              | Acc. of the original model(%)                     |                   Acc. of the debiased model(%) |
> > > |--------------------|:---------------------------:|:------------------------:|
> > > | Earring & male         | 30.77         |         81.32    |
> > > | Earring & female      | 79.33         |         74.08    |
> > > | No earring & male    | 99.64         |         95.29    |
> > > | No earring & female | 89.85         |         90.69    |
> > >
> > > [1] Meghdad Kurmanji, Peter Triantafillou, Jamie Hayes, and Eleni Triantafillou. Towards unbounded machine unlearning. Advances in neural information processing systems, 36:1957–1987, 2023.
> > >
> > > [2] Ayush K Tarun, Vikram S Chundawat, Murari Mandal, and Mohan Kankanhalli. Fast yet effective machine unlearning. IEEE Transactions on Neural Networks and Learning Systems, 35(9): 13046–13055, 2023.
> > >
> > > [3] Sangamesh Kodge, Gobinda Saha, and Kaushik Roy. Deep unlearning: Fast and efficient gradient free class forgetting. Transactions on Machine Learning Research, 2024.
> > >
> > > [4] Subhodip Panda, Shashwat Sourav, et al. Partially blinded unlearning: Class unlearning for deep networks from bayesian perspective. In Proceedings of the AAAI Conference on Artificial Intelligence, volume 39, pp. 6372–6380, 2025.
> > >
> > > [5] Vikram S Chundawat, Ayush K Tarun, Murari Mandal, and Mohan Kankanhalli. Can bad teaching induce forgetting? unlearning in deep networks using an incompetent teacher. In Proceedings of the AAAI Conference on Artificial Intelligence, pages 7210–7217, 2023.

---

> ### Author Response · Authors · 2025-11-21
> **Response to Reviewer ANbF (Part 3)**
>
> ### **Weakness (6)— Over-claiming and lack of a mechanistic explanation**
> ---
> We have strengthened these aspects in the revised manuscript. More precisely:
>
> **Mechanistic explanation:** As shown by Proposition 1, any embedding—real or synthetic—that is pseudo-labeled as the forget class induces the same boundary-shrinking update. The mechanism does not rely on the synthetic vectors lying on the real data manifold, but rather on populating the classifier’s decision region with sufficient synthetic samples.
>
> **Why random embeddings match real embeddings:** Since the gradient update depends only on pseudo-labels and logit derivatives, the marginal distribution of the sampled embeddings plays a limited role. As long as the synthetic samples fall into the forget-class region under the current model, they produce the correct unlearning gradients. Empirically, we observe that random embeddings from Gaussian, Laplace, and Uniform distributions (Appendix B) achieve essentially the same unlearning performance as real embeddings (Tables 1–2), confirming the distribution-agnostic behavior predicted by Proposition 1.
>
> **Robustness and saturation:** Effective unlearning occurs once the forget region is sufficiently covered by synthetic samples. This explains the empirical saturation at a certain number of embeddings in Figure 2: beyond this point, additional samples provide redundant gradients rather than new decision-boundary information. Appendix C further examines robustness with respect to embedding count, sampling distribution, and embedding layer depth. We also report failure cases showing that performance degrades when too few synthetic embeddings are generated, as the forget-region coverage becomes insufficient.
>
> ### **Weakness (7)— Why 100–200 embeddings suffice**
> ---
> We clarify this point more explicitly in the revised draft. Effective unlearning occurs once the synthetic embeddings sufficiently cover the decision region associated with the forget class. When this region is well covered, each synthetic point provides the correct boundary-shrinking gradient, and additional samples contribute little new information—explaining the empirical saturation observed in Figure 2. Beyond this threshold, further embeddings primarily generate redundant gradients rather than exposing new portions of the decision boundary.
>
> Importantly, in Figures 2-6, we observe that the number of synthetic embeddings required for adequate coverage naturally depends on the size and complexity of the underlying dataset. For larger or more diverse classes (e.g., CIFAR-10 with ~5,000 real samples per class), a higher number of synthetic embeddings is needed to approximate the extent of the forget region, with saturation occurring around 800–1,000 synthetic samples. In contrast, on datasets where each class occupies a smaller region in feature space (e.g., CIFAR-100 with ~500 samples per class), effective unlearning emerges with far fewer synthetic embeddings (≈100–200).

---

### Official Review · Reviewer_sU5Q · 2025-10-31

**Soundness:** 1
**Presentation:** 1
**Contribution:** 2
**Rating:** 2
**Confidence:** 3

**Summary:**

This paper considers the problem of source-free class unlearning. This work proposes to randomly generate intermediate features and utilize them for unlearning. A theoretical analysis is provided under an assumption that the forget loss function is monotonically increasing/decreasing with respect to the logit for forget/retain classes. Experimental results show the effectiveness of the proposed methods on image datasets.

**Strengths:**

+ Machine unlearning is a timely topic, and taking advantage of randomly generated intermediate features sounds interesting.

+ Experiments are conducted with both CNN and ViT backbones.

**Weaknesses:**

- The goal of this paper might not be considered to be "unlearning." The proposed method deliberately makes the model not to classify inputs to a target class, no matter what the inputs are. This still requires the model to be aware of the target class, to avoid classification to the target class. Rather, the definition in the intro that removing "the influence of specific instances or classes" would be more widely accepted definition of unlearning.

- Theoretical analysis is flawed. Eq. (6) is invalid, as it argues that a scalar is equal to a vector.

- It is not clear how Proposition 1 can be interpreted to draw new information.

- No explanation on the proposed method. L234--240 is not enough to understand the details of the method. For example, what is the probability distribution p_z(z)? If it is parametric, how is it determined? Figure 1 gives a hint that the output of the feature extractor is used, but there is no further description. Furthermore, if p_z(z) depends on the output of the feature extractor, then it is not really "randomly generated."

- No efficiency analysis on the proposed method. As this paper argues that "generating samples in the data space is computationally expensive," the proposed method should be compared with this setting to support this claim.

- Experimental setting is questionable. All baseline methods are using retain and/or forget class data, and adding the proposed method does not directly imply that they do not use such data anymore. That is, it is necessary to explain how they modified baseline methods to eliminate the necessity of retain/forget class data.

- No ablation study on the design choices of the proposed method.

- This paper employs a different font and it paragraph is more dense, compared to other ICLR submissions.

**Questions:**

What is the definition of unlearning, and why do you think so?

---

> ### Author Response · Authors · 2025-11-21
> **Response to Reviewer sU5Q (Part 1)**
>
> We thank the reviewer for the detailed comments. Several concerns stem from unclear exposition in the original draft, and we appreciate the opportunity to clarify and improve the manuscript. Below, we respond point-by-point.
>
> ### **&#9679; Weakness 1— Definition of unlearning**
> ---
> Our work adopts the standard and widely accepted definition of class unlearning used throughout recent approximate machine unlearning literature (for example, Kurmanji et al., 2023; Tarun et al., 2023; Kodge et al., 2024; Panda et al., 2025; Seo et al., 2025): a model is said to have unlearned a forget class if its predictions after unlearning are indistinguishable from those of a model retrained from scratch on the retain set. This is the definition implicitly or explicitly adopted in prior class unlearning works that use a Retrained-on-retain-only model as the gold-standard comparator, and evaluate through retain accuracy, forget accuracy, and composite metrics such as AUS. Our framework directly targets this retrained-equivalence notion, i.e., after applying unlearning methods, the model’s accuracy on the forget class collapses to the same level as the retrained baseline (≈0), while retain accuracy remains comparable, yielding AUS ≈ 1. It is correct that the model still contains a classifier parameter indexed by the forget class. However, this is also true for all existing approximate class unlearning methods, and even a model retrained on retain-only classes (which still has an output dimension for the removed class unless explicitly reparameterized). Thus, our work is fully consistent with the broadly used, retraining-equivalent definition of approximate class unlearning in prior work.
>
> [1] Meghdad Kurmanji, Peter Triantafillou, Jamie Hayes, and Eleni Triantafillou. Towards unbounded machine unlearning. Advances in neural information processing systems, 36:1957–1987, 2023.
>
> [2] Ayush K Tarun, Vikram S Chundawat, Murari Mandal, and Mohan Kankanhalli. Fast yet effective machine unlearning. IEEE Transactions on Neural Networks and Learning Systems, 35(9): 13046–13055, 2023.
>
> [3] Sangamesh Kodge, Gobinda Saha, and Kaushik Roy. Deep unlearning: Fast and efficient gradient free class forgetting. Transactions on Machine Learning Research, 2024.
>
> [4] Subhodip Panda, Shashwat Sourav, et al. Partially blinded unlearning: Class unlearning for deep networks from bayesian perspective. In Proceedings of the AAAI Conference on Artificial Intelligence, volume 39, pp. 6372–6380, 2025.
>
> [5] Seonguk Seo, Dongwan Kim, and Bohyung Han. Revisiting machine unlearning with dimensional alignment. In 2025 IEEE/CVF Winter Conference on Applications of Computer Vision (WACV), pp. 3206–3215. IEEE, 2025.
>
> ### **&#9679; Weakness 2— Validity of Eq. (6)**
> ---
> Respectfully, Eq. (6) is mathematically valid, and both sides are vectors of size $l$. Under the notation in Section 3.1, $\theta \in \mathbb{R}^{l\times C}$ denotes the classifier weight matrix. The left-hand side of Eq. (6) refers to the k-th column of the gradient of the loss w.r.t. $\theta$, which is a vector in $ \mathbb{R}^{l}$. On the right-hand side, the derivative $\frac{\partial \mathcal{L}_f}{\partial [h(g(\mathbf{z}_i))]_k}$ is a scalar and $g(\mathbf{z}_i) \in \mathbb{R}^{l}$ is a vector, so their product is a vector of length $l$. Therefore, both sides of Eq. (6) are indeed vectors in the same space.
>
> ### **&#9679; Weakness 3— Purpose and interpretation of Proposition 1**
> ---
> The goal of Proposition 1 is to formalize the key design insight behind our method. More concretely, under the monotonicity assumptions, it shows that embeddings predicted as the forget class always push the forget logit down and push retain logits up, shrinking the forget class decision region regardless of the distribution from which those embeddings were sampled. The conceptual contribution is thus a distribution-agnostic guarantee: Any embedding distribution that produces samples classified as the forget class is theoretically sufficient for effective class unlearning. This supports our central algorithmic choice: using simple priors (Gaussian, Laplace, Uniform) in an intermediate feature space, without approximating the original input distribution or training a generator. Appendix B (Tables 5–6) empirically confirms that unlearning performance is remarkably stable across embedding distributions, matching the theoretical statement.

---

> ### Author Response · Authors · 2025-11-21
> **Response to Reviewer sU5Q (Part 2)**
>
> ### **&#9679; Weakness 4— Definition and use of  $p_\mathbf{z}$**
> ---
> The theoretical analysis in Proposition 1 explicitly assumes that $p_\mathbf{z}$​ is an arbitrary distribution over the intermediate feature space. Our results do not require matching the true feature distribution nor computing statistics from the backbone. During unlearning, we simply draw synthetic embeddings $\mathbf{z} \sim p_\mathbf{z}$ with no adaptation. We experiment with several choices such as Standard Gaussian: $ \mathcal{N}(0, I)$, Uniform: $\mathcal{U}(-1, 1)$, and Laplace$(0,1)$. These distributions live in the same dimensional space as the backbone features but do not depend on actual feature extractor outputs. Appendix Tables 5 and 6 show that performance is insensitive to the choice of $p_\mathbf{z}$​, consistent with our distribution-agnostic Proposition 1.
>
> ### **&#9679; Weakness 5— Efficiency analysis**
> ---
> Our statement that “generating samples in the data space is computationally expensive” is specifically aimed at source-free unlearning methods that rely on image-level generators. Such approaches require, for each synthetic sample: (i) forward (and sometimes backward) passes through a large generator network at input resolution, (ii) followed by passes through the task model. By contrast, our method works entirely in the feature space where synthetic embeddings are merely drawn randomly from $\mathbf{z} \sim p_\mathbf{z}$​.
> Beyond the computational burdens, a key limitation of existing source-free unlearning approaches based on data-space generation is that they are usually tailored to a specific generator architecture and unlearning objective; the “source-free” mechanism is tightly coupled to that particular design and does not readily transfer to other unlearning methods. Our framework is intentionally more general: it is agnostic to the chosen class-unlearning loss and can, in principle, be combined with any differentiable objective loss function simply by replacing real inputs with synthetic intermediate features sampled from $p_\mathbf{z}$​.
>
> ### **&#9679; Weakness 6— Experimental setting and use of baselines**
> ---
> Tables 1–2 distinguish clearly between:
>
> &#9679; baseline methods using real retain/forget data (non-shaded rows),
>
> &#9679; and our source-free variants using only synthetic embeddings (shaded rows).
>
> In the source-free setting, we replace the real forget/retain datasets with synthetic pseudo-labeled sets obtained exactly as in Algorithm 1. The loss functions and optimization rules of each baseline remain unchanged; only the inputs are replaced with our synthetic intermediate embeddings. This ensures a fair evaluation of how each unlearning method performs in a fully source-free regime.
>
> ### **&#9679; Weakness 7 — Ablation studies**
> ---
> The paper already contains multiple ablations:
>
> 1) Embedding location (Sec. 4.2): We compare synthetic embeddings taken immediately before the classifier versus earlier layers.
>
> 2) Number of synthetic samples (Fig. 2, Appendix C): Performance saturates after a moderate number of synthetic embeddings.
>
> 3) Distribution of synthetic embeddings (Appendix B): Unlearning effectiveness is robust across Gaussian, Laplace, and Uniform distributions.
>
> 4) Uncertainty of pseudo-labeled embeddings (new in the revised version): We vary the confidence thresholds used to select synthetic intermediate embeddings for forget/retain sets. While high- vs. low-confidence thresholds mainly affect the number of epochs needed to reach optimal unlearning, the final AUS, retain accuracy, and forget accuracy remain essentially unchanged, indicating that our method is robust to the choice of confidence threshold.
>
> ### **&#9679; Weakness 8— Formatting**
> ---
> We acknowledge the formatting issue: the draft inadvertently used incorrect font and paragraph settings compared to the ICLR template, resulting in denser paragraphs. This has been corrected in the revised version, which now fully adheres to the official ICLR style.
>
> ### **&#9679; Question 1— Definition of unlearning**
> ---
> Our definition follows the standard practice in recent approximate class-unlearning literature (for example, Kurmanji et al., 2023; Tarun et al., 2023; Kodge et al., 2024; Panda et al., 2025; Seo et al., 2025): A model has unlearned a class if, after the unlearning procedure, its predictions on new data are indistinguishable from those of a model retrained on the retain set only, while maintaining performance on the retain classes. This definition underlies all works that use the retrained-on-retain-only model as the gold-standard reference. Under this criterion, our method successfully brings forget-class accuracy to the retrained baseline (~0) while preserving retain performance (AUS ≈ 1), meaning that the model has effectively lost its ability to recognize the forgotten class on unseen data.

---

### Official Review · Reviewer_bMou · 2025-11-03

**Soundness:** 3
**Presentation:** 3
**Contribution:** 3
**Rating:** 8
**Confidence:** 3

**Summary:**

This paper proposes a source-free class unlearning, utilizing the embedding in the intermediate space. It performs classification on these embeddings and generates pseudo-labels to construct synthetic forget and retain sets. And then use these sets for later training. The authors theoretically justify that effective unlearning can be achieved independent of the embedding distribution.  Empirical studies on  multiple datasets and backbones also demostrate the effectiveness of the proposed method. Further experiments show that existing unlearning algorithms can be seamlessly adapted to this source-free setting with minimal performance degradation.

**Strengths:**

1. the proposed method is simple but effective. The idea of leveraging intermediate embedding for creating the forget and retain sets is very good and effective.
2. the theoretical proof strengthen the paper's contribution
3. It is very good that the method could be easily applied to existing methods without performance degrade.
4. writing is neat and easy to follow.

**Weaknesses:**

1. It would be better if the experiments could address some critical applications that unlearning is suitable.
2.  Many results show near-perfect AUS (~1.00), raising concerns about the sensitivity or discriminative power of the evaluation metric under this setting.
3.  it would be better to do some analysis regarding the pseudo-label。
4. novelty could be a problem as it is essnetially create forget and retain sets and access them in a traditional way.

**Questions:**

The questions are related to the weakness.

1. Could the authors provide examples or experiments demonstrating how the proposed method applies to critical real-world applications where unlearning is particularly useful?
2. How do the authors ensure that the AUS metric remains sensitive and discriminative in evaluating subtle performance differences?
3. Could the authors include an analysis of the pseudo-labels such as their reliability, stability, and effect on unlearning performance?

---

> ### Author Response · Authors · 2025-11-21
> **Response to Reviewer bMou (Part 1)**
>
> We thank the reviewer for the careful reading of our paper and the positive assessment of its soundness, presentation, and contribution. We greatly appreciate that you found the idea of leveraging intermediate embeddings for constructing forget/retain sets both simple and effective, and that the theoretical analysis and empirical results were convincing.
>
> ### **1. Weakness 1. & Question 1.— Critical applications**
> ---
> To demonstrate how our source-free framework applies to a realistic and impactful scenario, we conducted a new post-hoc debiasing experiment on CelebA. We trained a ResNet-18 with two heads predicting Earrings and Gender. The resulting model exhibits a substantial intersectional bias across the “gender × earrings’’ subgroups: for test images with earrings, accuracy is 79.33% for females but only 30.77% for males; for images without earrings, accuracy is 89.85% for females and 99.64% for males. We then applied our source-free unlearning framework with the Negative Gradient+ method to this already-trained model using only synthetic intermediate embeddings, targeting the minority “male & earring” subgroup. Synthetic embeddings classified as “male & earring’’ define the retain set, while for the other three groups, we split synthetic embeddings by model confidence—high-confidence samples are kept as retain, and low-confidence samples serve as forget samples. After four unlearning epochs, accuracy for “male with earrings’’ improves from 30.77% → 81.32%, while performance on the remaining subgroups remains high (please see the table below). This shows that our framework can be used as a privacy-preserving, post-hoc fairness correction tool, appropriate when retraining is infeasible or original images cannot be accessed. In the table, we report test accuracy (%) of the original model and debiased model after debiasing with our source-free class unlearning.
>
> | Class              | Acc. of the original model(%)                     |                   Acc. of the debiased model(%) |
> |--------------------|:---------------------------:|:------------------------:|
> | Earring & male         | 30.77         |         81.32    |
> | Earring & female      | 79.33         |         74.08    |
> | No earring & male    | 99.64         |         95.29    |
> | No earring & female | 89.85         |         90.69    |
>
> ### **2. Weakness 2. & Question 2.— Sensitivity of AUS**
>
> ---
>
> We follow the Adaptive Unlearning Score (AUS) introduced by Cotogni et al. (2023), which is specifically designed so that AUS ≈ 1 is achieved only when (i) retain accuracy after unlearning matches the original model and (ii) forget accuracy is close to the ideal target (≈ 0). In our setting, many methods indeed achieve this regime, which explains the near-perfect AUS values. Importantly, we always report AUS together with the underlying retain and forget accuracies and their standard deviations, which expose performance differences even when AUS is saturated. Moreover, in more challenging regimes—such as reducing the number of synthetic embeddings or using embeddings from early backbone layers (Figure 2, Table 3)—AUS deviates substantially from 1 and differentiates methods clearly. This shows that AUS remains sensitive and discriminative when the unlearning task becomes harder.
>
> [1] Cotogni et al., Duck: distance-based unlearning via centroid kinematics. arXiv:2312.02052, 2023.
>
> ### **3. Weakness 3. & Question 3.— Pseudo-label analysis**
> ---
> We conducted an additional analysis to assess how the reliability and stability of pseudo-labels affect unlearning performance within our source-free framework. On CIFAR-10 with ResNet-18, we varied the confidence thresholds used to select synthetic embeddings for constructing the pseudo-labeled forget/retain sets. We observe that thresholding mainly changes the convergence speed rather than the final unlearning outcome: for the DELETE method, when forgetting class 0, high-confidence synthetic samples reach optimal AUS after 11 epochs, whereas lower-confidence samples require 21 epochs. A similar trend holds for NG+, which needs 20 epochs with high-confidence pseudo-labels and 17 epochs with lower-confidence ones. Importantly, despite these differences in convergence speed, the final unlearning performance is essentially unchanged. This indicates that our method is robust to pseudo-label noise as long as a reasonable number of synthetic embeddings are available, and that the framework does not rely on a small subset of highly confident samples. This observation is consistent with our theory, which only requires manipulating class logits—not matching the true embedding distribution. Finally, when too few synthetic embeddings are generated, unlearning performance degrades, which we illustrate in Figure 2 of the main paper and Figures 3–6 in the appendix C. These experiments serve as failure-case evidence and further confirm that sample quantity, rather than pseudo-label confidence, is the critical factor.

---

> ### Author Response · Authors · 2025-11-21
> **Response to Reviewer bMou (Part 2)**
>
> ### **4. Weakness 4.— Novelty**
> ---
> Although the final optimization uses forget and retain sets, our contribution lies in how these sets are constructed. Existing source-free or retain-free unlearning approaches either (a) synthesize input-space data and then distill a new student, or (b) rely on specialized loss designs tied to specific data-access assumptions. In contrast, our framework shows that arbitrary random embeddings in an intermediate feature space, pseudo-labeled by the model itself, are sufficient for effective unlearning, independent of their marginal distribution (Proposition 1). This provides both (i) a new theoretical perspective on the minimal information needed for class unlearning, and (ii) a universal source-free framework that can wrap around a wide range of existing unlearning methods without retraining on real data or training synthetic generators, as evidenced by our experiments across four backbones and three datasets.

---

### Meta-Review · Area_Chair_L93k · 2026-01-07

**Summary:**

This work introduces a new source-free approach to class unlearning by leveraging intermediate feature representations to create synthetic training data for forgetting and retention. The method shows promising results across different datasets and model architectures, suggesting that effective unlearning can be achieved without access to original data. However, the paper lacks clarity in problem formulation and does not provide sufficient ablation studies to validate key design choices. Although the idea is reasonable, the current version requires stronger experimental support before being suitable for publication.

**Reviewer Concerns:**

While the idea is interesting, the paper lacks strong experimental validation, especially in demonstrating the practical relevance of unlearning in real-world scenarios. The methodological novelty appears limited, as the approach mainly adapts existing unlearning techniques rather than introducing fundamentally new ideas. The theoretical analysis and motivation are not sufficiently convincing to clearly support the proposed framework. Several aspects of the experimental design and evaluation raise concerns about the reliability of the conclusions. Overall, the work shows potential but requires clearer positioning, stronger analysis, and more thorough experiments to be considered ready for publication.

**Reviewer Scores:**

I think the reviewer who accepted the paper may lower its score after seeing all rebuttals, while others will maintain their scores.

---

### Decision · Program_Chairs · 2026-01-26

Reject